# Reliable Poisoned Sample Detection against Backdoor Attacks Enhanced by Sharpness Aware Minimization

**Mingda Zhang[1]  Mingli Zhu[1]  Zihao Zhu[1]  Li Shen[3]  Baoyuan Wu[1,2]***

[1]School of Artificial Intelligence, The Chinese University of Hong Kong, Shenzhen
[2]Shenzhen Loop Area Institute
[3]Shenzhen Campus of Sun Yat-sen University

## Abstract

This work investigates Poisoned Sample Detection (PSD), a promising defense approach against backdoor attacks. However, we observe that the effectiveness of many advanced PSD methods degrades significantly under weak backdoor attacks (*e.g.*, low poisoning ratios or weak trigger patterns). To substantiate this observation, we conduct a statistical analysis across various attacks and PSD methods, revealing a strong correlation between the strength of the backdoor effect and the detection performance. Inspired by this, we propose amplifying the backdoor effect through training with Sharpness-Aware Minimization (SAM). Both theoretical insights and empirical evidence validate that SAM enhances the activations of top Trigger Activation Change (TAC) neurons while suppressing others. Based on this, we introduce SAM-enhanced PSD, a simple yet effective framework that seamlessly improves existing PSD methods by extracting detection features from the SAM-trained model rather than the conventionally trained model. Extensive experiments across multiple benchmarks demonstrate that our approach significantly improves detection performance under both strong and weak backdoor attacks, achieving an average True Positive Rate (TPR) gain of +34.3% over conventional PSD methods. Overall, we believe that the revealed correlation between the backdoor effect and detection performance could inspire future research advancements.

## 1 Introduction

Deep Neural Networks (DNNs), while achieving remarkable success across a wide range of applications (Yatbaz et al., 2023; Panagoulias et al., 2024; Shu et al., 2024), are vulnerable to *backdoor attacks* (Wu et al., 2025). In such attacks, adversaries inject a small number of poisoned samples into the training dataset (Gu et al., 2019), enabling the model to produce targeted malicious outputs when a hidden trigger is present. This poses safety risks in security domains such as autonomous driving, where even a single malicious prediction can cause catastrophic consequences. Consequently, accurately identifying poisoned samples from training datasets is a fundamental defense objective.

**Observation.** To defend against data poisoning–based backdoor attacks, various pre-training stage poisoned sample detection (PSD) methods have been proposed (Wu et al., 2023). General pre-training stage PSD approaches typically involve training a model on the potentially poisoned dataset and exploiting performance disparities in the feature space between poisoned and clean samples for detection (Chen et al., 2019; Tang et al., 2021; Hayase et al., 2021; Tran et al., 2018; Yuan et al., 2025). However, recent studies show that PSD methods can be bypassed when the poisoning ratio is low or the trigger is weak (Tang et al., 2021; Qi et al., 2023a; Zhu et al., 2024). We argue that the underlying cause is a reduction in the **backdoor effect**, defined as the relative strength of trigger-induced neuron activations compared to activations from benign features, and measurable with the Trigger Activation Change (TAC) metric (Zheng et al., 2022). Notably, a weak backdoor effect does not necessarily imply a low attack success rate (ASR); in many cases, ASR remains high even as

---

*Corresponds to Baoyuan Wu (wubaoyuan@cuhk.edu.cn).

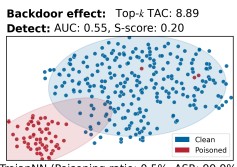 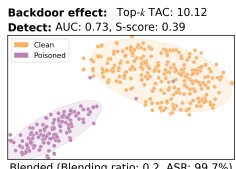 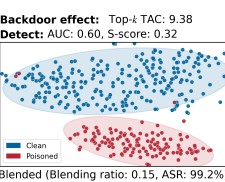

Figure 1: T-SNE visualizations of backdoor effects under high (clean, poisoned) and low (clean, poisoned) poisoning and blending ratios. Top-$k$ TAC (Trigger Activation Change) quantifies the backdoor effect, where $k = 30$. AUC is the average ROC-AUC across different detection methods, and S-score is the Silhouette coefficient measuring separation between clean and poisoned samples.

detection performance deteriorates significantly. As shown in Fig. 1, a strong backdoor effect yields well-separated feature clusters for poisoned and clean samples (first and third plots), facilitating detection, whereas a weak effect results in cluster overlap (second and fourth plots), thereby degrading PSD performance. Empirical evaluations across diverse attack configurations further confirm a strong positive correlation between Top-$k$ TAC and PSD performance (*Pearson correlation = 0.73*).

**Motivation and Method.** Building on the above observations, we address the challenge of detecting poisoned samples under weak backdoor conditions. Since defenders cannot alter trigger properties or poisoning ratios at the data level, we propose a new perspective: amplify the backdoor effect via the training of the model (used for feature extraction in PSD) to improve feature-space separability between clean and poisoned samples, thus enhancing PSD performance. Our approach adopts an optimization-based strategy inspired by the sparse feature property of Sharpness-Aware Minimization (SAM) (Andriushchenko et al., 2023). SAM encourages sparse neuron activations by amplifying dominant activations while suppressing weaker ones. Theoretical and empirical analyses show that SAM increases the activation of top TAC-ranked neurons (*i.e.*, neurons strongly associated with the backdoor trigger) while reducing the activation of others, thereby magnifying the backdoor effect. Leveraging this property, we design a SAM-enhanced, three-stage PSD pipeline: (1) train or fine-tune a model with SAM on the suspicious dataset; (2) extract backdoor-related features from intermediate activations (*e.g.*, TAC-ranked neurons); (3) apply an existing PSD detector that leverages feature-space disparities to identify poisoned samples. Note that our framework is model-agnostic and attack-independent, enabling seamless integration into existing PSD methodologies.

**Contributions.** Our work makes the following contributions: (1) We establish a strong positive correlation between backdoor effect and PSD effectiveness, demonstrating that weak backdoor effects significantly degrade detection performance. (2) We propose a simple yet effective SAM-based training method that amplifies the backdoor effect, thereby increasing feature-space separability and enhancing PSD performance. (3) We validate our approach through extensive experiments across diverse backdoor attack scenarios, showing that our SAM-enhanced pipeline consistently boosts multiple PSD methods, achieving an average True Positive Rate (TPR) gain of **+34.3%** over conventional PSD approaches.

## 2 RELATED WORK

**Backdoor attack.** Backdoor attacks aim to embed hidden malicious behaviors into models during training (Gu et al., 2019). Initial methods, like BadNets, add conspicuous triggers such as small patches to a subset of training data and alter their labels to a target class. Subsequent research focused on enhancing attack effectiveness and stealth through more diverse trigger designs, ranging from subtle image blends to modifications in the frequency domain (Chen et al., 2017; Li et al., 2021b; Zeng et al., 2021). To further evade detection, a significant line of work explores clean-label attacks, which preserve the original labels of poisoned samples (Turner et al., 2019). More advanced techniques even ensure that poisoned and clean samples are indistinguishable at the feature level, thus making them exceptionally difficult to detect (Tang et al., 2021; Qi et al., 2023a; Liang et al., 2024).

**Backdoor defense.** Backdoor defenses are categorized into four stages. Pre-training defenses (Chen et al., 2019; Gao et al., 2019; Qi et al., 2023a; Ma et al., 2022; Yao et al., 2024) proactively purify the dataset before the model learns malicious behaviors. In-training (Li et al., 2021a; Gao et al., 2023; Liu et al., 2023) and post-training (Li et al., 2023; Zhu et al., 2023; Hu et al., 2025; Wang

et al., 2024) defenses aim to build robust models or repair compromised ones, respectively. Inference-time defenses (Guo et al., 2023; Hou et al., 2024) offer flexibility by detecting attacks on-the-fly without altering the model. Our work focuses on the proactive pre-training stage. In this stage, most methods identify poisoned samples by finding statistical deviations in their feature representations. For instance, some leverage feature clustering or spectral analysis for this purpose (Chen et al., 2019; Hayase et al., 2021). Other paradigms exist, such as learning-based (Qi et al., 2023b) and perturbation-based detection (Huang et al., 2023; Pal et al., 2024). While most pre-training defenses are sophisticated detectors, our work introduces a novel enhancement paradigm. We leverage SAM during pre-training to widen the feature gap between clean and poisoned samples, making them easier to detect. This fundamentally contrasts with FT-SAM (Zhu et al., 2023), which uses a clean dataset for post-training model repair by suppressing backdoor effects. In contrast, our pre-training method uses the poisoned dataset to amplify these same effects to aid detection. Consequently, our method is not a new detector, but a plug-and-play module that improves the efficacy of existing ones.

**Sharpness aware minimization.** Sharpness-Aware Minimization (SAM) is a training technique that improves model generalization by seeking flat, rather than sharp, minima in the loss landscape (Foret et al., 2021). It achieves this by simultaneously minimizing both the loss value and the loss sharpness. Variants like ASAM (Kwon et al., 2021) and GSAM (Zhuang et al., 2022) offer alternative strategies to find these flat regions more efficiently. Beyond improving generalization, empirical studies have shown that SAM can also induce other beneficial properties in models, such as increased neuron sparsity and better compressibility (Andriushchenko et al., 2023).

## 3 METHOD

### 3.1 PROBLEM SETTING

**Threat model.** In this work, we consider data poisoning-based backdoor attacks, where the attacker releases a poisoned training dataset $\mathcal{D}_{tr}$ to implant a backdoor into any model trained on it. Starting from a clean dataset $\mathcal{D}_{cl} = \{(\boldsymbol{x}_i, y_i)\}_{i=1}^N \subset \mathcal{X} \times \mathcal{Y}$, where $\mathcal{X} \subset \mathbb{R}^d$ and $\mathcal{Y} = \{0, 1, \ldots, K-1\}$ represent the input space and label set, the attacker selects a subset $\mathcal{D}_{sub} \subset \mathcal{D}_{cl}$ to poison, with poisoning ratio $p = \frac{|\mathcal{D}_{sub}|}{|\mathcal{D}_{cl}|}$. Each poisoned sample is generated using a predefined trigger $\Delta$, a generation function $g$, and a target label $y_t$, resulting in $(\tilde{\boldsymbol{x}} = g(\boldsymbol{x}, \Delta), y_t)$. The poisoned set $\mathcal{D}_{poi}$ replaces $\mathcal{D}_{sub}$ in $\mathcal{D}_{cl}$ to form the final poisoned dataset $\mathcal{D}_{tr} = (\mathcal{D}_{cl} \setminus \mathcal{D}_{sub}) \cup \mathcal{D}_{poi}$.

**Defender's goal.** Defenders aim to detect the poisoned samples $\mathcal{D}_{poi}$ within the released training dataset $\mathcal{D}_{tr}$ without knowing attack details (*e.g.*, the poisoning ratio $p$, the trigger $\Delta$, and the generation function $g$), assuming defenders can access a few clean samples. A common defense pipeline first trains a model via a standard method (*e.g.*, SGD) to extract features, then applies a detection algorithm on the extracted features. Our work introduces a new paradigm by innovating in the first stage. Instead of passively accepting the features from standard training, we propose a new training methodology designed to amplify the feature discrepancy between clean and poisoned samples. This serves as a foundational enhancement, boosting the performance of various subsequent detection methods.

### 3.2 ANALYSIS BETWEEN BACKDOOR EFFECT AND DETECTION PERFORMANCE

**Backdoor effect measured by Top-$k$ TAC.** To measure backdoor effect, we use Trigger Activation Change (TAC) (Zheng et al., 2022) which quantifies the differences in activation values between poisoned samples and their corresponding clean samples in a deep neural network (DNN). We denote a DNN as $f_{\boldsymbol{\theta}} = f^{(L)} \circ f^{(L-1)} \circ \cdots \circ f^{(1)}$. Given a clean input sample $\boldsymbol{x}$ and its poisoned counterpart $\tilde{\boldsymbol{x}}$, TAC is computed using the following equation:

$$TAC_j^{(l)}(\mathcal{D}) = \frac{1}{|\mathcal{D}|} \sum_{\boldsymbol{x} \in \mathcal{D}} \left\| f_j^{(l)}(\boldsymbol{x}) - f_j^{(l)}(\tilde{\boldsymbol{x}}) \right\|_2, \tag{1}$$

where $j$ represents the $j$-th neuron in layer $l$, and $\mathcal{D}$ is the set of clean samples. According to the definition, the magnitude of TAC reflects the neuron's sensitivity to trigger. A higher TAC indicates that the neuron responds more strongly to trigger and can therefore be regarded as a *backdoor neuron*.

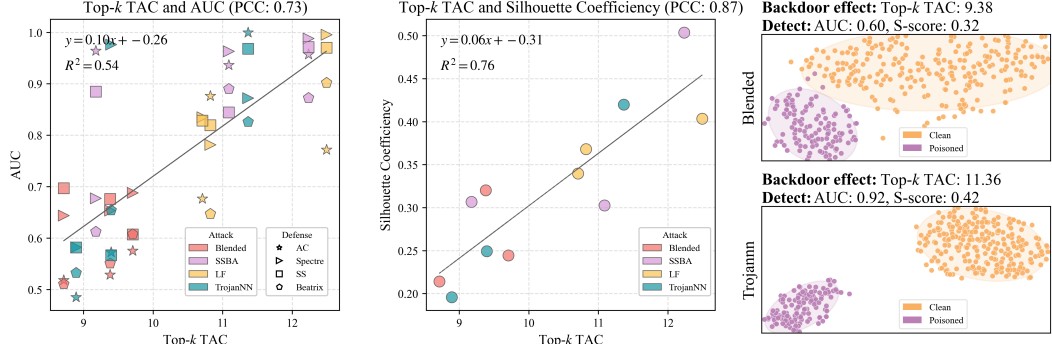

Figure 2: Correlation between backdoor effect and PSD performance on CIFAR-10 with ResNet-18 using $k = 30$ and poisoning ratios of 0.5%, 1% and 5%, quantified by Pearson correlation coefficient (PCC) and linear regression analysis. **Left**: AUC *vs.* Top-$k$ TAC across different attacks and defenses, showing a positive correlation. **Middle**: Silhouette coefficient *vs.* Top-$k$ TAC, indicating improved feature separability with stronger backdoor effect. **Right**: t-SNE visualizations showing that higher Top-$k$ TAC leads to better separation between clean (purple) and poisoned (orange) samples, aligned with improved detection. (S-score is the Silhouette coefficient.)

To quantify the overall backdoor effect in the model, we consider the Top-$k$ most responsive neurons in layer $l$. The Top-$k$ TAC is computed by averaging the TAC values of these top-$k$ neurons:

$$\text{Top-}k\ TAC^{(l)}(\mathcal{D}) = \frac{1}{|T_k|} \sum_{j \in T_k} TAC_j^{(l)}(\mathcal{D}), \tag{2}$$

where $T_k$ denotes the indices of the top-$k$ neurons with the highest TAC values in layer $l$. In our method, we specifically select the final convolutional layer for this computation, as it typically provides the most discriminative features for distinguishing between clean and poisoned samples.

**Backdoor effect and detection performance.** To explore the relationship between the backdoor effect and detection performance, we manipulate the strength of the backdoor effect by adjusting the poisoning ratio across various attacks, using CIFAR-10 dataset on ResNet-18. We measure the backdoor effect using Top-$k$ TAC and evaluate detection performance with AUC. As illustrated in Fig. 2, detection performance consistently declines as TAC decreases, with a Pearson correlation coefficient of 0.73, indicating a strong positive correlation. Regression analysis yields an R-squared[1] value of 0.54, indicating that TAC explains a significant portion of the variance in detection accuracy.

To further understand the impact of backdoor effect on detection, we compute the silhouette coefficient, a standard metric for evaluating how well two classes are separated in feature space. In our setting, it measures the separability between poisoned samples and target clean samples. As shown in Fig. 2, the silhouette coefficient increases with higher TAC, indicating that stronger backdoor effects lead to more distinct activation patterns between the two classes. The Pearson correlation coefficient between TAC and silhouette coefficient is 0.87, and the R-squared from regression analysis is 0.76, both supporting a clear linear relationship. These findings confirm that **stronger backdoor effects, as measured by higher Top-$k$ TAC, lead to more pronounced separation between poisoned and clean sample activations**, suggesting that amplifying TAC through training modifications can enhance detection effectiveness even without prior knowledge of the specific attack. We provide additional results on other models and datasets, and a statistical analysis of the activation gap in our TAC formulation, in Sec. D.6 and Sec. D.5, respectively.

### 3.3 ENHANCING BACKDOOR EFFECTS VIA SAM

**Backdoor learning with SAM.** To enhance the backdoor effect during training, we adopt the Sharpness-Aware Minimization (SAM) optimization algorithm. Prior studies (Springer et al., 2024; Andriushchenko et al., 2023) have shown that SAM improves feature structure, such as producing lower-rank representations. Intuitively, SAM guides the learning process toward more discriminative

---

[1]R-squared measures the proportion of the variance in the dependent variable that is predictable from the independent variables in a regression model. A higher R-squared indicates a better fit of the model to the data.

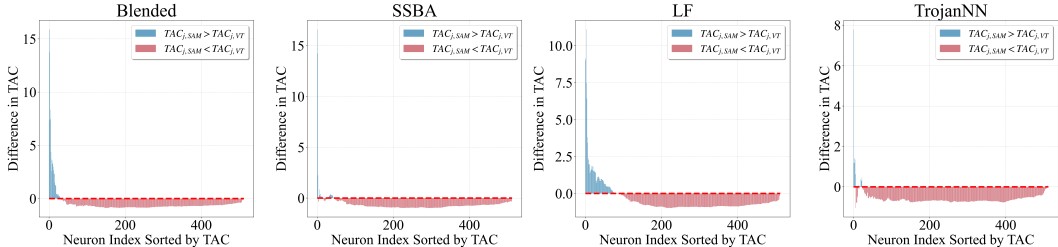

Figure 3: TAC differences between models trained with SAM and Vanilla Training. Neurons are sorted by their TAC (descending) in each model. Blue bars and red bars indicate increased and decreased TAC by SAM, respectively.

directions in the representation space, thereby amplifying the activations of neurons responsive to the features. The SAM objective is formulated as:

$$\min_{\boldsymbol{\theta}} \max_{\boldsymbol{\epsilon} \in \{\|\boldsymbol{\epsilon}\|_2 \le \rho\}} \mathcal{L}(\boldsymbol{\theta} + \boldsymbol{\epsilon}), \tag{3}$$

where $\mathcal{L}(\boldsymbol{\theta}) = \frac{1}{|\mathcal{D}_{tr}|} \sum_{(\boldsymbol{x},y) \in \mathcal{D}_{tr}} \ell(f_{\boldsymbol{\theta}}(\boldsymbol{x}; y))$ is the cross-entropy loss, and $\rho > 0$ is a hyperparameter that controls the budget for weight perturbations. To illustrate the effect of SAM on backdoor learning, we compare TAC values between models trained with SAM and Vanilla Training across multiple backdoor attacks. As shown in Fig. 3, SAM consistently increases the TAC values of neurons highly responsive to the trigger (*i.e.*, backdoor neurons), while reducing the TAC values of unrelated neurons. *These results suggest that SAM selectively amplifies the most discriminative backdoor features by encouraging sharper activation patterns, thereby enhancing the backdoor effect.*

**Theoretical analysis of SAM's effect.** To better understand how SAM enhances the activation of backdoor-related neurons, we provide a theoretical analysis of its effect on activation. Specifically, compared to SGD, the SAM update rule can be approximated as (Andriushchenko et al., 2023):

$$\boldsymbol{\theta}_{t+1}^{\text{SGD}} = \boldsymbol{\theta}_t - \eta \nabla \mathcal{L}(\boldsymbol{\theta}_t) \longrightarrow \boldsymbol{\theta}_{t+1}^{\text{SAM}} \approx \boldsymbol{\theta}_t - \eta \Big[ \nabla \mathcal{L}(\boldsymbol{\theta}_t) + \rho \nabla^2 \mathcal{L}(\boldsymbol{\theta}_t) \frac{\nabla \mathcal{L}(\boldsymbol{\theta}_t)}{\|\nabla \mathcal{L}(\boldsymbol{\theta}_t)\|} \Big], \tag{4}$$

where $\eta$ is the learning rate and $\rho$ is a hyperparameter to control the budget for weight perturbations.

To deepen our analysis of SAM's update mechanism, we adopt a simplified setting for theoretical tractability. Specifically, inspired by prior work (Andriushchenko et al., 2023), we consider a two-layer ReLU network, $f(\boldsymbol{x}; \boldsymbol{\theta}) = \boldsymbol{a}^\top \sigma(\boldsymbol{W}\boldsymbol{x})$, notated as $f(\boldsymbol{\theta})$ for simplicity. Our analysis focuses on the change in the pre-activation of the j-th hidden neuron, *i.e.*, $\boldsymbol{w}_j^\top \boldsymbol{x}$. While this setting simplifies complex deep architectures, it is instrumental in isolating and understanding the core mechanism by which SAM affects feature separation, providing us with key theoretical intuition. The universality of this intuition is empirically validated on large-scale models in Fig. 3.

We conduct a detailed theoretical decomposition of how the SAM update affects the pre-activation in Sec. A.1. This analysis reveals that, compared to standard SGD, SAM's update introduces a crucial additional regularization term. Based on a further analysis of this term, we can explain how it alters the learning dynamics of specific neurons when encountering poisoned samples. This finding serves as the theoretical foundation for our subsequent proposition regarding the TAC.

**Proposition 1.** *Based on the definition of TAC in Eq.* (1)*, the TAC of neuron $j$ at step $t$ is $TAC_{j,t}(\mathcal{D}) = \frac{1}{|\mathcal{D}|} \sum_{\boldsymbol{x} \in \mathcal{D}} \big| \boldsymbol{w}_{j,t}^\top \boldsymbol{x} - \boldsymbol{w}_{j,t}^\top \tilde{\boldsymbol{x}} \big|$, where $\tilde{\boldsymbol{x}} = g(\boldsymbol{x}, \Delta)$ is a poisoned input with target label 0. Suppose for all clean sample $\boldsymbol{x} \in \mathcal{D}$ and corresponding poisoned sample $\tilde{\boldsymbol{x}}$, the following conditions are satisfied: (i) $a_{j,t} \sigma'(\boldsymbol{w}_{j,t}^\top \tilde{\boldsymbol{x}}) < -\frac{\sigma(\boldsymbol{w}_{j,t}^\top \tilde{\boldsymbol{x}})}{(1 - s(f(\boldsymbol{\theta}_t)))\|\nabla f(\boldsymbol{\theta}_t)\|_2^2}$, (ii) $\sigma'(\boldsymbol{w}_{j,t}^\top \boldsymbol{x}) = 0$ and (iii) $\boldsymbol{w}_{j,t}^\top \boldsymbol{x} < \boldsymbol{w}_{j,t}^\top \tilde{\boldsymbol{x}}$. Compared with vanilla SGD, TAC change at step $t$ of neuron $j$ for SAM increases by at least $\frac{\eta \rho \, s(f(\boldsymbol{\theta}_t)) \|\tilde{\boldsymbol{x}}\|_2^2}{\|\nabla f(\boldsymbol{\theta}_t)\|_2} \big( -(1 - s(f(\boldsymbol{\theta}_t))) \|\nabla f(\boldsymbol{\theta}_t)\|_2^2 a_{j,t} \sigma'(\boldsymbol{w}_{j,t}^\top \tilde{\boldsymbol{x}}) - \sigma(\boldsymbol{w}_{j,t}^\top \tilde{\boldsymbol{x}}) \big)$.*

**Remark.** Our theory reveals how SAM enhances specific "backdoor neurons". These neurons are defined by their response to a poisoned sample $\tilde{\boldsymbol{x}}$ (target label 0): they activate on the poisoned input, remain inactive on clean ones, and possess a negative output weight. Our proof demonstrates that for neurons meeting these criteria, this occurs because SAM's optimization is driven to fit the

poisoned data point precisely, which in turn selectively amplifies their pre-activations. This selective amplification is the core mechanism that **raises the Top-$k$ TAC**, thereby increasing the feature separability between poisoned and clean data and boosting detection performance. Further analysis and proofs are detailed in Sec. 4.3 and Secs. A.2 and A.3, respectively.

## 3.4 SAM-ENHANCED PSD

Building on the above findings that (1) the backdoor effect is concentrated within a small set of neurons identified by Top-$k$ TAC, highlighting its significance for detection and suggesting that defenders should leverage the feature information of these Top-$k$ neurons, and (2) SAM can amplify this neuron-level effect, we propose **SAM-enhanced Poison Sample Detection (SAM-enhanced-PSD)**, a three-stage framework designed to improve downstream detection:

- **Stage-1: Backdoored model training.** Train a backdoored model $f_{\boldsymbol{\theta}_{\text{SAM}}}$ using SAM via Eq. (3).
- **Stage-2: Backdoor-related feature extraction.** For each input, we get the intermediate features $\boldsymbol{g} = \phi_{\boldsymbol{\theta}_{\text{SAM}}}(\boldsymbol{x})$. Because the defender lacks ground-truth TAC indices, we apply feature extraction, which is validated to enhance detection in prior work (Hayase et al., 2021), to simulate backdoor-related features. Specifically, we compute scaled features as $\boldsymbol{g}^s = \boldsymbol{\Sigma}^{-1/2}\boldsymbol{P}\boldsymbol{g}$, where $\boldsymbol{P}$ is a PCA projection matrix estimated from the training data, and $\boldsymbol{\Sigma}$ is a covariance matrix estimated from a reference clean set along with dynamically filtered candidate clean samples. A detailed comparison between this PCA-based surrogate and the true Top-$k$ neuron features is given in Sec. 4.3.
- **Stage-3: Integrating with off-the-shelf PSD.** Apply an off-the-shelf PSD method (*e.g.*, Activation Clustering) using the extracted features $\boldsymbol{g}^s$ as input.

The full algorithmic details of SAM-enhanced PSD are provided in Sec. A.4.

# 4 EXPERIMENT

## 4.1 EXPERIMENTAL SETUP

**Attack settings.** In this study, we evaluate the efficacy of backdoor attacks within an experimental framework. Specifically, we include thirteen backdoor attack methods: BadNets (Gu et al., 2019), both in its class-specific (BadNets-A2O) and universal forms (BadNets-A2A), Blended attack (Chen et al., 2017), Label-consistent attack (LC) (Turner et al., 2019), Low-frequency attack (LF) (Zeng et al., 2021), Sample-specific backdoor attack (SSBA) (Li et al., 2021b), Targeted contamination attack (TaCT) (Tang et al., 2021), Adaptive-Blend attack (Adap-Blend) (Qi et al., 2023a), Trojan attack (TrojanNN) (Liu et al., 2018), Warping-based attack (WaNet) (Nguyen & Tran, 2021), Input-aware dynamic backdoor attack(Input-aware) (Nguyen & Tran, 2020), Bit-per-pixel attack(BppAttack) (Wang et al., 2022), and dubbed sparse and invisible backdoor attack (SIBA) (Gao et al., 2024). Each attack is configured according to the default settings provided by BackdoorBench (Wu et al., 2025). The experimental evaluation is conducted on three benchmark datasets: CIFAR-10 (Krizhevsky et al., 2009), Tiny ImageNet (Russakovsky et al., 2015), and GTSRB (Stallkamp et al., 2011), and implemented on three neural network architectures, namely ResNet18 (He et al., 2016), VGG19-BN (Simonyan & Zisserman, 2014) and DenseNet-161 (Huang et al., 2017). Due to space constraints, the results for Tiny, VGG19-BN and DenseNet-161 are presented in Sec. C.1 and Sec. C.2. For our experiments, we set the poisoning ratio uniformly at 5% across all attack types, and the target label of BadNets-A2A is reassigned to $y_t = (y + 1) \mod K$ for each class y, where K represents the total number of classes. However, the LC attacks, which only poison clean samples within the target class, can only be implemented for CIFAR-10. We consider **weak backdoor attacks** as those whose poisoning ratio is low (*e.g.*, 1% or 0.5%), or those with weak trigger strength (*e.g.*, Adap-Blend).

**Detection settings.** In this study, we systematically evaluate the effectiveness of our proposed SAM-enhanced PSD, combined with a wide range of backdoor detection methods including Activation Clustering (AC) (Chen et al., 2019), Beatrix (Ma et al., 2022), SCAn (Tang et al., 2021) Spectral Signature (SS) (Tran et al., 2018), and Spectre (Hayase et al., 2021). Additionally, it is presumed that a small, clean dataset can be utilized to aid the detection process, a common practice in recent studies (Ma et al., 2022; Gao et al., 2019; Huang et al., 2023; Tang et al., 2021). For a balanced evaluation, each class in this auxiliary clean dataset contains 250 samples, which are carefully selected

Table 1: Detection comparisons (measured by TPR (%), FPR (%) and F1 (%)) between base PSD and SAM-enhanced PSD (+SAM) on CIFAR-10 and ResNet18, and the better result in each pair is highlighted in **bold**. In terms of each metric, the average change of SAM-enhanced PSD to base PSD across all attacks is presented at the bottom: performance improvements are highlighted in green, other changes in red.

| Detection → | Spectre / +SAM | | | SCAn / +SAM | | | SS / +SAM | | | AC / +SAM | | | Beatrix / +SAM | | |
| Attack ↓ | TPR↑ | FPR↓ | F1↑ | TPR↑ | FPR↓ | F1↑ | TPR↑ | FPR↓ | F1↑ | TPR↑ | FPR↓ | F1↑ | TPR↑ | FPR↓ | F1↑ |
|---|---|---|---|---|---|---|---|---|---|---|---|---|---|---|---|
| BadNets | 51.1/**88.4** | 4.9/**2.9** | 42.0/**72.6** | **96.0**/95.2 | 0.0/0.0 | **98.0**/97.6 | 70.8/**92.3** | 2.4/**1.2** | 65.6/**85.5** | **96.8**/95.4 | **0.1**/13.3 | **97.1**/42.5 | 56.6/**98.8** | 5.0/**0.5** | 44.9/**94.5** |
| Blended | 29.9/**59.7** | 6.0/**4.4** | 24.6/**49.0** | **99.2**/98.7 | 0.0/0.0 | **99.6**/99.3 | 32.9/**94.6** | 4.4/**1.1** | 30.5/**87.6** | 2.3/**98.8** | **7.7**/11.9 | 1.9/**46.4** | 5.0/**99.8** | 5.0/**1.5** | 5.0/**87.6** |
| SSBA | 36.6/**72.8** | 5.6/**3.7** | 30.1/**59.8** | 93.9/**96.5** | 0.0/0.0 | 96.9/**98.2** | 80.4/**89.9** | 1.9/**1.4** | 74.5/**83.3** | **99.3**/96.5 | **3.3**/16.2 | **76.1**/38.3 | 16.8/**98.9** | 5.0/**0.4** | 15.8/**95.5** |
| LF | 32.0/**54.1** | 5.9/**4.7** | 26.3/**44.5** | 94.1/**96.1** | 0.0/0.0 | 97.0/**98.0** | 68.2/**86.5** | 2.5/**1.5** | 63.2/**80.2** | 95.6/**96.1** | 10.4/**10.5** | 48.7/**48.7** | 2.4/**98.8** | 5.0/**0.8** | 2.4/**92.3** |
| Adap-Blend | 24.1/**65.9** | 5.6/**3.7** | 20.9/**56.0** | 92.5/**97.3** | 10.5/10.5 | 47.3/**49.1** | 20.2/**91.0** | 4.5/**1.2** | 19.6/**85.4** | 1.5/**97.1** | **7.1**/7.6 | 1.2/**57.0** | 6.2/**99.9** | **5.0**/8.2 | 6.2/**56.2** |
| LC | 17.0/**41.7** | 4.3/**3.0** | 17.1/**41.9** | **100.0**/99.9 | 0.0/0.0 | **100.0**/99.9 | 40.5/**47.8** | 2.1/**1.7** | 45.0/**53.2** | 0.0/**100.0** | 0.0/0.0 | 0.0/**100.0** | 2.2/**99.9** | 5.0/**3.2** | 2.2/**76.6** |
| TaCT | 36.1/**78.6** | 7.0/**2.3** | 26.9/**70.9** | 100.0/100.0 | 0.0/0.0 | 100.0/100.0 | 42.3/**46.4** | 4.2/**3.7** | 38.1/**42.7** | 100.0/100.0 | 0.1/**0.0** | 99.5/**100.0** | 13.4/**100.0** | 5.0/**2.6** | 12.9/**80.2** |
| TrojanNN | 30.2/**62.4** | 6.0/**4.3** | 24.8/**51.3** | 100.0/100.0 | 0.0/0.0 | 100.0/100.0 | 63.4/**97.2** | 2.8/**1.0** | 58.7/**90.1** | 99.9/**100.0** | **3.2**/12.0 | **76.7**/46.7 | 4.6/**100.0** | 5.0/**3.5** | 4.6/**75.3** |
| WaNet | 66.4/**97.7** | 4.1/**2.6** | 54.3/**79.2** | 66.3/**90.1** | 0.0/0.0 | 79.7/**94.8** | 71.1/**86.0** | 1.5/**0.7** | 71.4/**85.9** | 85.1/**90.1** | 0.0/0.0 | 91.9/**94.8** | 1.2/**95.5** | 5.0/5.0 | 1.2/**65.7** |
| Input-aware | 53.9/**99.0** | 4.7/**2.5** | 44.2/**80.2** | 97.4/**97.9** | 0.1/0.1 | 97.6/**98.4** | **83.5**/82.8 | 0.9/0.9 | **83.5**/82.8 | 0.0/**91.9** | **0.0**/0.1 | 0.0/**95.2** | 3.4/**98.4** | 5.0/5.0 | 3.4/**67.0** |
| BppAttack | 21.5/**40.2** | 6.3/**5.4** | 17.8/**33.1** | 87.8/**96.6** | 0.0/0.0 | 93.5/**98.3** | 85.8/**92.6** | 0.8/**0.4** | 85.7/**92.3** | 94.2/**96.6** | 0.0/0.0 | 97.0/**98.3** | 0.1/**99.9** | 5.0/5.0 | 0.1/**67.7** |
| SIBA | 30.0/**65.2** | 6.0/**4.1** | 24.6/**53.6** | 98.7/**98.9** | 0.0/0.0 | 99.3/**99.4** | 72.9/**91.5** | 1.2/**0.3** | 74.2/**93.1** | 96.8/**98.9** | 0.0/0.0 | 98.4/**99.5** | 4.3/**90.3** | 5.0/5.0 | 4.3/**63.3** |
| BadNets-A2A | 99.5/**99.6** | 10.6/**10.5** | 49.7/**49.8** | 0.0/0.0 | 0.0/0.0 | 0.0/0.0 | 99.4/99.4 | 10.6/10.6 | 49.7/49.7 | 97.8/96.1 | 0.0/0.0 | 98.9/98.0 | 27.3/**99.2** | 5.0/**1.0** | 24.6/**91.3** |
| Average | **+30.6** | **−1.7** | **+26.1** | **+3.2** | **-0.0** | **+1.9** | **+20.5** | **−1.1** | **+19.4** | **+29.8** | +3.0 | **+13.7** | **+87.4** | **−1.8** | **+68.1** |

from the test dataset. Furthermore, as demonstrated in the Sec. D.2.2, our method remains robust even when using limited, sifted, or out-of-distribution auxiliary clean data.

**Evaluation metrics.** We adopt three common metrics to measure the detection performance: True Positive Rate (TPR), False Positive Rate (FPR) and F1 score. Higher TPR and F1 scores indicate better performance, while lower FPR denotes better.

## 4.2 MAIN RESULTS

**Effectiveness of SAM-enhanced PSD.** To validate the effectiveness of the SAM-enhanced PSD, we demonstrate the effects of different PSDs as well as these combined with SAM-enhanced PSD on CIFAR-10 and GTSRB datasets, as shown in Tab. 1 and Tab. 2, respectively. ❶ The SAM-enhanced PSD generally enhances various base off-the-shelf PSDs, as indicated in Tab. 1

Table 2: Detection comparisons (%) between base PSD and SAM-enhanced PSD (+SAM) on GTSRB and ResNet18 (same evaluation setup as Tab. 1).

| Detection → | SCAn / +SAM | | | AC / +SAM | | | Beatrix / +SAM | | |
| Attack ↓ | TPR↑ | FPR↓ | F1↑ | TPR↑ | FPR↓ | F1↑ | TPR↑ | FPR↓ | F1↑ |
|---|---|---|---|---|---|---|---|---|---|
| BadNets | **97.8**/91.2 | 0.0/0.0 | **98.9**/95.4 | **96.4**/91.0 | 0.2/0.3 | **96.3**/92.7 | 25.8/**99.8** | 5.1/5.1 | 23.2/**67.5** |
| Blended | 88.9/**99.6** | 0.0/0.0 | 94.1/**99.8** | 0.0/**99.7** | 0.2/0.2 | 0.0/**98.4** | 46.9/**100.0** | 5.1/5.1 | 38.6/**67.6** |
| SSBA | **100.0**/97.3 | 0.0/0.0 | **100.0**/98.7 | 91.0/**97.4** | 0.1/0.3 | 94.0/**96.0** | 35.7/**100.0** | 5.1/**5.0** | 30.8/**67.6** |
| LF | **91.2**/85.8 | 0.0/0.0 | **95.4**/92.3 | 0.0/**87.9** | 0.2/1.3 | 0.0/**82.9** | 32.5/**99.5** | 5.1/5.1 | 28.4/**67.4** |
| Adap-Blend | **99.8**/97.6 | 13.5/**0.0** | 43.7/**98.8** | 88.3/**96.8** | 0.3/0.3 | 90.9/**95.3** | 97.9/**99.9** | 4.5/4.5 | 69.0/**69.9** |
| TrojanNN | 99.9/99.9 | 0.0/0.0 | 100.0/100.0 | 98.3/**99.9** | 0.4/**0.3** | 95.4/**97.2** | 36.8/**100.0** | 5.1/5.1 | 31.6/**67.6** |
| WaNet | 0.0/**71.1** | 0.0/0.0 | 0.0/**83.1** | 0.0/**72.6** | 0.1/0.7 | 0.0/**78.0** | 6.3/**86.5** | 5.1/5.1 | 6.2/**61.2** |
| Input-aware | **98.5**/98.2 | 0.1/**0.0** | 98.7/**98.8** | **95.8**/93.5 | 0.2/2.0 | **95.6**/80.9 | 30.1/**91.9** | 5.0/5.0 | 26.6/**63.8** |
| BppAttack | **99.8**/86.8 | 3.1/**0.0** | 77.2/**93.0** | **99.9**/88.0 | 0.5/0.8 | **95.5**/86.7 | 97.2/**97.8** | 5.0/5.0 | 66.3/**66.6** |
| SIBA | 99.3/**99.6** | 0.0/0.0 | 99.7/**99.8** | 91.7/**99.6** | 0.3/0.5 | 93.0/**95.0** | 13.2/**100.0** | 5.1/5.1 | 12.6/**67.6** |
| BadNets-A2A | 0.0/0.0 | 0.0/0.0 | 0.0/0.0 | 92.1/**94.2** | 0.0/0.0 | 95.9/**97.0** | 73.7/**100.0** | 5.0/5.1 | 54.7/**67.6** |
| Average | **+4.7** | **−1.5** | **+13.8** | **+24.3** | +0.4 | **+22.1** | **+52.7** | **-0.0** | **+31.5** |

and Tab. 2. For CIFAR-10, we improved the True Positive Rate (TPR) by over 25% for four detection methods. Averaged over 13 attack types and 5 detection methods on the CIFAR-10 dataset, our approach yields a 34.3% improvement in TPR. On GTSRB, we enhanced the detection performance of most methods, with increases in TPR exceeding 20% for two methods. ❷ For methods like Spectre, SS and Beatrix, which are based on anomaly detection, SAM-enhanced PSD increases the prominence of poisoned samples at backdoor neurons relative to clean samples, making these samples more anomalous and thus enhancing detection. In GTSRB, the number of poisoned and target clean samples is similar, which breaks the key assumption of anomaly-based methods like Spectre and SS. In addition, since neither method leverages a surrogate clean dataset, they lack the reference needed to discriminate poisoned inputs. We exclude them from our evaluation. ❸ For the SCAn method, our approach shows significant improvements, especially under attacks where SCAn typically underperforms, such as WaNet. Since SCAn requires identification of the target label, it cannot detect BadNets-A2A. ❹ For the AC, since SAM-enhanced PSD increases the activation of poisoned samples in backdoor-related neurons, it causes poisoned samples to deviate more from clean samples and cluster more tightly, thereby enhancing detection effectiveness. We also design an adaptive attack targeting our method and, as detailed in Sec. C.5, demonstrate that an attacker who cannot alter the training process cannot weaken our detector. Furthermore, retraining on the purified dataset confirms our defense's effectiveness in Sec. C.6.

**Performance under different poisoning ratios.** To evaluate the impact of the poisoning ratio on the SAM-enhanced PSD, especially in the case of what we consider to be **weak backdoor attacks**, we present the average detection performance of four detection methods under all attacks on CIFAR-10

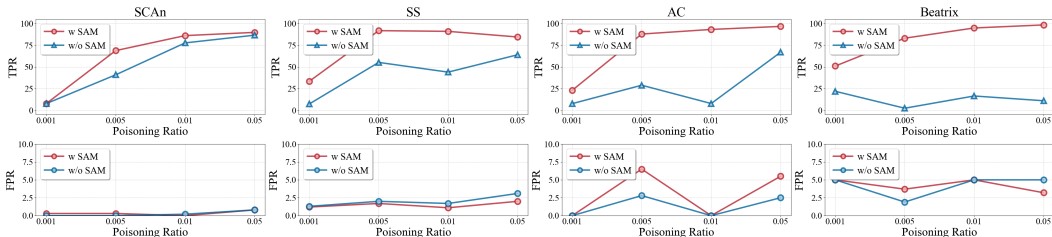

Figure 4: Detection performance of base PSD (w/o SAM) and SAM-enhanced PSD (w SAM) under different poisoning ratios on CIFAR10 and ResNet18.

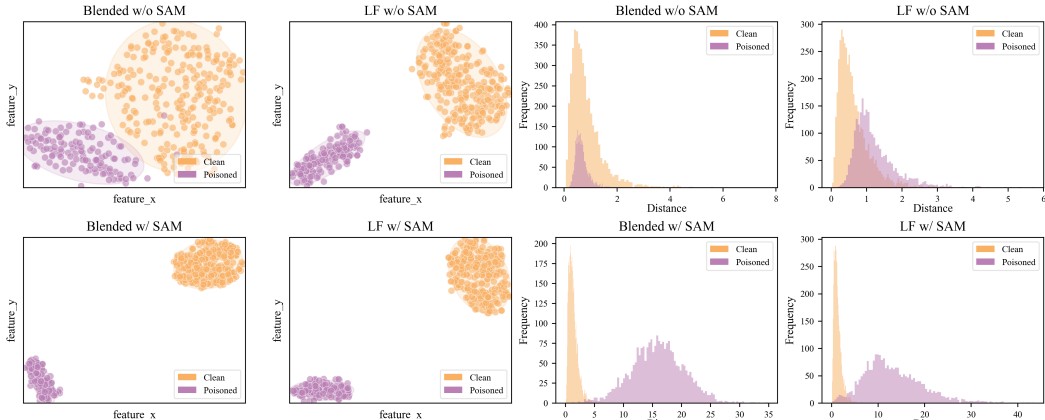

Figure 5: **Left:** t-SNE under different attacks in models trained with Vanilla training (Top row) and with SAM (Bottom row). **Right:** Distribution of distances between the target clean samples center and each sample in models trained with Vanilla training (Top row) and with SAM (Bottom row) under various backdoor attacks on CIFAR-10 and ResNet18.

and ResNet18, as shown in Fig. 4. The selected range for the poisoning ratio is {0.1%, 0.5%, 1%, 5%}. ❶ SAM-enhanced PSD enhances detection performance across different poisoning ratios as illustrated in Fig. 4. Notably, when the poisoning ratio is low such as $0.5\%$ and $1\%$, SAM-enhanced PSD significantly improves the performance of PSDs. ❷ When the poisoning ratio is 0.1%, even though SAM-enhanced PSD improves performance, its average True Positive Rate (TPR) does not exceed 60%. The average attack success rate is only 31.8%, meaning a complete backdoor cannot form, which makes poisoned samples harder to detect. Detailed results are provided in Sec. C.3.

### 4.3 ANALYSIS OF SAM-ENHANCED-PSD

**Ablation study in SAM-enhanced PSD.** As shown in Tab. 3, we evaluate the impact of sharpness-aware minimization (SAM) and backdoor-related feature extraction (BRF) on the detection of backdoor attacks.. Integrating SAM significantly enhances detection effectiveness by increasing the feature gap between clean and poisoned samples. Incorporating BRF pushes performance further by concentrating on the dimensions that are most informative for recognizing poisoned samples, thereby making the backdoor pattern more

Table 3: Ablation study with different components of SAM-enhanced PSD under various backdoor attacks and base PSDs on CIFAR-10 and ResNet18.

| Detection | SAM | BRF | SS | | Beatrix | |
|---|---|---|---|---|---|---|
| | | | TPR(%) | FPR(%) | TPR(%) | FPR(%) |
| BadNets | ✗ | ✗ | 70.8 | 2.4 | 56.6 | 5.0 |
| | ✓ | ✗ | 86.0 | **0.6** | 98.4 | 5.0 |
| | ✗ | ✓ | 72.8 | 1.2 | 67.0 | 5.0 |
| | ✓ | ✓ | **92.3** | 1.2 | **98.8** | **0.5** |
| Blended | ✗ | ✗ | 32.9 | 4.4 | 5.0 | 5.0 |
| | ✓ | ✗ | 90.6 | **0.3** | 79.8 | 5.0 |
| | ✗ | ✓ | 60.4 | 1.9 | 27.1 | 5.0 |
| | ✓ | ✓ | **94.6** | 1.1 | **99.8** | 1.5 |

salient. Combining SAM and BRF provides both stronger feature separability and a focus on backdoor-relevant directions, yielding the best results against both attack types.

**Effect of SAM on feature-space separation between poisoned and clean samples.** To gain a deeper understanding of the impact of the SAM on poisoned and clean samples, we conduct a detailed analysis and visualization of the distribution of samples in the feature space in Fig. 5. ❶ t-SNE visualization: As shown on the **left** of Fig. 5, we demonstrate, through t-SNE, the distribution of poisoned and target clean samples in the feature space, where we observe that after training with

Figure 6: Pearson correlation between PCA component loadings and TAC scores under four different backdoor attacks on CIFAR10 dataset with ResNet18 model. Each point represents the Pearson correlation between the neuron loadings of the $i$-th PCA component and the neuron-wise TAC scores.

the SAM, poisoned samples are more distinctly separated from the target class. ❷ Distribution of distances between the target clean sample center and each sample: The center of the target clean samples is defined by the average of their features within the current model. As illustrated on the **right** of Fig. 5, in models trained with vanilla training (SGD), the features of poisoned samples are closer to the center of clean samples compared to those in models trained with SAM, which increases the difficulty in distinguishing between poisoned and clean samples. Sec. D.7 provides additional t-SNE visualizations of various attacks. These results show that, after training with the SAM, poisoned samples are much more clearly separated from the target class in feature space.

**Relationship between Top-$k$ TAC neurons and principal components.** We analyze the alignment between the Top-$k$ TAC neurons and PCA components to investigate how effectively BRF captures backdoor-relevant information. Specifically, we calculate the Pearson correlation coefficients $r_i = \mathrm{corr}\left(\mathbf{l}^{(i)}, \mathbf{t}\right)$ between each principal component's loading vector $\mathbf{l}^{(i)}$ and the corresponding TAC score vector $\mathbf{t}$. By plotting these correlations for various attacks (see Fig. 6), we consistently observe that the strongest correlations occur among the initial principal components and gradually diminish as we move toward components with lower variance contributions. This indicates that neurons with higher TAC values have larger loadings in the leading principal components, thus confirming that BRF effectively captures the backdoor-related signals embedded in the Top-$k$ neuron subset.

**Additional analyses in appendix.** To further validate the robustness of our method, we provide several complementary analyses in Appendix Secs. C and D. ❶ **Adaptive attack.** We evaluate our framework under a stronger adversary with full knowledge of the SAM-based training procedure, who explicitly optimizes the trigger to minimize the feature-space gap. Even in this challenging setting, our method consistently achieves over 94% TPR across multiple detectors, whereas baseline performance drops to near zero. This result indicates that the proposed defense is difficult to circumvent unless the attacker can directly control the entire training process (details in Sec. C.5). ❷ **Computational efficiency.** Standard SAM approximately doubles training time. We demonstrate that our framework is fully compatible with efficient variants such as MSAM, which deliver comparable detection gains while incurring almost no additional runtime relative to the baseline. This ensures the method remains both effective and practical for real-world deployment (details in Sec. D.4). ❸ **Stability and data flexibility.** We examine performance sensitivity to hyperparameters and SAM variant choices, finding the results to be stable across settings. Importantly, the method maintains strong detection performance even with limited clean samples, "sifted-clean" data extracted from the poisoned set, or entirely out-of-distribution auxiliary data. These findings highlight the flexibility and practical utility of our approach in diverse application scenarios (details in Secs. D.1 to D.3).

## 5 CONCLUSION

This work revisits existing poisoned sample detection (PSD) methods and finds that they often struggle against weak backdoor attacks, such as those with low poisoning ratios or weak trigger strengths. Our statistical analysis reveals a positive correlation between the strength of the backdoor effect and detection performance. Based on this finding, we propose to amplify the backdoor effect by training the model using Sharpness-Aware Minimization (SAM), without changing the poisoning ratio or trigger strength, thereby making poisoned samples more detectable. Our method, called *SAM-enhanced PSD*, integrates easily with any feature-based PSD method. Experiments on diverse datasets and network architectures show that our method significantly improves detection performance. This work contributes to defending against backdoor attacks in deep neural networks, providing a new perspective that complements existing detection methods and has the potential to inspire further research in this critical area.

## 6 ETHICS STATEMENT

Our research is ethically motivated by the need to enhance the security of AI systems in critical applications, as vulnerabilities like backdoor attacks pose significant risks to public safety and trust. We acknowledge the dual-use nature of security research; however, our work is fundamentally defensive in its aim to create more robust detection methods. By publicly sharing these findings, we intend to empower the security community, believing the positive impact of creating more trustworthy AI outweighs the potential for misuse.

## 7 REPRODUCIBILITY STATEMENT

To ensure our results are reproducible, all experiments were conducted using public benchmarks such as the CIFAR-10 and GTSRB datasets with standard models like ResNet-18, all within the BackdoorBench framework. We have documented the specific configurations for all thirteen attack types and five detection methods evaluated. Key hyperparameters for both conventional SGD and SAM-based training, along with detailed pseudocode for our method, are provided in the paper and its appendix to allow for the complete replication of our findings.

## ACKNOWLEDGMENTS

Baoyuan Wu is supported by Guangdong Basic and Applied Basic Research Foundation (No. 2024B1515020095), Guangdong Provincial Program (No. 2023TQ07A352), Shenzhen Science and Technology Program (No. RCYX20210609103057050 and JCYJ20240813113608011), and Longgang District Key Laboratory of Intelligent Digital Economy Security.

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

# Reliable Poisoned Sample Detection against Backdoor Attacks Enhanced by Sharpness Aware Minimization

## Appendix

## A DETAILS OF SAM-ENHANCED PSD

### A.1 THEORETICAL ANALYSIS OF THE SAM UPDATE MECHANISM

For theoretical tractability, we consider a two-layer ReLU network defined as $f(\boldsymbol{x}; \boldsymbol{\theta}) = \boldsymbol{a}^\top \sigma(\boldsymbol{W}\boldsymbol{x})$, simplified as $f(\boldsymbol{\theta})$, with a logistic loss $\ell(\boldsymbol{\theta})$ to adapt to our task. Here, $\boldsymbol{W} = [\boldsymbol{w}_1, \ldots, \boldsymbol{w}_m]^\top$ represents the weight matrix of the first layer. We analyze the pre-activation value of each hidden neuron $j$, defined as $\boldsymbol{w}_j^\top \boldsymbol{x}$.

According to the SAM update rule, SAM introduces an extra second-order perturbation compared to SGD. This update acts on the step-$t$ weight $\boldsymbol{w}_{j,t}$ of neuron $j$ and, in turn, changes its pre-activation $\boldsymbol{w}_{j,t}^\top \boldsymbol{x}$. As stated in the lemma below, the one-step change splits into three parts: one *SGD part* plus two *SAM parts*.

**Lemma 1.** *At training step $t$, the change in the pre-activation of neuron $j$ approximately splits into:*

$$\boldsymbol{w}_{j,t+1}^\top \boldsymbol{x} - \boldsymbol{w}_{j,t}^\top \boldsymbol{x} \simeq \eta\big(① + ② + ③\big),$$

*where*

① $-a_{j,t}\,\sigma'(\boldsymbol{w}_{j,t}^\top \boldsymbol{x})\|\boldsymbol{x}\|_2^2\,\big(s(f(\boldsymbol{\theta}_t)) - y\big)$ *is the standard **SGD update**,*

② $-\rho\,\mathrm{sign}\big(s(f(\boldsymbol{\theta}_t)) - y\big)\,s(f(\boldsymbol{\theta}_t))(1 - s(f(\boldsymbol{\theta}_t)))\,\|\nabla f(\boldsymbol{\theta}_t)\|_2\,a_{j,t}\,\sigma'(\boldsymbol{w}_{j,t}^\top \boldsymbol{x})\,\|\boldsymbol{x}\|_2^2$ *is the **SAM data-fitting term**, and*

③ $-\rho\,\big|s(f(\boldsymbol{\theta}_t)) - y\big|\,\dfrac{\sigma(\boldsymbol{w}_{j,t}^\top \boldsymbol{x})\,\|\boldsymbol{x}\|_2^2}{\|\nabla f(\boldsymbol{\theta}_t)\|_2}$ *acts as a **SAM regularization component**.*

*where $s(z) = 1/(1 + \exp(-z))$ is the sigmoid function.*

The decomposition above provides the foundation for the proposition mentioned in the main text. Without loss of generality, let's assume the target label for a poisoned sample $\tilde{\boldsymbol{x}}$ is $y = 0$. For any neuron $j$ that is activated by this input (i.e., $\sigma(\boldsymbol{w}_{j,t}^\top \boldsymbol{x}) > 0$) and has a corresponding second-layer weight $a_{j,t} < 0$, the sum of the data-fitting term and the regularization term may be positive (② + ③ > 0).

This causes the SAM update to push the neuron's pre-activation more strongly than the SGD update would, potentially leading to a positive change in TAC. This observation clarifies the core idea of our proposition that SAM influences the model's feature representation by amplifying the activations of specific neurons.

### A.2 PROOF OF THE LEMMA 1

*Proof.* Considering the sample is $(\boldsymbol{x}, y)$. We have for the update of SAM:

$$
\begin{aligned}
&\nabla \ell\left(\boldsymbol{\theta} + \rho \frac{\nabla \ell(\boldsymbol{\theta}_t)}{\|\nabla \ell(\boldsymbol{\theta}_t)\|_2}\right) \\
=&\nabla \ell(\boldsymbol{\theta}_t) + \rho \nabla^2 \ell(\boldsymbol{\theta}_t) \frac{\nabla \ell(\boldsymbol{\theta}_t)}{\|\nabla \ell(\boldsymbol{\theta}_t)\|_2} + \mathcal{O}\left(\rho^2\right) \\
=&\nabla \left[\ell(\boldsymbol{\theta}_t) + \rho\|\nabla \ell(\boldsymbol{\theta}_t)\|_2 + \mathcal{O}\left(\rho^2\right)\right].
\end{aligned}
\tag{5}
$$

Thus, under the first-order approximation, a step of SAM corresponds to a gradient update on the regularized objective $\ell(\boldsymbol{\theta}_t) + \rho\|\nabla \ell(\boldsymbol{\theta}_t)\|_2$. Now recall that the layerwise gradients of a two-layer ReLU network can be written as:

$$
\begin{aligned}
\nabla_{\boldsymbol{a}_t} \ell(\boldsymbol{\theta}_t) =&(s(f(\boldsymbol{\theta}_t)) - y) \cdot \sigma(\boldsymbol{W}_t \boldsymbol{x}), \\
\end{aligned}
\tag{6}
$$

$$
\begin{aligned}
\nabla_{\boldsymbol{w}_{j,t}} \ell(\boldsymbol{\theta}_t) =&(s(f(\boldsymbol{\theta}_t)) - y) \cdot a_{j,t}\sigma'\left(\boldsymbol{w}_{j,t}^\top \boldsymbol{x}\right) \boldsymbol{x}, \\
\end{aligned}
\tag{7}
$$

where $s(f(\boldsymbol{\theta}_t)) = \frac{1}{1+exp(-f(\boldsymbol{\theta}_t))}$.

Then a direct computation gives us the following expression for the full gradient norm:

$$
\begin{aligned}
&\|\nabla\ell(\boldsymbol{\theta}_t)\|_2 \\
=&|s(f(\boldsymbol{\theta}_t))-y|\cdot\|\nabla f(\boldsymbol{\theta}_t)\|_2 \\
=&|s(f(\boldsymbol{\theta}_t))-y|\sqrt{\|\sigma(\boldsymbol{W}_t\boldsymbol{x})\|_2^2+\|\boldsymbol{x}\|_2^2\cdot\|\boldsymbol{a}_t\odot\sigma'(\boldsymbol{W}_t\boldsymbol{x})\|_2^2},
\end{aligned}
\tag{8}
$$

where $\odot$ denotes element-wise multiplication. Then the update of $\boldsymbol{w}_{j,t}$ for neuron $j$ on each step of SAM with step size $\eta$ can be written as:

$$
\begin{aligned}
\boldsymbol{w}_{j,t+1}=&\boldsymbol{w}_{j,t}-\eta\left(\nabla\ell(\boldsymbol{\theta}_t)+\rho\nabla\|\nabla\ell(\boldsymbol{\theta}_t)\|_2\right)+\mathcal{O}\left(\rho^2\right), \\
\boldsymbol{w}_{j,t+1}=&\boldsymbol{w}_{j,t}-\eta(s(f(\boldsymbol{\theta}_t))-y)a_{j,t}\sigma'\left(\boldsymbol{w}_{j,t}^\top\boldsymbol{x}\right)\boldsymbol{x} \\
&-\eta\rho\,\mathrm{sign}(s(f(\boldsymbol{\theta}_t))-y)s(f(\boldsymbol{\theta}_t))(1-s(f(\boldsymbol{\theta}_t)))\|\nabla f(\boldsymbol{\theta}_t)\|_2 a_{j,t}\sigma'\left(\boldsymbol{w}_{j,t}^\top\boldsymbol{x}\right)\boldsymbol{x} \\
&-\eta\rho|s(f(\boldsymbol{\theta}_t))-y|/\|\nabla f(\boldsymbol{\theta}_t)\|_2\sigma\left(\boldsymbol{w}_{j,t}^\top\boldsymbol{x}\right)\boldsymbol{x}+\mathcal{O}\left(\rho^2\right),
\end{aligned}
$$

where we used the fact that $\sigma'\left(\boldsymbol{w}_{j,t}^\top\boldsymbol{x}\right)\sigma\left(\boldsymbol{w}_{j,t}^\top\boldsymbol{x}\right)=\sigma\left(\boldsymbol{w}_{j,t}^\top\boldsymbol{x}\right)$ and second-order terms are zero almost everywhere for ReLUs. The $\rho$ refers to the radius of perturbation. If $\rho$ is sufficiently small, $\mathcal{O}(\rho^2)$ is negligible relative to other terms in Eq. (3) of the manuscript. The assumption of sufficiently small $\rho$ is commonly used to analyze the property of SAM in the community(Wen et al., 2022). The activation of Neuron can be approximately updated by:

$$
\begin{aligned}
\boldsymbol{w}_{j,t+1}^\top\boldsymbol{x}\simeq&\boldsymbol{w}_{j,t}^\top\boldsymbol{x}-\eta a_{j,t}\,\sigma'(\boldsymbol{w}_{j,t}^\top\boldsymbol{x})\|\boldsymbol{x}\|_2^2\left(s(f(\boldsymbol{\theta}_t))-y\right) \\
&-\eta\rho\,\mathrm{sign}\big(s(f(\boldsymbol{\theta}_t))-y\big)\,s(f(\boldsymbol{\theta}_t))(1-s(f(\boldsymbol{\theta}_t)))\,\|\nabla f(\boldsymbol{\theta}_t)\|_2\,a_{j,t}\,\sigma'(\boldsymbol{w}_{j,t}^\top\boldsymbol{x})\,\|\boldsymbol{x}\|_2^2 \\
&-\eta\rho\left|s(f(\boldsymbol{\theta}_t))-y\right|\frac{\sigma(\boldsymbol{w}_{j,t}^\top\boldsymbol{x})\,\|\boldsymbol{x}\|_2^2}{\|\nabla f(\boldsymbol{\theta}_t)\|_2} \\
=&\eta(①+②+③).
\end{aligned}
$$

$\square$

## A.3 PROOF OF THE PROPOSITION 1

*Proof.* We consider the clean sample and the corresponding poisoned sample is $(\boldsymbol{x},y)$ and $(\tilde{\boldsymbol{x}}=g(\boldsymbol{x},\Delta),y_t)$, where $\Delta$ is a predefined trigger, $g$ is a generation function and a target label $y_t$. Without loss of generality, we assume that the label of poisoned sample is 0. We have for the update of TAC:

$$
TAC_{j,t+1}(\mathcal{D})-TAC_{j,t}(\mathcal{D})=\frac{1}{|\mathcal{D}|}\sum_{\boldsymbol{x}\in\mathcal{D}}\left|\boldsymbol{w}_{j,t+1}^\top\boldsymbol{x}-\boldsymbol{w}_{j,t+1}^\top\tilde{\boldsymbol{x}}\right|-\frac{1}{|\mathcal{D}|}\sum_{\boldsymbol{x}\in\mathcal{D}}\left|\boldsymbol{w}_{j,t}^\top\boldsymbol{x}-\boldsymbol{w}_{j,t}^\top\tilde{\boldsymbol{x}}\right|.
\tag{9}
$$

Under the condition (iii), we have

$$\left|\boldsymbol{w}_{j,t+1}^{\top}\boldsymbol{x} - \boldsymbol{w}_{j,t+1}^{\top}\tilde{\boldsymbol{x}}\right| - \left|\boldsymbol{w}_{j,t}^{\top}\boldsymbol{x} - \boldsymbol{w}_{j,t}^{\top}\tilde{\boldsymbol{x}}\right|$$

$$=(\boldsymbol{w}_{j,t+1}^{\top}\tilde{\boldsymbol{x}} - \boldsymbol{w}_{j,t}^{\top}\tilde{\boldsymbol{x}}) - (\boldsymbol{w}_{j,t+1}^{\top}\boldsymbol{x} - \boldsymbol{w}_{j,t}^{\top}\boldsymbol{x})$$

$$= -\eta a_{j,t}\,\sigma'(\boldsymbol{w}_{j,t}^{\top}\tilde{\boldsymbol{x}})\|\tilde{\boldsymbol{x}}\|_2^2\left(s(f(\boldsymbol{\theta}_t)) - y\right)$$

$$\quad - \eta\rho\,\mathrm{sign}\big(s(f(\boldsymbol{\theta}_t)) - y\big)\,s(f(\boldsymbol{\theta}_t))(1 - s(f(\boldsymbol{\theta}_t)))\,\|\nabla f(\boldsymbol{\theta}_t)\|_2\,a_{j,t}\,\sigma'(\boldsymbol{w}_{j,t}^{\top}\tilde{\boldsymbol{x}})\,\|\tilde{\boldsymbol{x}}\|_2^2$$

$$\quad - \eta\rho\left|s(f(\boldsymbol{\theta}_t)) - y\right|\frac{\sigma(\boldsymbol{w}_{j,t}^{\top}\tilde{\boldsymbol{x}})\,\|\tilde{\boldsymbol{x}}\|_2^2}{\|\nabla f(\boldsymbol{\theta}_t)\|_2}$$

$$\quad + \eta a_{j,t}\,\sigma'(\boldsymbol{w}_{j,t}^{\top}\boldsymbol{x})\|\boldsymbol{x}\|_2^2\left(s(f(\boldsymbol{\theta}_t)) - y\right)$$

$$\quad + \eta\rho\,\mathrm{sign}\big(s(f(\boldsymbol{\theta}_t)) - y\big)\,s(f(\boldsymbol{\theta}_t))(1 - s(f(\boldsymbol{\theta}_t)))\,\|\nabla f(\boldsymbol{\theta}_t)\|_2\,a_{j,t}\,\sigma'(\boldsymbol{w}_{j,t}^{\top}\boldsymbol{x})\,\|\boldsymbol{x}\|_2^2$$

$$\quad + \eta\rho\left|s(f(\boldsymbol{\theta}_t)) - y\right|\frac{\sigma(\boldsymbol{w}_{j,t}^{\top}\boldsymbol{x})\,\|\boldsymbol{x}\|_2^2}{\|\nabla f(\boldsymbol{\theta}_t)\|_2}$$

$$= -\eta a_{j,t}\,\sigma'(\boldsymbol{w}_{j,t}^{\top}\tilde{\boldsymbol{x}})\|\tilde{\boldsymbol{x}}\|_2^2\,s(f(\boldsymbol{\theta}_t))$$

$$\quad - \eta\rho\,s(f(\boldsymbol{\theta}_t))(1 - s(f(\boldsymbol{\theta}_t)))\,\|\nabla f(\boldsymbol{\theta}_t)\|_2\,a_{j,t}\,\sigma'(\boldsymbol{w}_{j,t}^{\top}\tilde{\boldsymbol{x}})\,\|\tilde{\boldsymbol{x}}\|_2^2$$

$$\quad - \eta\rho\,s(f(\boldsymbol{\theta}_t))\frac{\sigma(\boldsymbol{w}_{j,t}^{\top}\tilde{\boldsymbol{x}})\,\|\tilde{\boldsymbol{x}}\|_2^2}{\|\nabla f(\boldsymbol{\theta}_t)\|_2}$$

$$\quad + \eta\rho\,(1 - s(f(\boldsymbol{\theta}_t)))\frac{\sigma(\boldsymbol{w}_{j,t}^{\top}\boldsymbol{x})\,\|\boldsymbol{x}\|_2^2}{\|\nabla f(\boldsymbol{\theta}_t)\|_2}$$

$$\geq -\eta a_{j,t}\,\sigma'(\boldsymbol{w}_{j,t}^{\top}\tilde{\boldsymbol{x}})\|\tilde{\boldsymbol{x}}\|_2^2\,s(f(\boldsymbol{\theta}_t))$$

$$\quad \frac{\eta\rho\,s(f(\boldsymbol{\theta}_t))\|\tilde{\boldsymbol{x}}\|_2^2}{\|\nabla f(\boldsymbol{\theta}_t)\|_2}\left(-(1 - s(f(\boldsymbol{\theta}_t)))\,\|\nabla f(\boldsymbol{\theta}_t)\|_2^2\,a_{j,t}\,\sigma'(\boldsymbol{w}_{j,t}^{\top}\tilde{\boldsymbol{x}}) - \sigma(\boldsymbol{w}_{j,t}^{\top}\tilde{\boldsymbol{x}})\right)$$

where the third equality uses condition (ii) together with the fact that the poisoned sample $\tilde{\boldsymbol{x}}$ has label $y = 0$. Hence

$$TAC_{j,t+1}(\mathcal{D}) - TAC_{j,t}(\mathcal{D})$$

$$\geq \frac{1}{|\mathcal{D}|}\sum_{\boldsymbol{x}\in\mathcal{D}}\underbrace{-\eta a_{j,t}\,\sigma'(\boldsymbol{w}_{j,t}^{\top}\tilde{\boldsymbol{x}})\|\tilde{\boldsymbol{x}}\|_2^2\,s(f(\boldsymbol{\theta}_t))}_{\text{SGD update term}}$$

$$\quad + \underbrace{\frac{\eta\rho\,s(f(\boldsymbol{\theta}_t))\|\tilde{\boldsymbol{x}}\|_2^2}{\|\nabla f(\boldsymbol{\theta}_t)\|_2}\left(-(1 - s(f(\boldsymbol{\theta}_t)))\,\|\nabla f(\boldsymbol{\theta}_t)\|_2^2\,a_{j,t}\,\sigma'(\boldsymbol{w}_{j,t}^{\top}\tilde{\boldsymbol{x}}) - \sigma(\boldsymbol{w}_{j,t}^{\top}\tilde{\boldsymbol{x}})\right)}_{\text{positive under condition (i)}}$$

where the first term is the standard SGD update and the second term is strictly positive under condition (i), ensuring an overall increase in the neuron's TAC.

$\square$

## A.4 Detailed algorithm of SAM-enhanced PSD

We present a detailed algorithmic procedure for SAM-enhanced PSD in Algorithm 1. When training with Sharpness-Aware Minimization (SAM), we select dataset-specific sharpness parameter values, setting $\rho = 0.1$ for CIFAR-10 and $\rho = 0.4$ for both Tiny ImageNet and GTSRB. Additionally, we utilize a reference dataset containing 250 clean samples per class. Dimensionality reduction parameters are set specifically at 30 for CIFAR-10, 20 for GTSRB, and 10 for Tiny ImageNet, with each configuration undergoing 10 iterations. The detection thresholds for these datasets are established as 10, 25, and 5, respectively.

---

**Algorithm 1** Full Algorithm of SAM-enhanced PSD

---

1: **Input:** Dataset $\mathcal{D}_{tr} = \{(\boldsymbol{x}_i, y_i)\}_{i=1}^N$ to be cleansed, reference clean set $\mathcal{D}_{ref}$, loss function $\ell$, model $f_{\boldsymbol{\theta}}$, epochs $E$, learning rate $\eta > 0$, perturbation bound $\rho > 0$, class $K$, reduced dimension $d$, iteration $iter$, threshold $\epsilon$, detection algorithm $\mathcal{A}$.

2: **Output:** The cleansed dataset $D^*$

3: Initialize $\boldsymbol{\theta}_0$.

4: **for** $t = 1, \ldots, E$ **do**

5:     Sample a mini-batch $B$ from $\mathcal{D}_{tr}$;

6:     Update $\epsilon_{t+1}$ via $\rho \frac{\nabla_{\boldsymbol{\theta}} \mathcal{L}(\boldsymbol{\theta}_t)}{\|\nabla_{\boldsymbol{\theta}} \mathcal{L}(\boldsymbol{\theta}_t)\|_2}$ w.r.t. $B$;

7:     Update weights: $\boldsymbol{\theta}_{t+1} = \boldsymbol{\theta}_t - \eta \nabla_{\boldsymbol{\theta}} \mathcal{L}(\boldsymbol{\theta}_t + \epsilon_{t+1})$ w.r.t. $B$;

8: **end for**

9: **for** $k = 1, \ldots, K$ **do**

10:     Get the representation sets of class c for training dataset $\mathcal{G}_{tr,k} = \{\boldsymbol{g} = f_{\boldsymbol{\theta}}(\boldsymbol{x}) \mid (x, y) \in \mathcal{D}_{tr}, y = c\}$ and reference dataset $\mathcal{G}_{ref,k} = \{\boldsymbol{g} = f_{\boldsymbol{\theta}}(\boldsymbol{x}) \mid (x, y) \in \mathcal{D}_{ref}, y = c\}$

11:     Perform PCA on normalized $\mathcal{G}_{tr,k}$ to extract the top-$d$ features and obtain the projection matrix $P \in \mathbb{R}^{p \times d}$, where $p$ is the original feature dimension.

12:     Obtain projected features $\tilde{\mathcal{G}}_{tr,k}$ and $\tilde{\mathcal{G}}_{ref,k}$.

13:     **for** $i = 1, \ldots, iter$ **do**

14:         Estimate the mean $\boldsymbol{\mu}_k$ and covariance matrix $\boldsymbol{\Sigma}_k$ based on the features of the reference dataset $\tilde{\mathcal{G}}_{ref,k}$.

15:         Compute the Mahalanobis distance from $\tilde{\mathcal{G}}_{tr,k}$ to the mean $\boldsymbol{\mu}_k$.

16:         Add samples to the reference dataset features $\tilde{\mathcal{G}}_{ref,k}$ if Mahalanobis distance is less than $\frac{i*\epsilon}{iter}$.

17:     **end for**

18:     Project the features $\tilde{\mathcal{G}}_{tr,k}$ for class $k$ using the final covariance matrix $\boldsymbol{\Sigma}_k$ to get $\tilde{\mathcal{G}}_{tr,k}^* = \{\boldsymbol{g}^* = \boldsymbol{\Sigma}_k^{-1/2} \boldsymbol{g} \mid \boldsymbol{g} \in \tilde{\mathcal{G}}_{tr,k}\}$.

19: **end for**

20: Input the projected features $\tilde{\mathcal{G}}_{tr}^*$ into the detection algorithm $\mathcal{A}$ to obtain the filtered dataset $D^* = \mathcal{A}(\tilde{\mathcal{G}}_{tr}^*)$.

21: **return** $D^*$

---

# B DETAILS OF EXPERIMENT SETTINGS

## B.1 DETAILED TRAINING SETTINGS

In our experiments, all vanilla training sessions utilize Stochastic Gradient Descent (SGD) with a learning rate $lr$ of 0.1, momentum of 0.9, and a weight decay of $1 \times 10^{-4}$. However, for training on the VGG19-BN model, we adjust the learning rate to 0.01 to optimize performance. All training sessions across these experiments are standardized to 100 epochs, ensuring consistency in our approach to model development and evaluation across different datasets.

## B.2 DETAILED ATTACK SETTINGS

The methodologies in our experiment, including BadNets (Gu et al., 2019), Blended attack (Chen et al., 2017), Label-consistent attack (LC) (Turner et al., 2019), Low-frequency attack (LF) (Zeng et al., 2021), Sample-specific backdoor attack (SSBA) (Li et al., 2021b), Trojan attack (TrojanNN) (Liu et al., 2018), Warping-based attack (WaNet) (Nguyen & Tran, 2021), Input-aware dynamic backdoor attack(Input-aware) (Nguyen & Tran, 2020), Bit-per-pixel attack(BppAttack) (Wang et al., 2022), and dubbed sparse and invisible backdoor attack (SIBA) (Gao et al., 2024), are configured following the standard settings provided by BackdoorBench (Wu et al., 2025), a recognized benchmark for backdoor attacks. The target label for GTSRB is 2 and the target label for CIFAR-10 is 0. The TaCT (Tang et al., 2021) method targets a specific class (1,3) by inserting a trigger into the samples and changing their labels to a target label. Additionally, it uses another class (5,7), where the same trigger is applied but without changing the labels. Only about 0.05% of the samples receive the trigger without changing the labels. The Adap-Blend (Qi et al., 2023a) method uses an adaptive

approach where a masking parameter $m = 0.5$ controls the probability of concealing parts of the images with a trigger.

### B.3 DETAILED DETECTION SETTINGS

The methodologies in our experiment, including Activation Clustering (AC) (Chen et al., 2019), Beatrix (Ma et al., 2022), SCAn (Tang et al., 2021) Spectral Signature (SS) (Tran et al., 2018) and Spectre (Hayase et al., 2021), are configured following the standard settings provided by Backdoor-Bench (Wu et al., 2025). For the Spectral Signature (SS) and Spectre methodologies, we adjust the proportion of potentially harmful samples to $1.5 * p$, where $p$ is the poisoning ratio and we adopt the strategy outlined in previous studies (referenced as Yuan et al. (2025)) to determine the target label for these potentially poisoned samples, which are applied in both the base detection and SAM-enhanced PSD. The Activation Clustering (AC) method typically requires that the volume of potential poisoned samples be less than 35% of the number of samples in the class. However, given the large number of classes in the GTSRB dataset and the proximity of the poisoned sample volume to 50% of the number of samples in the class, we disregard the usual 35% threshold in this specific case.

## C ADDITIONAL EXPERIMENT RESULTS

### C.1 MAIN EXPERIMENTS ON TINY IMAGENET

To validate the detection performance of SAM-enhanced PSD on large datasets, we conduct a series of tests using Tiny ImageNet. Due to the limited number of samples per category in the Tiny ImageNet, it poses a challenge for various attack and detection methods. In our experiments, we limited the poisoning rate to 0.5% and only tested attack types that could meet the prerequisites for the number of poisoned samples under this condition. Detection methods like AC, SS, and Spectre are also excluded from our tests because these methods require fewer poisoned samples than there are samples in the target class, a condition difficult to meet with the Tiny ImageNet.

As shown in Tab. 4, our experimental results indicate that SAM-enhanced PSD performs exceptionally well in enhancing detection. We found that despite the limited number of clean samples per category, this method effectively improved the recognition of poisoned samples.

Table 4: Comparison of TPR (%) and FPR (%) between base PSD and SAM-enhanced PSD (+SAM) on Tiny ImageNet and ResNet18 with poisoning ratio = 0.5%. Top 1 are **bold**. When comparing SAM-enhanced PSD to base PSD, performance improvements are highlighted in green, other changes in red.

| Detection → Attack ↓ | SCAn / +SAM | | | Beatrix / +SAM | | |
|---|---|---|---|---|---|---|
| | TPR ↑ | FPR ↓ | F1 ↑ | TPR ↑ | FPR ↓ | F1 ↑ |
| BadNets | 0.0/**43.8** | **0.0**/0.8 | 0.0/**29.5** | 79.4/**81.4** | **5.3**/5.8 | **12.9**/12.2 |
| Blended | 0.0/**75.6** | **0.0**/**0.0** | 0.0/**86.1** | 15.2/**73.4** | **5.3**/5.7 | 2.6/**11.2** |
| SSBA | 0.0/**73.0** | **0.0**/**0.0** | 0.0/**80.3** | 13.2/**55.2** | **5.3**/5.9 | 2.3/**8.3** |
| LF | 0.0/**52.2** | **0.0**/0.2 | 0.0/**53.2** | 33.6/**35.6** | **5.3**/5.9 | **5.7**/5.5 |
| TrojanNN | 0.0/**100.0** | **0.0**/**0.0** | 0.0/**100.0** | 4.8/**99.6** | **5.3**/5.7 | 0.8/**15.0** |
| Average | +68.9 | +0.2 | +69.8 | +39.8 | +0.5 | +5.5 |

### C.2 MAIN EXPERIMENTS ON DIFFERENT MODELS

To explore the effects of SAM-enhanced PSD across different model architectures, we conduct a series of experiments using DenseNet-161, VGG19-BN, and the CIFAR-10 dataset. As shown in Tab. 5 and Tab. 6, the experimental results demonstrate that the implementation of SAM-enhanced PSD significantly improves detection performance in both the DenseNet-161 and the VGG19-BN models. However for the VGG19-BN model, although SAM-enhanced PSD enhanced the detection capabilities of this model, the improvement in TPR is still unsatisfactory under some backdoor attacks. The deep network architecture and large feature dimensionality of VGG19-BN result in a dispersion of backdoor signals within the feature space, making it challenging for SAM to effectively

concentrate and enhance these signals. Based on these findings, selecting an appropriate model architecture is crucial for enhancing detection outcomes. It is beneficial for the defender to adjust the model structure to improve detection performance, which can be fully controlled by the defender.

Table 5: Comparison of TPR (%) and FPR (%) between base PSD and SAM-enhanced PSD (+SAM) on CIFAR-10 and DenseNet-161 with poisoning ratio = 5%. Top 1 are **bold**. When comparing SAM-enhanced PSD to base PSD, performance improvements are highlighted in green, other changes in red.

| Detection → | Spectre / +SAM | | | SCAn / +SAM | | | SS / +SAM | | | AC / +SAM | | | Beatrix / +SAM | | |
|---|---|---|---|---|---|---|---|---|---|---|---|---|---|---|---|
| Attack ↓ | TPR ↑ | FPR ↓ | F1 ↑ | TPR ↑ | FPR ↓ | F1 ↑ | TPR ↑ | FPR ↓ | F1 ↑ | TPR ↑ | FPR ↓ | F1 ↑ | TPR ↑ | FPR ↓ | F1 ↑ |
| BadNets | 88.3/**96.4** | 2.9/**2.5** | 72.6/**79.2** | 47.7/**85.8** | 4.6/**0.0** | 40.7/**92.3** | 79.6/**81.2** | 0.9/**0.8** | 81.0/**82.7** | **90.4**/87.7 | 0.0/0.0 | **95.0**/93.5 | 25.7/**97.6** | 5.0/5.0 | 23.3/**66.7** |
| Blended | 43.6/**97.2** | 5.3/**2.4** | 35.8/**79.9** | 45.0/**89.2** | 4.5/**0.0** | 38.9/**94.3** | **84.1**/80.2 | **0.7**/0.9 | **85.6**/81.6 | **98.0**/95.0 | 0.0/0.0 | **98.9**/97.5 | 3.3/**98.8** | 5.0/5.0 | 3.3/**67.2** |
| SSBA | 53.8/**89.1** | 4.7/**2.9** | 44.2/**73.2** | 46.6/**68.6** | 4.6/**0.0** | 39.8/**81.3** | 79.4/**88.1** | **0.9**/1.5 | 80.8/**81.4** | **94.0**/88.5 | 0.0/0.0 | **96.7**/93.9 | 16.2/**92.4** | 5.0/5.0 | 15.4/**64.3** |
| LF | 59.0/**84.5** | 4.4/**3.1** | 48.4/**69.4** | **48.0**/20.8 | 4.7/**0.0** | **40.3**/34.4 | 79.3/**83.4** | **0.9**/2.3 | **80.7**/73.6 | **94.6**/91.3 | 0.0/0.0 | **97.1**/95.4 | 11.1/**62.4** | 5.0/5.0 | 10.8/**48.4** |
| Adap-Blend | 54.2/**92.8** | 4.2/**2.4** | 46.3/**78.0** | 47.4/**81.2** | **4.2**/10.4 | 41.6/**42.8** | **78.4**/74.2 | **0.8**/1.0 | **80.6**/76.5 | **96.2**/89.2 | 0.0/0.0 | **98.0**/94.3 | 7.0/**96.8** | 5.0/5.0 | 6.9/**66.3** |
| LC | 41.3/**65.6** | 3.0/**1.8** | 41.5/**65.9** | 0.0/**99.3** | 0.0/0.0 | 0.0/**99.7** | 29.1/**42.2** | 2.0/**1.3** | 34.9/**50.6** | **100.0**/99.3 | 0.0/0.0 | **99.9**/99.7 | 0.5/**90.4** | 5.0/5.0 | 0.5/**63.3** |
| TaCT | 19.3/**70.3** | 6.2/**0.5** | 16.3/**78.0** | 47.9/**99.3** | 5.3/**0.0** | 38.4/**99.6** | 39.6/**49.6** | 3.0/**1.9** | 40.2/**53.4** | 98.5/**99.3** | **0.0**/0.1 | **99.3**/98.9 | 2.5/**100.0** | 5.0/5.0 | 2.5/**67.8** |
| TrojanNN | 19.8/**79.7** | 6.5/**3.4** | 16.2/**65.5** | 48.0/**97.2** | 4.6/**0.0** | 40.7/**98.6** | 80.0/**89.7** | 0.9/**0.4** | 81.4/**91.3** | 95.8/**97.2** | 0.0/0.0 | 97.8/**98.6** | 3.2/**99.9** | 5.0/5.0 | 3.2/**67.7** |
| WaNet | 61.4/**82.5** | 4.4/**3.3** | 50.3/**67.1** | 48.2/**76.6** | 4.8/**0.0** | 40.3/**86.6** | 54.3/**73.4** | 2.3/**1.4** | 54.8/**73.6** | 0.0/**77.8** | 0.0/0.0 | 0.0/**87.3** | 6.6/**93.3** | 5.0/5.0 | 6.5/**64.7** |
| Input-aware | 63.9/**93.0** | 14.4/**0.3** | 29.2/**93.2** | 87.9/**98.6** | 0.0/0.0 | 93.5/**99.3** | 79.5/**82.0** | **0.0**/0.5 | **88.6**/85.9 | 0.0/**90.2** | 0.0/0.0 | 0.0/**94.9** | 4.3/**100.0** | 5.0/5.0 | 4.3/**67.8** |
| BppAttack | 25.5/**40.3** | 54.0/**7.1** | 4.4/**29.3** | 69.8/**93.6** | 0.0/0.0 | 82.2/**96.7** | **94.8**/83.3 | 7.5/**7.0** | **56.3**/52.6 | **99.7**/97.4 | **0.0**/0.6 | **99.8**/93.4 | 0.0/**95.2** | 5.0/5.0 | 0.0/**65.6** |
| SIBA | 39.8/**67.7** | **0.0**/4.8 | **56.9**/52.2 | 96.4/**100.0** | 16.6/**1.2** | 37.7/**90.1** | 75.1/**92.3** | 17.2/**0.0** | 30.0/**96.0** | 81.3/**99.4** | 25.7/**3.5** | 24.3/**74.8** | 3.2/**90.6** | 5.0/5.0 | 3.2/**63.4** |
| BadNets-A2A | 99.3/**99.4** | 10.6/10.6 | 49.7/49.7 | 0.0/0.0 | 0.0/0.0 | 0.0/0.0 | **99.6**/99.4 | 10.6/10.6 | **49.8**/49.7 | **97.2**/95.6 | 0.0/0.0 | **98.5**/97.7 | 32.7/**99.4** | 5.0/5.0 | 28.7/**67.5** |
| Average | **+29.9** | **−5.8** | **+28.3** | **+29.0** | **−3.2** | **+37.0** | **+5.1** | **−1.4** | **+8.0** | **+12.5** | **−1.7** | **+16.5** | **+84.7** | **−0.0** | **+56.3** |

Table 6: Comparison of TPR (%) and FPR (%) between base PSD and SAM-enhanced PSD (+SAM) on CIFAR-10 and VGG19-BN with poisoning ratio = 5%. Top 1 are **bold**. When comparing SAM-enhanced PSD to base PSD, performance improvements are highlighted in green, other changes in red.

| Detection → | Spectre / +SAM | | | SCAn / +SAM | | | SS / +SAM | | | AC / +SAM | | | Beatrix / +SAM | | |
|---|---|---|---|---|---|---|---|---|---|---|---|---|---|---|---|
| Attack ↓ | TPR ↑ | FPR ↓ | F1 ↑ | TPR ↑ | FPR ↓ | F1 ↑ | TPR ↑ | FPR ↓ | F1 ↑ | TPR ↑ | FPR ↓ | F1 ↑ | TPR ↑ | FPR ↓ | F1 ↑ |
| BadNets | 53.2/**74.2** | 4.7/**3.6** | 43.7/**61.0** | 0.0/**86.4** | **0.0**/0.1 | 0.0/**91.5** | 34.2/**57.8** | 3.3/**2.0** | 34.8/**58.8** | 0.0/0.0 | 0.0/0.0 | 0.0/0.0 | 4.0/**64.3** | 5.0/5.0 | 4.0/**49.6** |
| Blended | **76.8**/73.8 | **3.5**/6.1 | **63.1**/50.9 | 44.6/**93.9** | 4.4/**0.0** | 38.9/**96.9** | 42.5/**76.6** | 3.3/**2.0** | 43.2/**77.9** | 64.2/**97.9** | 0.0/0.0 | 78.0/**98.6** | 7.7/**99.4** | 5.0/5.0 | 7.6/**67.5** |
| SSBA | 40.3/**86.8** | 5.4/**3.0** | 33.1/**71.4** | 0.0/**61.0** | 4.6/**0.0** | 0.0/**75.7** | 33.4/**58.7** | 3.3/**2.0** | 34.0/**59.7** | 0.0/**58.1** | 0.0/0.0 | 0.0/**73.4** | 2.6/**77.4** | 5.0/5.0 | 2.7/**56.8** |
| LF | **84.5**/83.8 | 3.1/3.1 | **69.4**/68.9 | **47.2**/42.9 | 4.4/**0.0** | **41.0**/59.8 | 68.1/**68.8** | 1.5/1.5 | 69.3/**70.0** | 0.0/**22.0** | 0.0/0.0 | 0.0/**36.0** | 36.6/**80.2** | 5.0/5.0 | 31.6/**58.2** |
| Adap-Blend | 26.0/**55.9** | 5.5/**4.1** | 22.5/**47.7** | **45.4**/40.1 | 13.3/**10.5** | **22.8**/23.7 | 27.4/**40.7** | 3.3/**2.6** | 29.0/**42.7** | 0.0/**2.4** | **0.0**/0.1 | 0.0/**4.7** | 0.4/**15.2** | 5.0/5.0 | 0.4/**14.5** |
| LC | 28.8/**56.2** | **3.7**/4.5 | 28.9/**46.4** | 0.0/**100.0** | 0.0/0.0 | 0.0/**100.0** | 31.4/**40.2** | 1.9/**1.4** | 37.7/**48.2** | 99.6/**100.0** | 0.0/0.0 | 99.5/**100.0** | 1.0/**100.0** | 5.0/5.0 | 1.0/**67.8** |
| TaCT | 26.7/**45.4** | **5.4**/5.7 | 23.4/**35.7** | 46.5/**100.0** | 5.3/**0.0** | 37.6/**100.0** | 30.6/**37.7** | 4.0/**3.2** | 29.6/**37.9** | 100.0/100.0 | 0.0/0.0 | 100.0/100.0 | 1.1/**100.0** | 5.0/5.0 | 1.2/**67.8** |
| TrojanNN | 32.9/**55.5** | 5.8/**4.6** | 27.0/**45.6** | 42.5/**99.2** | 4.6/**0.0** | 36.8/**99.6** | 61.3/**92.6** | 1.9/**0.2** | 62.4/**94.2** | 99.2/**99.4** | 0.0/0.0 | 99.6/**99.7** | 0.8/**99.8** | 5.0/5.0 | 0.9/**67.7** |
| WaNet | 52.9/**72.8** | 4.8/**3.8** | 43.4/**59.4** | 0.0/0.0 | 0.0/0.0 | 0.0/0.0 | 18.0/**43.9** | 4.1/**2.8** | 18.4/**44.5** | 0.0/**9.6** | 0.0/0.0 | 0.0/**17.5** | 0.7/**44.4** | 5.0/5.0 | 0.7/**37.1** |
| Input-aware | 51.1/**99.0** | 6.0/**2.3** | 38.4/**81.5** | 95.5/**98.3** | 1.4/**0.0** | 85.9/**98.8** | **85.1**/82.1 | **0.0**/1.5 | **91.9**/78.3 | 0.0/**92.7** | 3.5/**0.3** | 0.0/**93.8** | 0.0/**100.0** | 5.0/5.0 | 0.0/**67.8** |
| BppAttack | 22.1/**40.4** | **5.1**/5.4 | 20.1/**33.2** | 83.9/**96.6** | 0.0/0.0 | 91.2/**98.3** | 81.7/**92.7** | **0.0**/1.2 | **89.9**/86.3 | 94.2/**96.5** | 0.0/0.0 | 97.0/**98.2** | 0.0/**99.3** | 5.0/5.0 | 0.0/**67.5** |
| SIBA | 25.2/**64.2** | 6.2/**5.0** | 20.8/**49.7** | **100.0**/99.0 | 0.0/0.0 | **100.0**/99.5 | 74.6/**91.4** | 7.0/**0.0** | 79.6/**95.4** | 95.3/**99.2** | 5.9/**0.0** | 61.9/**99.6** | 5.1/**90.9** | 5.0/5.0 | 5.1/**63.6** |
| BadNets-A2A | **98.8**/91.2 | **10.6**/11.0 | **49.4**/45.6 | 0.0/0.0 | 0.0/0.0 | 0.0/0.0 | 23.8/**97.8** | 14.5/**10.6** | 11.9/**48.9** | 0.0/**64.9** | **0.0**/0.3 | 0.0/**76.5** | 24.6/**74.2** | 5.0/5.0 | 22.4/**55.1** |
| Average | **+21.6** | **−0.6** | **+16.4** | **+31.7** | **−2.1** | **+37.7** | **+20.7** | **−0.9** | **+16.2** | **+22.3** | **−0.6** | **+27.9** | **+73.9** | **−0.0** | **+51.0** |

## C.3 EXPERIMENTS UNDER DIFFERENT POISONING RATIOS

To demonstrate the stability and effectiveness of SAM-enhanced PSD against weak backdoor attacks, we evaluate its detection performance at various low poisoning ratios, focusing on the 1% and 0.5% settings. We also discuss the extreme scenario of 0.1% poisoning.

As shown in Tab. 7 and Tab. 8, our method provides substantial and consistent performance gains in these weak backdoor scenarios. At a 1% poisoning ratio, SAM-enhanced PSD boosts the average True Positive Rate, TPR, across all detection methods, with particularly large average gains for SS at +47.0% and AC at +85.4%. This strong performance is maintained at the more challenging 0.5% ratio, where our method again significantly improves the average TPR for detectors like Beatrix and SCAn with gains of +80.7% and +27.9%, respectively, all while keeping FPR low. These results confirm our method's efficacy in amplifying the backdoor signal in weak attack settings.

As shown in Tab. 9, we also investigate the extreme scenario of 0.1% poisoning. At this ratio, the detection challenge becomes fundamental. We find that the potency of the backdoor attacks themselves is intrinsically limited, with the average ASR, across all tested attacks being merely 31.8%, which implies that the backdoor signal is inherently weak and ambiguous. Despite this inherent challenge, SAM-enhanced PSD still demonstrates a significant enhancement effect, providing considerable relative performance improvements for the base detectors. For instance, it achieves a +27.7% TPR gain for Spectre. This shows that our approach effectively amplifies backdoor features, even when the signal is at its weakest.

## C.4 ANALYSIS WITH NON-FEATURE-BASED DETECTORS

To comprehensively define the scope of our SAM-enhanced framework, we investigate its effectiveness when applied to other categories of detectors, including perturbation-based and topology-based methods.

Table 7: Comparison of TPR (%) and FPR (%) between base PSD and SAM-enhanced PSD (+SAM) on CIFAR-10 and ResNet18 with poisoning ratio=1%. Top 1 are **bold**. When comparing SAM-enhanced PSD to base PSD, performance improvements are highlighted in green, other changes in red.

| Detection → | Spectre / +SAM | | | SCAn / +SAM | | | SS / +SAM | | | AC / +SAM | | | Beatrix / +SAM | | |
|---|---|---|---|---|---|---|---|---|---|---|---|---|---|---|---|
| Attack ↓ | TPR ↑ | FPR ↓ | F1 ↑ | TPR ↑ | FPR ↓ | F1 ↑ | TPR ↑ | FPR ↓ | F1 ↑ | TPR ↑ | FPR ↓ | F1 ↑ | TPR ↑ | FPR ↓ | F1 ↑ |
| BadNets | 78.0/**83.8** | 0.7/0.7 | 62.4/**67.0** | **92.4**/91.6 | 0.0/0.0 | **96.0**/95.6 | 66.4/**96.2** | 0.3/0.0 | 66.7/**96.6** | 0.0/**91.6** | 0.0/0.0 | 0.0/**95.6** | 49.8/**99.2** | 5.0/5.0 | 15.4/**28.5** |
| Blended | 19.8/**57.4** | 1.3/0.9 | 15.8/**45.9** | 95.4/**96.2** | 0.0/0.0 | 97.6/**98.1** | 8.2/**99.0** | 0.9/0.0 | 8.2/**99.4** | 0.0/**96.2** | 0.0/0.0 | 0.0/**98.1** | 8.2/**100.0** | 5.0/5.0 | 2.7/**28.7** |
| SSBA | 70.4/**97.4** | 0.8/0.5 | 56.3/**77.9** | **92.2**/90.4 | 0.0/0.0 | **95.9**/95.0 | 3.6/**97.4** | 1.0/0.0 | 3.6/**97.8** | 0.0/**90.4** | 0.0/0.1 | 0.0/**90.6** | 3.4/**99.4** | 5.0/5.0 | 1.1/**28.6** |
| LF | **60.6**/52.0 | 0.9/1.0 | **48.5**/41.6 | 88.4/**90.6** | 0.0/0.0 | 93.8/**95.1** | 54.2/**96.2** | 0.5/0.0 | 54.4/**96.6** | 0.0/**90.6** | 0.0/0.0 | 0.0/**95.1** | 2.6/**95.6** | 5.0/5.0 | 0.9/**27.6** |
| Adap-Blend | 71.9/**88.5** | 0.8/0.6 | 57.3/**70.5** | **93.1**/91.5 | 2.0/0.0 | 47.4/**95.6** | 55.9/**98.0** | 0.4/0.0 | 56.5/**98.8** | 0.0/**91.5** | 0.0/0.0 | 0.0/**95.6** | 9.7/**98.8** | 5.0/5.0 | 3.2/**28.4** |
| LC | 4.0/**42.2** | **1.5**/2.6 | 3.2/**21.1** | 100.0/**100.0** | 0.0/0.0 | 100.0/**100.0** | 60.4/**91.0** | 0.3/0.0 | 63.2/**95.3** | 0.0/**100.0** | 0.0/0.0 | 0.0/**100.0** | 23.4/**100.0** | 5.0/5.0 | 7.6/**28.8** |
| TaCT | 0.1/**33.0** | 1.6/1.1 | 0.1/**27.5** | 100.0/**100.0** | 0.0/0.0 | 100.0/**100.0** | 8.0/**24.3** | 1.0/0.0 | 7.6/**39.1** | 0.0/**100.0** | 0.0/0.0 | 0.0/**100.0** | 36.0/**100.0** | 5.0/5.0 | 11.4/**28.8** |
| TrojanNN | **96.4**/51.4 | 0.5/0.7 | **77.1**/16.1 | 100.0/**100.0** | 0.0/0.0 | 0.0/**83.6** | 50.9/**84.9** | 0.6/0.3 | 48.8/**78.6** | 0.0/**71.1** | 0.0/0.0 | 0.0/**83.1** | 25.8/**100.0** | 5.0/5.0 | 8.3/**28.7** |
| WaNet | 71.4/**86.4** | 0.9/0.8 | 53.8/**64.4** | 0.0/**71.9** | 0.0/0.0 | 0.0/**83.6** | 50.9/**84.9** | 0.6/0.3 | 48.8/**78.6** | 0.0/**71.1** | 0.0/0.0 | 0.0/**83.1** | 1.8/**83.6** | 5.0/5.0 | 0.6/**24.6** |
| Input-aware | 69.6/**96.7** | 1.0/0.7 | 52.5/**71.4** | 81.6/**92.8** | 0.0/0.0 | 88.2/**94.2** | 75.7/**99.5** | 0.4/0.2 | 70.7/**90.8** | 0.0/**93.6** | 0.0/0.1 | 0.0/**93.5** | 15.9/**62.1** | 5.0/5.0 | 5.2/**18.9** |
| BppAttack | 94.9/**98.0** | 0.8/0.7 | 70.2/**72.2** | 67.0/**94.9** | 0.0/0.0 | 80.2/**97.4** | 94.1/**98.0** | 0.2/0.2 | 86.3/**89.5** | 69.6/**94.6** | 0.0/0.0 | 82.1/**97.2** | 2.0/**98.5** | 5.0/5.0 | 0.7/**28.4** |
| SIBA | 0.2/**16.2** | 1.5/1.4 | 0.2/**13.0** | 99.8/98.8 | 0.0/0.0 | 99.9/99.4 | 0.0/**99.2** | 1.0/0.0 | 0.0/**99.6** | 0.0/**99.0** | 0.0/0.0 | 0.0/**99.5** | 2.0/**97.2** | 5.0/5.0 | 0.7/**28.0** |
| BadNets-A2A | 99.8/**100.0** | 14.1/14.1 | 12.5/12.5 | 0.0/0.0 | 0.0/0.0 | 0.0/0.0 | 93.8/**100.0** | 14.2/14.1 | 11.7/**12.5** | 31.2/**93.6** | 0.0/0.0 | 47.6/**96.7** | 33.4/**99.4** | 5.0/5.0 | 10.6/**28.6** |
| Average | +12.8 | +0.3 | +7.0 | +8.4 | −0.2 | +12.0 | +47.0 | -0.6 | +47.4 | +85.4 | -0.0 | +85.8 | +78.4 | -0.0 | +22.1 |

Table 8: Comparison of TPR (%) and FPR (%) between base PSD and SAM-enhanced PSD (+SAM) on CIFAR-10 and ResNet18 with poisoning ratio=0.5%. Top 1 are **bold**. When comparing SAM-enhanced PSD to base PSD, performance improvements are highlighted in green, other changes in red.

| Detection → | Spectre / +SAM | | | SCAn / +SAM | | | SS / +SAM | | | AC / +SAM | | | Beatrix / +SAM | | |
|---|---|---|---|---|---|---|---|---|---|---|---|---|---|---|---|
| Attack ↓ | TPR ↑ | FPR ↓ | F1 ↑ | TPR ↑ | FPR ↓ | F1 ↑ | TPR ↑ | FPR ↓ | F1 ↑ | TPR ↑ | FPR ↓ | F1 ↑ | TPR ↑ | FPR ↓ | F1 ↑ |
| BadNets | 90.0/**98.4** | 1.1/1.0 | 45.0/**49.2** | 0.0/**88.4** | 0.0/0.0 | 0.0/**93.8** | 78.4/**97.6** | 1.2/1.1 | 37.9/**47.2** | 84.0/**88.4** | **1.9**/8.2 | 30.1/9.7 | 2.0/**97.6** | 0.5/1.6 | 2.0/**37.7** |
| Blended | 28.4/**99.2** | 1.4/1.0 | 14.2/**49.6** | 0.0/**96.8** | 0.0/0.0 | 0.0/**98.4** | 39.2/**99.2** | 1.4/1.1 | 19.0/**48.0** | 0.0/**96.4** | **0.0**/12.0 | 0.0/7.5 | 0.0/**98.4** | 0.4/0.9 | 0.0/**52.0** |
| SSBA | **92.4**/92.0 | 1.0/1.0 | **46.2**/46.0 | **77.6**/66.0 | 0.0/0.0 | **87.4**/79.5 | 78.8/**91.2** | 1.2/1.1 | 38.1/**44.1** | **66.8**/66.0 | **3.3**/8.0 | **16.4**/7.5 | 0.0/**78.0** | 0.5/1.9 | 0.0/**21.3** |
| LF | 94.0/**95.6** | 1.0/1.0 | 47.0/**47.8** | 0.0/0.0 | 0.0/0.0 | 0.0/0.0 | 58.8/**95.6** | 1.3/1.1 | 28.5/**46.3** | 0.0/**79.6** | **6.6**/14.0 | 0.0/**5.4** | 0.0/**78.0** | 0.5/1.9 | 0.0/**27.9** |
| Adap-Blend | 97.2/**99.2** | 1.0/1.0 | 48.6/**49.6** | 0.0/**90.4** | 0.0/0.0 | 0.0/**95.0** | 78.8/**98.4** | 1.2/1.1 | 38.4/**48.0** | 62.4/**90.0** | **6.5**/14.4 | **8.6**/5.9 | 0.0/**98.4** | 0.3/4.1 | 1.0/**19.5** |
| LC | **14.0**/11.6 | 1.4/1.4 | **7.0**/5.8 | 100.0/**100.0** | 0.0/0.0 | 100.0/**100.0** | 97.6/**100.0** | 1.0/1.0 | 48.8/**50.0** | 99.6/**100.0** | **7.6**/13.5 | **11.7**/6.9 | 0.0/**99.6** | 0.5/5.0 | 0.0/**16.7** |
| TaCT | 0.7/**10.3** | 1.5/1.0 | 0.3/**6.7** | 100.0/**100.0** | 0.0/0.0 | 100.0/**100.0** | 16.0/**42.3** | 1.5/0.0 | 7.6/**59.4** | 0.0/**100.0** | 3.4/0.0 | 0.0/**100.0** | 0.0/**100.0** | 5.0/5.0 | 0.0/**16.7** |
| TrojanNN | **12.4**/5.6 | **1.4**/1.5 | **6.2**/2.8 | 100.0/**100.0** | **0.0**/4.3 | **100.0**/19.1 | 11.6/**100.0** | 1.5/1.1 | 5.6/**48.4** | 0.0/**100.0** | **6.8**/14.7 | 0.0/**6.4** | 0.4/**96.0** | 0.7/5.0 | 0.3/**16.1** |
| WaNet | 50.8/**84.0** | 0.5/0.3 | 40.6/**67.2** | 0.0/**61.2** | 0.0/0.0 | 0.0/**75.9** | 36.8/**81.6** | 0.3/0.1 | 36.8/**81.6** | 0.0/**60.8** | 0.0/0.0 | 0.0/**75.6** | 4.8/**84.0** | 5.0/5.0 | 0.9/**14.2** |
| Input-aware | 28.0/**97.6** | 0.6/0.3 | 22.4/**77.6** | 0.0/0.0 | 0.0/0.0 | 0.0/0.0 | 82.5/**91.1** | 0.1/0.1 | 82.1/**90.6** | 0.0/**79.3** | **0.0**/0.1 | 0.0/**77.2** | 1.2/**77.6** | 5.0/5.0 | 0.2/**13.2** |
| BppAttack | 92.4/**98.0** | 0.3/0.3 | 73.9/**78.4** | 55.2/**92.8** | 0.0/0.0 | 71.1/**96.3** | 90.8/**97.6** | 0.0/0.0 | 90.8/**97.6** | 63.2/**92.8** | 0.0/0.0 | 77.5/**96.3** | 0.4/**98.0** | 5.0/5.0 | 0.1/**16.4** |
| SIBA | 1.6/**5.6** | 0.7/0.7 | 1.3/**4.5** | 99.2/**99.2** | 0.0/0.0 | 99.6/**99.6** | 0.0/**99.6** | 0.5/0.0 | 0.0/**99.8** | 0.0/**98.0** | 0.0/0.0 | 0.0/**99.0** | 7.6/**21.6** | 5.0/5.0 | 1.4/**3.9** |
| BadNets-A2A | 98.0/**98.4** | 14.6/14.6 | 6.3/6.3 | 0.0/0.0 | 0.0/0.0 | 0.0/0.0 | 48.8/**98.8** | 14.8/14.6 | 3.1/**6.4** | 0.0/**90.8** | 0.0/0.0 | 0.0/**95.2** | 13.6/**71.6** | 0.7/2.4 | 10.7/**21.9** |
| Average | +15.1 | −0.1 | +10.2 | +27.9 | +0.3 | +23.1 | +36.6 | -0.3 | +25.4 | +59.0 | +3.7 | +34.5 | +80.7 | +1.8 | +20.0 |

We applied our framework to representative defenses from both categories. The results presented in Tab. 10 show that the performance gains for these non-feature-based methods are less pronounced than the substantial gains demonstrated for feature-based detectors in Table 1. We posit that this outcome is not a limitation, but rather an important finding that clarifies the core mechanism of our framework and defines its operational boundaries.

The reasons are linked to SAM's optimization objective. For perturbation-based detectors, their effectiveness relies on the inference instability of poisoned samples. SAM's objective of finding flat minima, however, improves overall model stability, which inadvertently suppresses the signal these detectors require. Similarly, topology-based detectors identify structural anomalies in the feature manifold. The smoothening effect of SAM's optimization can mask the subtle topological artifacts these methods are designed to find.

This investigation, therefore, clarifies the contribution of our framework: **it acts as a dedicated signal amplifier for defenses that leverage feature-space disparity**. Because SAM's core effect is to increase the separability between poisoned and clean samples in the feature space, it creates a powerful synergy with feature-based detectors. This analysis provides a clear scope for our method's application and reinforces the understanding of its underlying mechanism.

## C.5 ADAPTIVE ATTACK

In this chapter, we propose an *adaptive backdoor attack* specifically designed to compromise the separation capability of SAM-enhanced-PSD. Unlike traditional adaptive attacks that primarily consider the detector's mechanisms, our threat model extends further: we assume that the adversary possesses comprehensive knowledge of the training method, including the Sharpness-Aware Minimisation (SAM) that SAM-enhanced-PSD employs during model training to improve detection performance.

**Attack goal and assumptions.** The goal of the adversary is to diminish the feature-space distinction between poisoned samples and target clean samples, thereby evading detection. Under our adaptive attack setting, the adversary enjoys complete white-box access, encompassing full visibility into the network architecture, optimization hyperparameters, and SAM optimization. By integrating the SAM training step into the adversary's optimization process, the adversary explicitly aims to neutralize

Table 9: Comparison of TPR (%) and FPR (%) between base PSD and SAM-enhanced PSD (+SAM) on CIFAR-10 and ResNet18 with poisoning ratio=0.1%. Top 1 are **bold**. When comparing SAM-enhanced PSD to base PSD, performance improvements are highlighted in green, other changes in red.

| Detection → | Spectre / +SAM | | | SCAn / +SAM | | | SS / +SAM | | | AC / +SAM | | | Beatrix / +SAM | | |
|---|---|---|---|---|---|---|---|---|---|---|---|---|---|---|---|
| Attack ↓ | TPR ↑ | FPR ↓ | F1 ↑ | TPR ↑ | FPR ↓ | F1 ↑ | TPR ↑ | FPR ↓ | F1 ↑ | TPR ↑ | FPR ↓ | F1 ↑ | TPR ↑ | FPR ↓ | F1 ↑ |
| BadNets | 2.0/58.0 | 0.1/0.1 | 1.6/46.4 | 0.0/0.0 | 0.0/0.0 | 0.0/0.0 | 0.0/28.0 | 0.1/0.1 | 0.0/28.0 | 0.0/0.0 | 0.0/0.0 | 0.0/0.0 | 36.0/62.0 | 5.0/5.0 | 1.4/2.4 |
| Blended | 8.0/8.0 | 0.1/0.1 | 6.4/6.4 | 0.0/0.0 | 0.0/0.0 | 0.0/0.0 | 0.0/6.0 | 0.1/0.1 | 0.0/6.0 | 0.0/0.0 | 0.0/0.0 | 0.0/0.0 | 8.0/38.0 | 5.0/5.0 | 0.3/1.5 |
| SSBA | 16.0/50.0 | 0.1/0.1 | 12.8/40.0 | 0.0/0.0 | 0.0/0.0 | 0.0/0.0 | 0.0/30.0 | 0.1/0.1 | 0.0/30.0 | 0.0/0.0 | 0.0/0.0 | 0.0/0.0 | 28.0/42.0 | 5.0/5.0 | 1.1/1.6 |
| LF | 2.0/12.0 | 0.1/0.1 | 1.6/9.6 | 0.0/0.0 | 0.0/4.2 | 0.0/0.0 | 0.0/8.0 | 0.1/0.1 | 0.0/8.0 | 0.0/0.0 | 0.0/0.0 | 0.0/0.0 | 30.0/14.0 | 5.0/5.0 | 1.2/0.5 |
| Adap-Blend | **14.0**/8.0 | 0.1/0.1 | 11.2/6.4 | 0.0/0.0 | 0.0/0.0 | 0.0/0.0 | 0.0/6.0 | 0.1/0.1 | 0.0/6.0 | 0.0/0.0 | 0.0/0.0 | 0.0/0.0 | 30.0/28.0 | 5.0/5.0 | 1.2/1.1 |
| LC | 0.0/100.0 | 0.2/0.1 | 0.0/80.0 | 0.0/0.0 | 0.0/0.0 | 0.0/0.0 | 0.0/100.0 | 0.1/0.0 | 0.0/100.0 | 0.0/100.0 | 0.0/0.0 | 0.0/100.0 | **10.0**/0.0 | 5.0/5.0 | **0.4**/0.0 |
| TaCT | 0.1/0.3 | 0.2/0.1 | 0.1/0.2 | 100.0/100.0 | 0.0/0.0 | 100.0/100.0 | 0.0/2.9 | 0.2/0.0 | 0.0/5.7 | 100.0/100.0 | 0.0/0.0 | 100.0/100.0 | 22.2/100.0 | 5.0/5.0 | 0.9/3.8 |
| TrojanNN | 12.0/100.0 | 0.1/0.1 | 9.6/80.0 | 0.0/0.0 | 0.0/0.0 | 0.0/0.0 | 0.0/100.0 | 0.1/0.0 | 0.0/100.0 | 0.0/100.0 | 0.0/0.0 | 0.0/100.0 | 10.0/100.0 | 5.0/5.0 | 0.4/3.8 |
| WaNet | **43.8**/20.8 | 0.1/0.1 | 34.8/16.6 | 0.0/0.0 | 0.0/0.0 | 0.0/0.0 | 41.7/4.2 | 0.1/0.1 | 41.3/4.2 | 0.0/0.0 | 0.0/0.1 | 0.0/0.0 | 4.2/60.4 | 5.0/5.0 | 0.2/2.3 |
| Input-aware | 94.2/95.8 | 0.1/0.1 | 76.5/56.4 | 0.0/0.0 | 0.0/0.0 | 0.0/0.0 | 17.3/32.7 | 0.1/0.1 | 17.4/32.9 | 0.0/0.0 | 0.0/0.0 | 0.0/0.0 | 3.8/55.8 | 5.0/5.0 | 0.2/2.2 |
| BppAttack | 41.7/52.9 | 0.1/0.1 | 33.1/37.7 | 0.0/0.0 | 0.0/0.0 | 0.0/0.0 | 10.4/10.4 | 0.1/0.1 | 10.4/10.4 | 0.0/0.0 | 0.0/0.0 | 0.0/0.0 | 18.8/27.1 | 5.0/5.0 | 0.7/1.1 |
| SIBA | 0.0/84.0 | 0.2/0.1 | 0.0/67.2 | 0.0/0.0 | 0.0/0.0 | 0.0/0.0 | 2.0/50.0 | 0.1/0.1 | 2.0/50.0 | 0.0/0.0 | 0.0/0.0 | 0.0/0.0 | 4.0/58.0 | 5.0/5.0 | 0.2/2.2 |
| BadNets-A2A | 76.0/80.0 | 14.9/**14.9** | 1.0/1.1 | 0.0/0.0 | 0.0/0.0 | 0.0/0.0 | 24.0/56.0 | 15.0/15.0 | 0.3/0.7 | 0.0/0.0 | 0.0/0.0 | 0.0/0.0 | **80.0**/78.0 | 5.0/5.0 | **3.1**/3.0 |
| Average | +27.7 | −0.1 | +20.0 | −0.0 | +0.3 | −0.0 | +26.1 | −0.1 | +23.9 | +15.4 | −0.0 | +15.4 | +29.1 | −0.0 | +1.1 |

Table 10: Comparison of TPR (%) and FPR (%) between other non-feature-based PSD and SAM-enhanced PSD (+SAM) on CIFAR-10 and ResNet18 with poisoning ratio = 5%. Top 1 are **bold**. When comparing SAM-enhanced PSD to base PSD, performance improvements are highlighted in green, other changes in red.

| Detection → | CD / +SAM | | | STRIP / +SAM | | | SentiNet / +SAM | | | TED / +SAM | | |
|---|---|---|---|---|---|---|---|---|---|---|---|---|
| Attack ↓ | TPR ↑ | FPR ↓ | F1 ↑ | TPR ↑ | FPR ↓ | F1 ↑ | TPR ↑ | FPR ↓ | F1 ↑ | TPR ↑ | FPR ↓ | F1 ↑ |
| BadNets | 63.8/**67.2** | 5.0/**5.0** | 49.5/**51.3** | 85.6/**93.3** | 11.1/**9.4** | 43.3/**50.1** | 35.3/**36.7** | 13.1/**12.4** | 18.4/**19.7** | **86.5**/55.3 | 33.0/**20.3** | **21.3**/20.4 |
| Blended | 4.2/**14.4** | **4.9**/5.0 | 4.2/**13.8** | **71.0**/59.8 | 11.0/**10.7** | **37.3**/33.0 | 2.9/**22.9** | **4.2**/21.2 | 3.2/**8.7** | 26.0/**35.2** | 23.3/**20.4** | 9.1/**13.5** |
| SSBA | **66.4**/62.8 | 5.0/**5.0** | **50.8**/48.7 | 78.7/**86.8** | 8.9/**8.5** | 45.3/**49.9** | 1.9/**20.9** | 1.8/1.8 | 2.7/**27.0** | 66.3/**71.6** | 22.2/**18.2** | 22.6/**27.6** |
| LF | 5.8/**42.9** | 5.0/**5.0** | 5.8/**36.1** | 86.8/**89.2** | **11.1**/11.3 | 43.6/**44.2** | 0.0/**36.2** | **0.0**/1.2 | 0.0/**45.8** | 42.4/**51.8** | 15.2/**10.8** | 19.6/**29.0** |
| Average | +11.7 | −0.0 | +9.9 | +1.8 | −0.5 | +1.9 | +19.2 | +4.4 | +19.2 | −1.8 | −6.0 | +4.4 |

SAM's beneficial effects, specifically its capacity to widen feature gaps, making poisoned samples indistinguishable from target clean samples during detection.

**Attack methodology.** Our proposed attack extends Adapt-Blend (Qi et al., 2023a), a backdoor attack method that reduces detectability by randomly cropping the commonly used "Hello Kitty" trigger, assigning random labels, and lowering the poisoning rate, thereby diminishing the feature-space gap between poisoned samples and target clean samples. To further enhance this reduction of the feature gap and effectively counteract Sharpness-Aware Minimization (SAM), we propose replacing the commonly used trigger with an optimized trigger generated through a novel bi-level optimization approach. Specifically, we formulate this optimization as:

- *Outer minimization*: shrinks the distance between features of poisoned samples and target clean samples, explicitly pulling poisoned representations towards the clean target class.
- *Inner maximization*: replicates the perturbation step inherent to SAM training, ensuring the optimized trigger anticipates and counters SAM's attempt to re-expand feature distances.

This approach can be formally expressed as:

$$\min_{\boldsymbol{\epsilon}, \boldsymbol{\theta}} \max_{\boldsymbol{\delta}} \ \mathbb{E}_{(\boldsymbol{x},y)\in\mathcal{D}_c}\left[\mathcal{L}(f_{\boldsymbol{\theta}+\boldsymbol{\delta}}(\boldsymbol{x}), y)\right]$$
$$+ \ \mathbb{E}_{(\boldsymbol{x},y)\in\mathcal{D}_p, (\boldsymbol{x}_t, y_t)\in\mathcal{D}_t}\left[\mathcal{L}(f_{\boldsymbol{\theta}+\boldsymbol{\delta}}((1-\alpha)\boldsymbol{x} + \alpha\boldsymbol{\epsilon}), y_t) + \lambda\|f_{\boldsymbol{\theta}+\boldsymbol{\delta}}((1-\alpha)\boldsymbol{x} + \alpha\boldsymbol{\epsilon}) - f_{\boldsymbol{\theta}+\boldsymbol{\delta}}(\boldsymbol{x}_t)\|_2^2\right], \quad (10)$$

where $\mathcal{D}_c$ is the clean subset, $\mathcal{D}_p$ contains images to be poisoned, and $\mathcal{D}_t$ is the set of clean samples with the target label. The blending ratio $\alpha$ controls the proportion between the original images $\boldsymbol{x}$ and the optimized trigger $\boldsymbol{\epsilon}$. The first expectation maintains model performance on clean data, while the second explicitly learns poisoned samples and reduces the feature-space distance between poisoned samples and target clean samples. The inner maximization step ensures the trigger adapts to SAM's perturbations, weakening SAM's subsequent effectiveness.

**Implementation details.** We follow the Adapt-Blend recipe pasted with a 20 % opacity onto each poisoned image. The poisoning ratio is set to 0.3 %. Both the outer training loop and the inner maximization use the same SAM hyperparameters as SAM-enhanced-PSD, namely a neighbourhood radius of $\rho = 0.1$. In the loss of Eq. (10) we fix the gap-shrinking weight to $\lambda = 0.1$.

**Results and analysis.** As demonstrated in Tab. 11, applying the Adapt-Blend severely degrades the performance of standard PSD detectors (Spectre, SS, AC, and Beatrix). Specifically, their

Table 11: Comparison of TPR (%) and FPR (%) between baseline PSD and SAM-enhanced PSD (+SAM) under adaptive attacks on CIFAR-10 with ResNet-18.

| Trigger → | Hello-Kitty | | | Optimized Trigger | | |
|---|---|---|---|---|---|---|
| Detection ↓ | TPR ↑ | FPR ↓ | F1 ↑ | TPR ↑ | FPR ↓ | F1 ↑ |
| Spectre | 14.67 / **97.33** | 0.41 / **0.16** | 11.69 / **77.71** | 1.33 / **100.0** | 0.45 / **0.15** | 1.06 / **80.05** |
| SS | 6.00 / **96.67** | 0.28 / **0.01** | 6.03 / **96.67** | 0.00 / **99.33** | 0.30 / **0.00** | 0.00 / **99.67** |
| AC | 0.00 / **94.67** | 0.00 / 0.00 | 0.00 / **97.26** | 0.00 / **100.0** | 0.00 / 0.00 | 0.00 / **100.0** |
| Beatrix | 10.67 / **96.67** | 5.01 / 5.01 | 1.20 / **10.39** | 0.67 / **97.33** | 5.01 / 5.01 | 0.08 / **10.45** |

average True Positive Rate (TPR) drops below 11% for the manually crafted Hello-Kitty patch and near-zero for the optimized trigger, validating the effectiveness of feature-gap minimization. Nevertheless, the SAM-enhanced PSD consistently recovers detection capability, achieving TPRs above 94% and False Positive Rates (FPRs) under 0.2% for both triggers. These outcomes highlight SAM's intrinsic robustness, whose feature-space expansion capability effectively counters even an explicitly adaptive adversary. Consequently, unless attackers gain full control over the training procedure, neutralizing SAM-enhanced-PSD remains notably challenging, ensuring strong and persistent detection performance.

## C.6 PERFORMANCE ON THE MODEL TRAINED BY FILTERED DATA.

To validate the effectiveness of the SAM-enhanced PSD, we also systematically measure performances in accuracy (ACC), attack success rate (ASR), and robust accuracy (RA) during model retraining after removing poisoned samples identified by PSDs and SAM-enhanced PSDs, where higher ACC/RA and lower ASR indicate better defense performance.

As shown in Tab. 12, experimental results reveal SAM-enhanced PSD impact across multiple attack scenarios. In BadNets (Gu et al., 2019) attacks, SCAn (Tang et al., 2021) detection coupled with SAM improved ACC from 94.1% to 94.4% and RA from 94.0% to 94.3% while maintaining ASR at 0.7%. More strikingly, against challenging Blended (Chen et al., 2017) attacks, SAM reduced AC (Chen et al., 2019) method's ASR from 99.7% to 2.7% while boosting RA from 0.3% to 79.1%. Similar patterns emerged in adaptive attacks like Adap-Blend (Qi et al., 2023a), LF (Zeng et al., 2021), and WaNet (Nguyen & Tran, 2021), where SAM achieved over 90% ASR reduction and 80% RA improvement in critical cases. Aggregate statistics across all attack types show SAM delivers 0.2%-0.5% ACC gains, up to 93.8% ASR suppression, and remarkable RA enhancements reaching 84%. These improvements stem from SAM's ability to amplify activation pattern differences between clean and poisoned samples in feature space through sharpness-aware optimization, thereby increasing PSD discriminative power.

The findings conclusively demonstrate that incorporating Sharpness-Aware Minimization into PSD training frameworks significantly strengthens model resilience against diverse backdoor threats. By simultaneously improving detection accuracy (ACC/RA) and suppressing attack success rates (ASR), SAM establishes a robust defense paradigm applicable to complex adversarial environments, providing critical insights for developing next-generation AI security solutions.

## C.7 COMPARISON WITH INFERENCE-TIME DETECTION METHODS

In this section, we extend our analysis to include a comparison with state-of-the-art training-free detection methods that operate at inference time, specifically Scale-up (Guo et al., 2023) and CBD (Xiang et al., 2023). It is important to first clarify that our pre-training defense and these inference-time defenses address different stages of the security pipeline and are thus complementary. Our method aims to provide a one-time, permanent "cure" by cleansing the training dataset itself, ensuring that any model trained on it is inherently more secure without runtime overhead. In contrast, inference-time defenses act as a perpetual "treatment," requiring continuous monitoring and incurring per-inference costs to block malicious inputs at runtime, while the underlying model remains vulnerable.

Despite these conceptual differences, we conduct a direct performance comparison on the CIFAR-10 dataset to evaluate their practical detection capabilities. To simulate a realistic scenario where false

Table 12: Detection comparisons (measured by ACC (%), ASR (%) and RA (%)) between base PSD and SAM-enhanced PSD (+SAM) on CIFAR-10 and ResNet18, and the better result in each pair is highlighted in **bold**. In terms of each metric, the average change of SAM-enhanced PSD to base PSD across all attacks is presented at the bottom: performance improvements are highlighted in green, other changes in red.

| Detection → | SCAn / +SAM | | | AC / +SAM | | | Beatrix / +SAM | | |
|---|---|---|---|---|---|---|---|---|---|
| Attack ↓ | ACC ↑ | ASR ↓ | RA ↑ | ACC ↑ | ASR ↓ | RA ↑ | ACC ↑ | ASR ↓ | RA ↑ |
| BadNets | 94.1/**94.4** | 0.7/**0.7** | 94.0/**94.3** | **94.6**/94.5 | **0.7**/0.8 | **94.5**/94.3 | 93.3/**94.5** | 91.0/**0.7** | 8.5/**94.2** |
| Blended | **94.6**/94.5 | 6.7/**4.4** | 76.8/**77.8** | 93.1/**94.8** | 99.7/**2.7** | 0.3/**79.1** | 93.8/**94.2** | 99.4/**3.3** | 0.6/**78.4** |
| SSBA | 94.4/**94.4** | 1.2/**1.0** | 91.6/**92.2** | 94.0/**94.4** | **0.8**/1.0 | 91.7/**92.2** | **94.1**/93.8 | 95.3/**0.8** | 4.6/**91.6** |
| LF | 94.2/**94.6** | 9.5/**3.0** | 85.1/**90.7** | 93.6/**94.6** | 7.4/**3.0** | 86.4/**90.7** | 94.0/**94.1** | 98.0/**1.6** | 1.9/**92.1** |
| Adap-Blend | 94.0/**94.3** | 83.0/**6.0** | 15.8/**75.8** | 91.7/**93.9** | 99.9/**6.9** | 0.0/**74.6** | **93.8**/93.6 | 100.0/**1.9** | 0.0/**79.1** |
| LC | **94.6**/94.3 | 0.7/**0.7** | **92.9**/92.5 | 94.2/**94.3** | 100.0/**0.7** | 0.0/**92.8** | **94.4**/94.0 | 100.0/**0.6** | 0.0/**92.9** |
| TaCT | 94.0/**94.0** | 1.2/**1.2** | 86.7/**86.7** | **94.2**/94.0 | 1.6/**1.2** | **89.1**/86.7 | 93.8/**94.2** | 100.0/**0.6** | 0.0/**87.8** |
| TrojanNN | **94.6**/94.5 | **3.1**/3.6 | 86.6/**87.0** | 94.2/**94.4** | 2.7/**2.9** | 86.4/**87.3** | **94.4**/94.2 | 100.0/**2.0** | 0.0/**87.4** |
| WaNet | 94.2/**94.2** | 11.8/**0.9** | 82.7/**93.0** | 94.2/**94.3** | 1.5/**0.9** | 92.6/**92.9** | **94.2**/93.5 | 90.0/**0.9** | 9.9/**92.3** |
| Input-aware | 94.5/**94.5** | **1.0**/1.8 | **92.0**/90.9 | 94.6/**94.8** | 80.5/**1.9** | 18.6/**91.5** | 93.5/**94.5** | 82.1/**1.2** | 17.2/**91.1** |
| BppAttack | **94.6**/94.4 | 30.3/**11.7** | 73.9/**90.2** | 94.4/**94.4** | 13.0/**11.7** | 88.8/**90.2** | 94.0/**94.0** | 99.9/**10.4** | 10.1/**90.4** |
| Average | +0.1 | −10.4 | +8.5 | +0.5 | −34.0 | +29.5 | +0.2 | −93.8 | +84.0 |

Table 13: Comparison of achievable TPR (%) for different defense methods on CIFAR-10 with $FPR \leq 10\%$. For SCAn/+SAM, we show the performance of the base detector and our SAM-enhanced version.

| Attack ↓ | Scale-up TPR ↑ | CBD TPR ↑ | SCAn/+SAM TPR ↑ |
|---|---|---|---|
| BadNets | 100.00 | 100.00 | 96.01 / **95.21** |
| Blended | 12.01 | 91.21 | 99.22 / **98.73** |
| WaNet | 16.72 | 78.23 | 66.31 / **90.12** |

alarms are highly undesirable, we assess the achievable True Positive Rate (TPR) while constraining the False Positive Rate (FPR) to be no more than 10%. As presented in Tab. 13, the results show that while training-free methods are effective against some attacks, their TPR can be inconsistent under this practical constraint, especially for more sophisticated attacks like Blended and WaNet. In contrast, our SAM-enhanced method demonstrates consistently superior and more stable detection performance across all scenarios.

Ultimately, the key advantage of our pre-training approach is its ability to enable the training of a final model that is both secure and high-performing. As demonstrated in Sec. E.6, retraining a model on the dataset purified by our SAM-enhanced method not only maintains high clean accuracy (ACC) but also significantly reduces the attack success rate (ASR). This confirms the practical, end-to-end effectiveness of our framework in producing a robust final model, a benefit not offered by inference-time defenses.

# D ADDITIONAL ANALYSIS OF SAM-ENHANCED-PSD

## D.1 DETECTION PERFORMANCE WITH DIFFERENT $\rho$

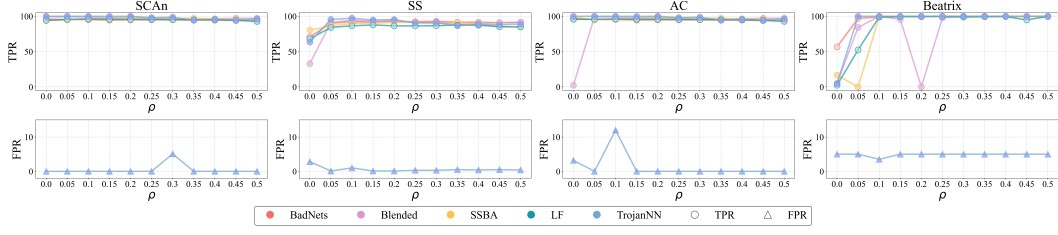

Figure 7: Detection performance of base PSD with SAM-enhanced PSD with different $\rho$ on CIFAR10 and ResNet18.

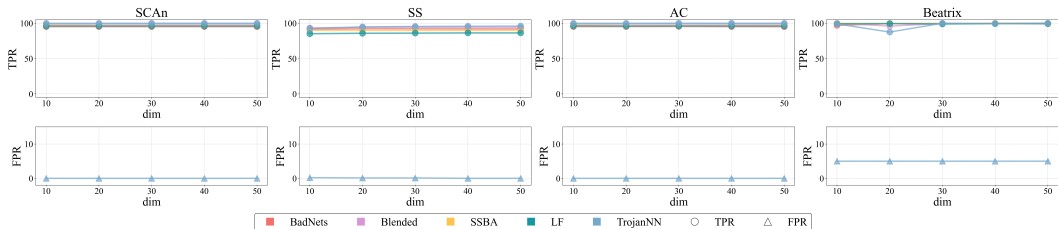

Figure 8: Detection performance of base PSD with SAM-enhanced PSD with different dimensions on CIFAR10 and ResNet18

The constraint bound $\rho$ is a key hyperparameter in our detection strategy, as it governs the extent of perturbation $\epsilon$. The excessively small $\rho$ may result in a weak enhancement of the backdoor effect, whereas an excessively large $\rho$ can degrade the model's performance by disrupting the extraction of information, such as features, from both poisoned and clean samples. We assess the sensitivity of $\rho$ by executing five complex attacks and employing four detection methods. Figure 7 illustrates the results of these detections combined with SAM-enhanced PSD. While a smaller $\rho$ may not fully amplify backdoor effects, resulting in poor performance of Beatrix against SSBA and TrojanNN attacks, it demonstrates that SAM-enhanced PSD can successfully identify poisoned samples and maintain a reasonable false positive rate across different $\rho$ settings. Overall, $\rho$ proves to be relatively insensitive; a broad range of values can be selected with good detection performance.

### D.2 DETECTION PERFORMANCE ON DIFFERENT PARAMETERS OF EXTRACTING BACKDOOR-RELATED FEATURE

In this study, we analyze two crucial parameters in the extracting backdoor-related feature stage of our proposed defense mechanism, SAM-enhanced PSD: dimension and the choice of surrogate dataset.

#### D.2.1 IMPACT OF FEATURE DIMENSION

The dimension of the feature space plays a pivotal role in the effectiveness of backdoor detection methods. A dimension that is too large complicates the estimation of the covariance matrix, while a dimension that is too small may result in excessive removal of the backdoor signal. Through empirical testing, we evaluate the impact of varying the dimension from 10 to 50 on the performance of SAM-enhanced PSD across different attacks and detection methods. As shown in Fig. 8, our results indicate that the performance remains relatively stable across this range for most scenarios. However, we observe some variability in performance at a dimension of 20 for the Beatrix method. Based on these findings, we select a dimension of 30 for our main experiments, balancing the trade-offs between complexity and signal retention effectively.

#### D.2.2 IMPACT OF THE SURROGATE DATASET

To further investigate the practical applicability and robustness of our method, we evaluate its performance under various conditions for the clean auxiliary set. The flexibility of a defense method regarding its auxiliary set is critical, especially in real-world scenarios where a large, verified clean dataset may not be available. Therefore, we systematically designed three parallel experimental scenarios: (1) **Limited in-distribution data**, where we investigate the method's sensitivity to the quantity of clean samples by varying the auxiliary set size from as few as 25 per class (a 0.5% clean ratio) to 250 per class (a 5% clean ratio); (2) **Sifted-clean (S-Clean) Data**, where we test the method's reliance on the purity of the auxiliary set by simulating a case where defenders identify a reference set from the poisoned dataset itself using an existing technique (META-SIFT (Zeng et al., 2023)) to sift out 250 samples per class; and (3) **Out-of-distribution (OOD) data**, where we assess the method's ability to leverage OOD data by using 250 samples per class from the CIFAR-5m (Nakkiran et al., 2020) dataset as the auxiliary set for the CIFAR-10 task.

In the latter two scenarios, we use a fixed quantity of 250 samples per class. This is because the primary challenge for these data sources lies in their purity or distribution mismatch, not their

Table 14: Detection comparisons (measured by TPR (%), FPR (%) and F1 (%)) between base PSD and SAM-enhanced PSD (+SAM optimized with ASAM) on CIFAR-10 and ResNet18, and the better result in each pair is highlighted in **bold**. In terms of each metric, the average change of SAM-enhanced PSD to base PSD across all attacks is presented at the bottom: performance improvements are highlighted in green, other changes in red.

| Detection → | Spectre / +SAM | | | SCAn / +SAM | | | SS / +SAM | | | AC / +SAM | | | Beatrix / +SAM | | |
|---|---|---|---|---|---|---|---|---|---|---|---|---|---|---|---|
| Attack ↓ | TPR↑ | FPR↓ | F1↑ | TPR↑ | FPR↓ | F1↑ | TPR↑ | FPR↓ | F1↑ | TPR↑ | FPR↓ | F1↑ | TPR↑ | FPR↓ | F1↑ |
| BadNets | 51.1/**84.4** | 4.9/**4.4** | 42.0/**62.9** | **96.0**/95.0 | 0.0/0.0 | **98.0**/97.5 | 70.8/**91.7** | 2.4/**0.6** | 65.6/**95.7** | **96.8**/95.1 | **0.1**/13.8 | **97.1**/41.5 | 56.6/**92.6** | 5.0/**0.4** | 44.9/**92.6** |
| Blended | 29.9/**60.6** | 6.0/**0.0** | 24.6/**75.5** | **99.2**/98.8 | **0.0**/3.8 | **99.6**/73.1 | 32.9/**100.0** | **4.4**/6.1 | 30.5/**63.2** | 2.3/**87.3** | **7.7**/11.7 | 1.9/**42.7** | 5.0/**99.4** | 5.0/**1.3** | 5.0/**89.0** |
| SSBA | 36.6/**70.8** | **5.6**/7.5 | 30.1/**45.2** | 93.9/**98.2** | 0.0/0.8 | 96.9/**99.1** | 80.4/**84.9** | **1.9**/5.9 | **74.5**/57.2 | 99.3/**100.0** | **3.3**/20.8 | **76.1**/33.6 | 16.8/**98.1** | 5.0/**0.0** | 15.8/**99.0** |
| LF | 32.0/**57.4** | **5.9**/6.1 | 26.3/**41.9** | 94.1/**99.6** | 0.0/0.7 | **97.0**/93.4 | 68.2/**86.1** | **2.5**/4.8 | **63.2**/62.3 | 95.6/**100.0** | 10.4/**7.7** | 48.7/**57.9** | 2.4/**100.0** | 5.0/**0.7** | 2.4/**93.9** |
| Adap-Blend | 24.1/**64.5** | **5.6**/7.4 | 20.9/**42.4** | 92.5/**93.7** | 10.5/**2.4** | 47.3/**78.2** | 20.2/**85.0** | 4.5/**1.4** | 19.6/**80.1** | 1.5/**95.4** | 7.1/**3.1** | 1.2/**74.8** | 6.2/**100.0** | **5.0**/7.8 | 6.2/**57.5** |
| LC | 17.0/**33.9** | **4.3**/7.5 | 17.1/**24.5** | 100.0/100.0 | 0.0/0.0 | 100.0/100.0 | 40.5/**41.4** | 2.1/**0.9** | 45.0/**51.9** | 0.0/**100.0** | 0.0/0.0 | 0.0/**100.0** | 2.2/**100.0** | 5.0/**2.8** | 2.2/**78.9** |
| TaCT | 36.1/**82.7** | **7.0**/13.6 | 26.9/**37.6** | 100.0/100.0 | 0.0/0.0 | 100.0/100.0 | **42.3**/40.9 | **4.2**/10.0 | **38.1**/24.7 | 100.0/100.0 | 0.1/**0.0** | 99.5/**100.0** | 13.4/**100.0** | 5.0/**2.5** | 12.9/**81.0** |
| TrojanNN | 30.2/**51.0** | 6.0/**4.7** | 24.8/**42.5** | **100.0**/97.0 | 0.0/0.0 | **100.0**/98.5 | 63.4/**99.5** | 2.8/**1.6** | 58.7/**86.9** | 99.9/**100.0** | **3.2**/15.3 | **76.7**/40.7 | 4.6/**100.0** | 5.0/**3.1** | 4.6/**77.1** |
| WaNet | 66.4/**100.0** | 4.1/**0.0** | 54.3/**100.0** | 66.3/**88.2** | **0.0**/0.3 | 79.7/**90.7** | 71.1/**87.4** | 1.5/**0.0** | 71.4/**93.2** | 85.1/**96.1** | 0.0/0.0 | 91.9/**98.0** | 1.2/**96.6** | 5.0/5.0 | 1.2/**66.2** |
| Input-aware | 53.9/**100.0** | 4.7/**0.0** | 44.2/**100.0** | 97.4/**100.0** | **0.1**/4.2 | **97.6**/71.2 | **83.5**/81.8 | 0.9/**0.0** | 83.5/**90.0** | 0.0/**88.3** | 0.0/0.0 | 0.0/**93.8** | 3.4/**100.0** | 5.0/5.0 | 3.4/**67.8** |
| BppAttack | 21.5/**28.6** | **6.3**/5.3 | 17.8/**25.0** | 87.8/**100.0** | **0.0**/2.8 | **93.5**/78.9 | 85.8/**96.7** | 0.8/**0.0** | 85.7/**98.3** | 94.2/**96.7** | 0.0/0.0 | 97.0/**98.3** | 0.1/**100.0** | 5.0/5.0 | 0.1/**67.8** |
| SIBA | 30.0/**64.6** | 6.0/**1.3** | 24.6/**68.4** | **98.7**/93.6 | **0.0**/3.9 | **99.3**/70.1 | 72.9/**88.8** | **1.2**/6.1 | **74.2**/58.4 | 96.8/**100.0** | 0.0/0.0 | 98.4/**100.0** | 4.3/**92.4** | 5.0/5.0 | 4.3/**64.3** |
| BadNets-A2A | 99.5/**100.0** | 10.6/**10.0** | 49.7/**51.3** | 0.0/0.0 | 0.0/0.0 | 0.0/0.0 | **99.4**/96.2 | 10.6/**7.8** | 49.7/**55.9** | 97.8/90.3 | 0.0/0.0 | 98.9/94.9 | 27.3/**99.8** | 5.0/**0.3** | 24.6/**97.0** |
| Average | +28.5 | −0.7 | +24.2 | +2.9 | +0.7 | −4.5 | +19.1 | +0.3 | +12.2 | +29.2 | +3.1 | +14.5 | +87.4 | −2.0 | +69.6 |

Table 15: Detection comparisons (measured by TPR (%), FPR (%) and F1 (%)) between base PSD and SAM-enhanced PSD (+SAM optimized with SAM-ON) on CIFAR-10 and ResNet18, and the better result in each pair is highlighted in **bold**. In terms of each metric, the average change of SAM-enhanced PSD to base PSD across all attacks is presented at the bottom: performance improvements are highlighted in green, other changes in red.

| Detection → | Spectre / +SAM | | | SCAn / +SAM | | | SS / +SAM | | | AC / +SAM | | | Beatrix / +SAM | | |
|---|---|---|---|---|---|---|---|---|---|---|---|---|---|---|---|
| Attack ↓ | TPR↑ | FPR↓ | F1↑ | TPR↑ | FPR↓ | F1↑ | TPR↑ | FPR↓ | F1↑ | TPR↑ | FPR↓ | F1↑ | TPR↑ | FPR↓ | F1↑ |
| BadNets | 51.1/**77.6** | 4.9/**1.4** | 42.0/**76.2** | **96.0**/95.8 | 0.0/0.0 | **98.0**/97.9 | 70.8/**89.4** | 2.4/**0.6** | 65.6/**88.8** | 96.8/**98.9** | **0.1**/8.8 | **97.1**/54.1 | 56.6/**100.0** | 5.0/**0.2** | 44.9/**98.4** |
| Blended | 29.9/**56.8** | **6.0**/7.7 | 24.6/**37.4** | **99.2**/98.3 | 0.0/0.0 | **99.6**/99.2 | 32.9/**93.3** | 4.4/**0.0** | 30.5/**96.6** | 2.3/**100.0** | **7.7**/14.0 | 1.9/**43.0** | 5.0/**100.0** | 5.0/**1.2** | 5.0/**90.0** |
| SSBA | 36.6/**78.8** | 5.6/**2.3** | 30.1/**71.0** | 93.9/**96.7** | 0.0/0.0 | 96.9/**98.3** | 80.4/**88.9** | **1.9**/4.1 | **74.5**/66.5 | 99.3/97.8 | **3.3**/13.3 | **76.1**/43.4 | 16.8/**97.0** | 5.0/**0.3** | 15.8/**95.6** |
| LF | 32.0/**51.9** | 5.9/**8.2** | 26.3/**33.8** | 94.1/**100.0** | 0.0/0.0 | 97.0/**100.0** | 68.2/**83.7** | **2.5**/3.6 | 63.2/**65.4** | 95.6/**97.0** | 10.4/15.0 | 48.7/**40.2** | 2.4/**92.2** | 5.0/**0.8** | 2.4/**91.7** |
| Adap-Blend | 24.1/**67.3** | 5.6/**2.4** | 20.9/**63.4** | 92.5/**100.0** | 10.5/**7.0** | 47.3/**60.1** | 20.2/**91.5** | 4.5/**3.6** | 19.6/**70.4** | 1.5/**98.2** | 7.1/7.5 | 1.2/**57.5** | 6.2/**100.0** | **5.0**/7.9 | 6.2/**57.1** |
| LC | 17.0/**40.7** | **4.3**/5.5 | 17.1/**33.2** | 100.0/96.0 | 0.0/0.0 | 100.0/98.0 | 40.5/**44.0** | **2.1**/5.4 | **45.0**/35.6 | 0.0/**100.0** | 0.0/0.9 | 0.0/**91.9** | 2.2/**100.0** | 5.0/**2.9** | 2.2/**78.6** |
| TaCT | 36.1/**77.5** | 7.0/**4.7** | 26.9/**58.3** | 100.0/100.0 | 0.0/0.0 | 100.0/100.0 | 42.3/**46.8** | 4.2/4.7 | **38.1**/39.5 | 100.0/96.8 | 0.1/**0.5** | 99.5/93.7 | 13.4/**100.0** | 5.0/**2.7** | 12.9/**84.2** |
| TrojanNN | 30.2/**61.7** | 6.0/**0.0** | 24.8/**76.3** | **100.0**/93.0 | 0.0/0.0 | **100.0**/96.4 | 63.4/**100.0** | 2.8/**4.2** | 58.7/**71.6** | 99.9/99.1 | **3.2**/14.8 | **76.7**/41.3 | 4.6/**93.7** | 5.0/**3.3** | 4.6/**73.0** |
| WaNet | 66.4/**95.1** | 4.1/**0.0** | 54.3/**97.5** | 66.3/**100.0** | **0.0**/1.9 | 79.7/**85.0** | 71.1/**93.7** | **1.5**/7.3 | 71.4/56.4 | 85.1/85.1 | 0.0/3.2 | 91.9/69.2 | 1.2/**100.0** | 5.0/5.0 | 1.2/**67.8** |
| Input-aware | 53.9/**97.2** | **4.7**/7.8 | 44.2/**56.4** | 97.4/91.6 | 0.1/**0.0** | 97.6/95.6 | **83.5**/79.4 | 0.9/**0.0** | 83.5/**88.5** | 0.0/**89.6** | 0.0/5.1 | 0.0/**62.7** | 3.4/**97.0** | 5.0/5.0 | 3.4/**66.4** |
| BppAttack | 21.5/**44.7** | **6.3**/6.9 | 17.8/**32.5** | 87.8/**93.4** | 0.0/0.2 | 93.5/**95.1** | 85.8/**89.5** | 0.8/**0.5** | 85.7/**90.1** | 94.2/**97.3** | 0.0/0.0 | 97.0/**98.6** | 0.1/**91.7** | 5.0/5.0 | 0.1/**63.9** |
| SIBA | 30.0/**62.4** | 6.0/**1.9** | 24.6/**62.8** | **98.7**/90.9 | 0.0/0.0 | **99.3**/95.2 | 72.9/**86.8** | **1.2**/6.3 | **74.2**/56.5 | 96.8/93.8 | 0.0/0.0 | 98.4/96.8 | 4.3/**86.4** | 5.0/5.0 | 4.3/**61.4** |
| BadNets-A2A | 99.5/**100.0** | 10.6/15.2 | **49.7**/40.8 | 0.0/0.0 | 0.0/0.0 | 0.0/0.0 | **99.4**/96.6 | 10.6/16.0 | **49.7**/38.6 | 97.8/89.3 | 0.0/2.1 | 98.9/78.2 | 27.3/**100.0** | 5.0/**0.9** | 24.6/**92.2** |
| Average | +29.5 | −1.0 | +25.9 | +2.3 | -0.1 | +0.9 | +19.4 | +1.2 | +8.1 | +28.7 | +4.1 | +6.4 | +85.8 | −2.0 | +68.8 |

availability, as they are generally easier to acquire than verified in-distribution clean data. Thus, our goal here is to validate their *viability* as substitutes for a standard clean set.

As shown in Fig. 9, our method demonstrates remarkable robustness across all three challenging scenarios. Detection performance remains high even when the quantity of clean samples is severely limited (Scenario 1). Critically, our method is equally effective when using the S-Clean set (Scenario 2) or data from a completely different distribution (Scenario 3). These findings collectively confirm that SAM-enhanced PSD has a low dependency on the clean auxiliary set, highlighting its flexibility and practical value for real-world deployment.

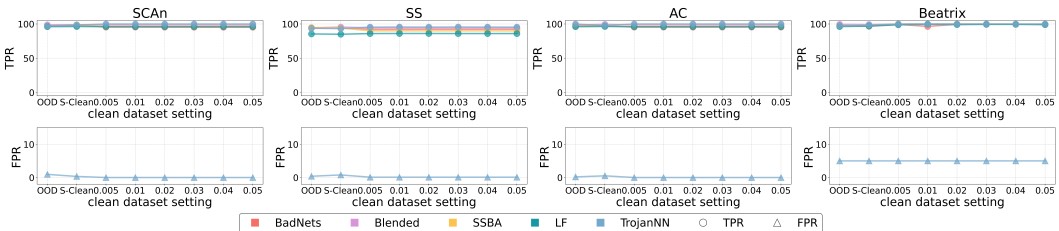

Figure 9: Detection performance of SAM-enhanced PSD with different clean auxiliary sets on CIFAR10 and ResNet18. The x-axis compares results using three types of auxiliary data: an Out-of-Distribution (OOD) set, an S-Clean set, and varying ratios of true in-distribution clean data. The S-Clean set refers to data sifted from the poisoned dataset.

### D.3 DETECTION PERFORMANCE WITH DIFFERENT SAM VARIANTS

To examine how alternative sharpness-aware optimizers influence SAM-enhanced-PSD, we replace the original SAM with three popular variants, ASAM (Kwon et al., 2021), SAM-ON (Mueller et al.,

Table 16: Detection comparisons (measured by TPR (%), FPR (%) and F1 (%)) between base PSD and SAM-enhanced PSD (+SAM optimized with MSAM) on CIFAR-10 and ResNet18, and the better result in each pair is highlighted in **bold**. In terms of each metric, the average change of SAM-enhanced PSD to base PSD across all attacks is presented at the bottom: performance improvements are highlighted in green, other changes in red.

| Detection → | Spectre / +SAM | | | SCAn / +SAM | | | SS / +SAM | | | AC / +SAM | | | Beatrix / +SAM | | |
|---|---|---|---|---|---|---|---|---|---|---|---|---|---|---|---|
| Attack ↓ | TPR ↑ | FPR ↓ | F1 ↑ | TPR ↑ | FPR ↓ | F1 ↑ | TPR ↑ | FPR ↓ | F1 ↑ | TPR ↑ | FPR ↓ | F1 ↑ | TPR ↑ | FPR ↓ | F1 ↑ |
| BadNets | 51.1/**86.7** | **4.9**/9.6 | 42.0/**47.0** | **96.0**/95.4 | 0.0/0.0 | **98.0**/97.6 | 70.8/**91.9** | 2.4/**0.0** | 65.6/**95.8** | **96.8**/96.3 | **0.1**/9.9 | **97.1**/50.1 | 56.6/**100.0** | 5.0/**0.5** | 44.9/**95.4** |
| Blended | 29.9/**58.4** | **6.0**/8.3 | 24.6/**37.0** | **99.2**/98.6 | **0.0**/4.3 | **99.6**/70.3 | 32.9/**94.9** | 4.4/**0.0** | 30.5/**97.4** | 2.3/**100.0** | **7.7**/11.9 | 1.9/**46.9** | 5.0/**93.2** | 5.0/**1.1** | 5.0/**86.9** |
| SSBA | 36.6/**68.4** | 5.6/**4.0** | 30.1/**56.0** | 93.9/**96.7** | **0.0**/0.5 | **96.9**/94.1 | 80.4/**94.0** | 1.9/**0.0** | 74.5/**96.9** | 99.3/**100.0** | **3.3**/14.3 | **76.1**/42.4 | 16.8/**98.0** | 5.0/**0.4** | 15.8/**95.1** |
| LF | 32.0/**53.0** | 5.9/**0.0** | 26.3/**69.3** | 94.1/**97.8** | 0.0/0.0 | 97.0/**98.9** | 68.2/**91.2** | 2.5/**0.0** | 63.2/**95.4** | 95.6/**100.0** | 10.4/**10.2** | 48.7/**50.7** | 2.4/**98.9** | 5.0/**0.2** | 2.4/**97.7** |
| Adap-Blend | 24.1/**60.1** | 5.6/**3.8** | 20.9/**51.6** | 92.5/**100.0** | 10.5/**9.1** | 47.3/**53.5** | 20.2/**91.4** | 4.5/**0.0** | 19.6/**95.5** | 6.2/**100.0** | 7.1/**6.1** | 1.2/**63.1** | 6.2/**100.0** | **5.0**/7.8 | 6.2/**57.5** |
| LC | 17.0/**41.8** | 4.3/**3.1** | 17.1/**41.7** | **100.0**/95.3 | **0.0**/1.8 | **100.0**/82.8 | 40.5/**50.7** | 2.1/**0.0** | 45.0/**67.2** | 0.0/**99.3** | 0.0/0.0 | 0.0/**99.6** | 2.2/**100.0** | 5.0/**3.1** | 2.2/**77.3** |
| TaCT | 36.1/**80.9** | **7.0**/10.2 | 26.9/**43.1** | **100.0**/95.4 | 0.0/0.0 | **100.0**/97.6 | 42.3/**49.3** | 4.2/**2.5** | 38.1/**50.0** | **100.0**/93.4 | 0.1/**0.0** | 99.5/**96.6** | 13.4/**91.6** | 5.0/**2.3** | 12.9/**78.1** |
| TrojanNN | 30.2/**61.6** | **6.0**/13.7 | 24.8/**29.2** | 100.0/100.0 | 0.0/0.0 | 100.0/100.0 | 63.4/**97.3** | 2.8/**0.0** | 58.7/**98.6** | **99.9**/96.9 | **3.2**/22.2 | **76.7**/31.3 | 4.6/**100.0** | 5.0/**3.0** | 4.6/**78.1** |
| WaNet | 66.4/**98.8** | 4.1/**0.0** | 54.3/**99.4** | 66.3/**87.8** | 0.0/0.0 | 79.7/**93.5** | 71.1/**93.9** | 1.5/**0.0** | 71.4/**96.9** | 85.1/**94.2** | **0.0**/2.7 | **91.9**/76.9 | 1.2/**93.9** | 5.0/5.0 | 1.2/**65.0** |
| Input-aware | 53.9/**100.0** | 4.7/**2.2** | 44.2/**82.9** | 97.4/**100.0** | 0.1/**0.0** | 97.6/**99.8** | **83.5**/81.9 | 0.9/**0.0** | 83.5/**90.1** | 0.0/**92.0** | **0.0**/4.1 | 0.0/**68.3** | 3.4/**100.0** | 5.0/5.0 | 3.4/**67.8** |
| BppAttack | 21.5/**41.3** | 6.3/**0.3** | 17.8/**56.1** | 87.8/**99.0** | **0.0**/4.4 | **93.5**/70.1 | 85.8/**93.3** | 0.8/**0.0** | 85.7/**96.6** | 94.2/**98.4** | 0.0/0.0 | 97.0/**99.2** | 0.1/**96.9** | 5.0/5.0 | 0.1/**66.4** |
| SIBA | 30.0/**68.5** | 6.0/**4.7** | 24.6/**53.3** | **98.7**/94.9 | 0.0/0.0 | **99.3**/97.4 | 72.9/**89.3** | **1.2**/0.8 | 74.2/**87.2** | 96.8/**100.0** | **0.0**/0.5 | 98.4/**95.8** | 4.3/**90.1** | 5.0/5.0 | 4.3/**63.1** |
| BadNets-A2A | 99.5/**100.0** | 10.6/**6.3** | 49.7/**62.4** | 0.0/0.0 | 0.0/0.0 | 0.0/0.0 | 99.4/**100.0** | 10.6/**9.4** | 49.7/**52.9** | 97.8/**98.2** | 0.0/0.0 | 98.9/**99.1** | 27.3/**100.0** | 5.0/**0.1** | 24.6/**99.1** |
| Average | +30.1 | −0.8 | +25.1 | +2.7 | +0.7 | −4.1 | +22.1 | −2.1 | +27.8 | +30.7 | +3.8 | +10.2 | +86.1 | −2.0 | +69.2 |

2023), and MSAM (Becker et al., 2024), and evaluate them on CIFAR-10 with ResNet-18. Following common practice, we set $\rho = 1.0$ for ASAM, $\rho = 0.5$ for SAMON, and $\rho = 0.1$ for MSAM. The results ( Tab. 14, Tab. 15 and Tab. 16) reveal a consistent pattern: regardless of the optimizer variant, the SAM-enhanced-PSD markedly outperforms their vanilla PSD baselines, delivering substantial gains in true-positive rate while maintaining a low false-positive rate across all tested backdoor attacks. This robustness is in line with our theoretical insight in Proposition 1, which shows that the additional parameter step introduced by the SAM update amplifies backdoor-relevant neurons, thereby enlarging the separation between poisoned and clean samples; because all SAM variants share the extra ascent–descent mechanism, SAM-enhanced-PSD retains the same advantage and therefore remains broadly effective even when the defender replaces one variant with another.

## D.4 COMPUTATIONAL ANALYSIS

To thoroughly analyze the computational complexity of PSD and SAM-enhanced-PSD, we divide the process into three distinct stages: training, feature extraction, and detection. The training stage is widely adopted by most existing PSD methods, typically using conventional SGD optimization. In contrast, SAM-enhanced-PSD replaces SGD with the original SAM optimizer, introducing an additional ascent-descent step per mini-batch, which approximately doubles the computational complexity compared to standard SGD. The feature extraction stage is exclusive to SAM-enhanced-PSD, specifically designed to capture backdoor-related features through a single PCA operation and variance estimation. The detection stage remains identical for both PSD and SAM-enhanced-PSD, contributing minimally to overall computational costs.

We estimate the algorithmic complexities for both baseline PSD and SAM-enhanced-PSD across these three stages and measure their actual runtimes on an NVIDIA GeForce RTX 4090 GPU, as shown in Tab. 17. Our measurements confirm that the model training stage dominates the runtime, while the SAM-enhanced-PSD's feature extraction stage is shorter than the detection stage. Although SAM approximately doubles the training overhead, our theoretical analysis in Proposition 1 guarantees compatibility with various efficient SAM variants. Specifically, we test MSAM (Becker et al., 2024), an efficient variant of SAM, in Tab. 16, demonstrating that MSAM significantly enhances detection performance and can seamlessly replace the original SAM in SAM-enhanced-PSD. Furthermore, Tab. 17 illustrates that MSAM's computational complexity and runtime closely approximate those of the conventional SGD baseline.

In summary, the primary computational cost of SAM-enhanced-PSD lies within its training stage due to the optimization method used. However, employing a computationally efficient SAM variant such as MSAM mitigates this overhead while maintaining robust backdoor detection performance.

## D.5 STATISTICAL ANALYSIS OF TAC VALUES

To validate that our reported average Top-k TAC is a robust metric, we conduct a deeper statistical analysis of its properties. A potential concern is that a high average TAC value could be skewed by a few samples exhibiting extreme activation differences, masking the true distribution across the entire

Table 17: Computation complexity and time of PSD and SAM-enhanced-PSD on CIFAR-10. General setting: Epoch $E$; Samples $N$; Perturbation samples $N_p$; Feature Dimension $D$; Class $K$; Forward $F$; Backward $B$

| 1. Training | | | 2. Extract Features | 3. Detection |
|---|---|---|---|---|
| Conventional SGD | SAM | Momentum-SAM | | |
| $O((N/B)E(f + b + p))$ | $O(2 * (N/B)E(f + b + 3p))$ | $O((N/B)E(f + b + 4p))$ | $O(NKd^2)$ | Varies by method |
| 1165s | 2266s | 1253s | 26s | 124s (average) |

dataset. To rule out such statistical artifacts, we directly analyze the distribution of the **per-sample activation differences** that constitute the final TAC value.

**Experimental setup and conclusion** For the Top-5 (most responsive) and Bottom-5 (least responsive) neurons under three different attacks, we calculated the activation difference ($||f_j(x) - f_j(\tilde{x})||_2$ from Eq. (1)) for each clean sample $x$ and its poisoned counterpart $\tilde{x}$. We then computed the mean and standard deviation of these per-sample difference values.

The results, presented in Tab. 18, show a clear dichotomy. The Top-5 neurons exhibit consistently high mean activation differences with low standard deviations relative to their means. For instance, under the Blended attack, the Top-1 neuron has a mean activation difference of 22.55 versus a standard deviation of only 1.82. This demonstrates that the backdoor induces a stable and strong activation change across the vast majority of samples. In contrast, the Bottom-5 neurons consistently show negligible activation differences. This analysis confirms that our reported average TAC is driven by a robust and consistent effect across the sample population, rather than being a statistical artifact caused by a few outliers, thus validating its reliability as a metric for a backdoor effect.

Table 18: Statistical analysis of per-sample activation differences for the top-5 and bottom-5 neurons under different attacks. The values are presented in Mean (Standard Deviation) format.

| Attack | Top 1 | Top 2 | Top 3 | Top 4 | Top 5 | Bottom 1 | Bottom 2 | Bottom 3 | Bottom 4 | Bottom 5 |
|---|---|---|---|---|---|---|---|---|---|---|
| Blended | 22.55 (1.82) | 11.44 (1.48) | 11.34 (1.32) | 10.80 (0.84) | 10.27 (1.16) | 1.18 (0.34) | 1.04 (0.20) | 0.96 (0.20) | 0.93 (0.00) | 0.90 (0.00) |
| SSBA | 26.26 (2.28) | 15.01 (1.32) | 11.52 (1.00) | 10.83 (1.31) | 10.23 (1.35) | 1.18 (0.28) | 1.16 (0.28) | 1.15 (0.20) | 1.14 (0.28) | 0.94 (0.20) |
| LF | 20.27 (1.66) | 16.66 (1.35) | 12.93 (1.07) | 11.37 (0.93) | 10.23 (1.16) | 1.45 (0.34) | 1.44 (0.40) | 1.35 (0.20) | 1.11 (0.28) | 0.97 (0.00) |

## D.6 GENERALIZATION OF THE TAC-DETECTION CORRELATION

To further validate the generalizability of the positive correlation we identified between Trigger Activation Change (TAC) and backdoor detection performance, we extended our evaluation to additional datasets and model architectures.

**Experimental setup and analysis** We designed two additional experimental settings to verify the consistency of our findings: the first on the GTSRB dataset with a ResNet-18 model, and the second on the CIFAR-10 dataset with a VGG19-BN model. In this expanded analysis, we report not only the TAC values and detection performance (AUROC) but also the Pearson correlation coefficient and the linear regression coefficient ($R^2$) to quantify the strength and consistency of the relationship.

As illustrated in Fig. 10, the results from our expanded analysis demonstrate a consistently positive correlation between TAC and detection performance across all tested configurations. The high values for both the Pearson correlation and $R^2$ coefficients shown in the figure confirm that the strong relationship between TAC and detection efficacy is not a statistical artifact but a robust outcome that holds across different datasets and models. These findings strengthen our paper's core claim that amplifying the backdoor effect, as measured by TAC, is an effective strategy for improving poisoned sample detection.

## D.7 T-SNE VISUALIZATION

We have provided the t-SNE visualization results, as shown in Fig. 11. The first row displays the feature distribution after training with vanilla training, while the second row shows the distribution after training with SAM. It is evident that the poisoned samples are more distant from the target

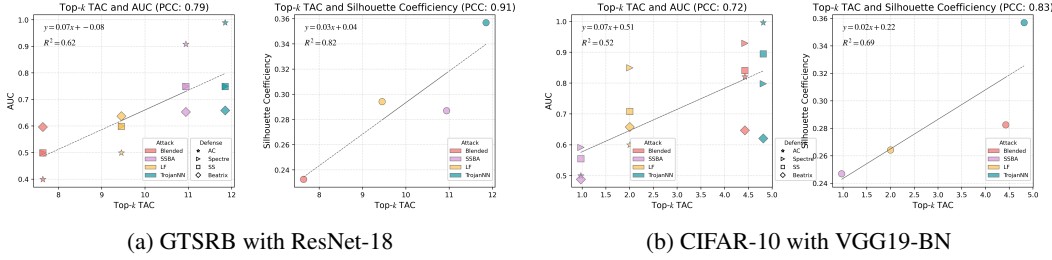

(a) GTSRB with ResNet-18             (b) CIFAR-10 with VGG19-BN

Figure 10: Validation of the correlation between Top-k TAC, detection performance (AUC), and feature separability (Silhouette Score) on new experimental settings. The left image (a) displays the correlation plots for GTSRB with ResNet-18, while the right image (b) displays the plots for CIFAR-10 with VGG19-BN.

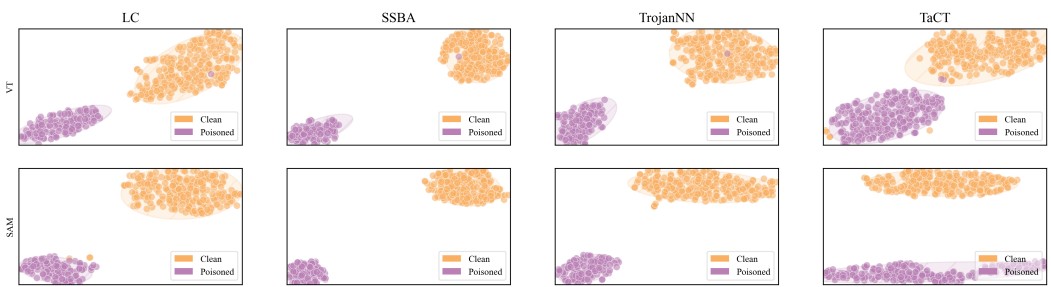

Figure 11: T-SNE visualization under different backdoor attacks on CIFAR10 and ResNet18 with different training algorithms. The first row represents the model using vanilla training, and the second row represents the model using SAM.

clean samples when trained with SAM. This indicates that SAM enhances the separability between poisoned and target clean samples, thereby improving the detection performance.

## E  THE USE OF LARGE LANGUAGE MODELS

In the preparation of this manuscript, we utilize a large language model (LLM) to enhance the quality and clarity of the text. The primary application of the LLM involves grammatical correction and language polishing. We employ this assistive tool to identify and rectify syntactical errors, improve sentence structure, and ensure the consistent use of academic terminology. The core intellectual content, including all research, analysis, and conclusions, is generated entirely by the authors. The role of the LLM is strictly limited to that of a writing aid to improve the readability and professionalism of the prose.

