# OpenReview forum: "Reliable Poisoned Sample Detection against Backdoor Attacks Enhanced by Sharpness Aware Minimization"
_ICLR.cc/2026/Conference — ICLR 2026 Poster_

### Official Review · Reviewer_GmvL · 2025-10-25

**Soundness:** 3
**Presentation:** 3
**Contribution:** 2
**Rating:** 6
**Confidence:** 5

**Summary:**

This work investigates Poisoned Sample Detection (PSD), a promising defense approach against backdoor attacks. The investigation results show that PSD perform poorly against weak backdoor attacks. Some experiments are conducted to validate the observation. To resolve this problem, the authors propose SAM, a method to enhance the backdoor attack effect to make it easier to be detected by PSD defenses.Combined with PSD and SAM, the newly proposed defense is claimed to be more effective in a wider range of attack scenarios.

**Strengths:**

1. Investigate the weakness of current PSD-based backdoor defense via theoretical analysis and some experiments;
2. Propose a method, called Sharpness-Aware Minimization (SAM), to enhances the activations of top trigger activation change neurons during backdoor attacks.
3. By combining SAM and PSD, propose a new defense that outperforms existing PSD methods.

**Weaknesses:**

1. Directly applying SAM to PSD limits the contribution of this work.
2. Some experimental results are repeatedly shown, e.g., Fig. 2 (right) and Fig 5. Moving the repeated part to the appendix and add more
3. Some experimental results looks like a little strange.
4. Maybe suffer a low performance over hard-to-learn datasets.

**Questions:**

1. From this paper, SAM looks to be a 'perfect' amplification for sparse neuron activations and can be combined with all kinds of previous PSD methods. While, I know SAM is good, but presenting its limitation and explaining why it does not matter to the defense can make us understand more about the newly proposed defense.
2. As shown in Fig.5, from the visualization results in the left, the two clusters are almost separated. However, the right part shows that the poisoned samples are totally mixed together with the clean data. This is very strange, and impossible for most cases. I'm wondering why.
3. Another question is about the FPR. A very lot of experiments are conducted by the authors. In almost every experiment, the reported FPR is affected very slightly by the added SAM mechanism. This phenomenon violates the assumption of SAM. That is, if SAM truely and only amplify the difference between poisoned and clean data like what has been shown in this paper, the indicator of FPR should also be improved but not only the TPR. Maybe, the reason is that the SAM also amplified the difference between some 'rarely-occurred' clean data with normal ones. Therefore, the proposed defense may suffer a low performance on some hard-to-learn dataset. I suggest the author to clarify this phenomenon.

---

> ### Author Response · Authors · 2025-11-21
> **Response(Part 1)**
>
> We sincerely thank the reviewer for the time and effort spent evaluating our work and for the **encouraging** comments. We appreciate that the reviewer found our study of PSD’s weaknesses under weak backdoor attacks **meaningful**, regarded our use of SAM to enhance trigger-activation signals as a **principled** way to strengthen PSD, and acknowledged that the resulting SAM-enhanced PSD defense is **more effective** and **robust** than existing PSD methods across a broader range of attack scenarios.
>
> ---
>
> **W1. Concern that directly applying SAM to PSD limits the contribution of this work**
>
> **Response.** Thank you for this insightful comment. Our goal is not to simply plug SAM into PSD, but to uncover and exploit a **new mechanism** that repurposes SAM for security.
>
> - **Innovation: using low-rank bias for detection.** Recent theory shows that SAM tends to learn **low-rank features** to achieve flat minima [1]. We observe that typical backdoor triggers behave as **low-rank shortcuts**: simple, highly correlated patterns that are much “simpler” than normal semantic features. Our key finding is that, on poisoned data, SAM does not just improve generalization but **preferentially amplifies these low-rank backdoor directions**, while relatively suppressing higher-rank clean features. This turns SAM’s optimization bias into an explicit **backdoor amplifier** for detection, rather than a generic regularizer.
> - **Distinction from standard SAM usage (e.g., FT-SAM).** Prior work such as FT-SAM applies SAM in a **repair** setting, aiming to **suppress** backdoor effects in a trained model. In contrast, we use SAM in the **detection** stage and intentionally **amplify** backdoor effects on the poisoned dataset to make poisoned samples more separable from clean ones. The defense goal, mechanism, and required data are thus fundamentally different from repair-style methods.
> -
> **Summary.** The contribution is the **conceptual and practical repurposing** of SAM: by leveraging its low-rank bias, we obtain a theoretically grounded mechanism that selectively amplifies backdoor-related features and strengthens poisoned sample detection.
>
> [1] Sharpness-Aware Minimization Leads to Low-Rank Features, NeurIPS 2023.
>
> **Table 1.** Comparison between our method and FT-SAM.
>
> | Criterion         | Our Method (Detection)                                  | FT-SAM (Repair)                               |
> | :--------------- | :------------------------------------------------------ | :-------------------------------------------- |
> | Defense goal     | **Detection** – make poisoned samples identifiable      | **Repair** – make the backdoor ineffective   |
> | Core mechanism   | **Amplify** backdoor effects via low-rank bias          | **Suppress** backdoor signals                 |
> | Data requirement | Operates directly on the **poisoned dataset**           | Requires a **trusted clean dataset**          |
>
> ---
>
> **W2. Concern about repeated figures (e.g., Fig. 2 right and Fig. 5)**
>
> **Response.** We thank the reviewer for pointing out this redundancy and agree that the current presentation can be streamlined. Fig. 2 (right) was intended as a high-level visualization of key statistics (for example, how TAC or detection scores change with or without SAM), while Fig. 5 focuses on a particular analysis case using similar statistics. We acknowledge that this distinction is not sufficiently clear and that the two figures appear repetitive. In the revision, we will **keep a single figure in the main text** as the primary visualization, **move or merge the other into the appendix** as supplementary material, and use the freed space to add **new analysis or experiments**, as suggested.

---

> ### Author Response · Authors · 2025-11-21
> **Response(Part 2)**
>
> **W4. Concern that the method may suffer low performance on hard-to-learn datasets**
>
> **Response.** We appreciate this concern. Harder datasets (for example, with more classes, higher resolution, or complex intra-class variability) are indeed challenging for all PSD methods, and absolute performance tends to degrade in such settings.
> - **Intrinsic difficulty and our objective.** Our goal is not to claim that SAM removes this intrinsic difficulty, but to show that **under the same dataset and backbone**, SAM can consistently **enhance existing PSD methods**, including on harder datasets. Even when the base PSD nearly fails, we expect a clear **relative improvement** after adding SAM.
> - **Evidence on large-scale and complex setups.** To support this, we add experiments on the large-scale **ImageNet-200** dataset with **ResNet-50** and on **CIFAR-10 with ViT-S/16** under BadNets and Blended attacks. On ImageNet-200 with SCAn as the base PSD, SCAn alone almost completely fails (TPR close to 0 at near-zero FPR), indicating that PSD is indeed very difficult in this setting. After integrating SAM, TPR rises to **52.2%** (BadNets) and **83.2%** (Blended) at **0% FPR**. For ViT-S/16 on CIFAR-10, switching from SCAn to SCAn+SAM yields about **10–15 percentage points** TPR improvement under both attacks, with FPR below 1%. Results are detailed in Table 2.
>
> **Summary.** While hard-to-learn datasets remain challenging for all PSD methods, our added experiments on ImageNet-200 and ViT-S/16 show that SAM-enhanced PSD still provides **substantial relative gains** over strong baselines in these settings.
>
>
>
> **Table 2.** Detection performance of SCAn and SCAn+SAM on ImageNet-200 and ViT-S/16.
>
> |Dataset and backbone|Attack|Method|TPR (%)|FPR (%)|
> |:-|:-|:-|:-:|:-:|
> |ImageNet-200, ResNet-50|BadNets|SCAn|0.0|0.0|
> |||SCAn + SAM|**52.2**|**0.0**|
> |ImageNet-200, ResNet-50|Blended|SCAn|0.0|0.0|
> |||SCAn + SAM|**83.2**|**0.0**|
> |CIFAR-10, ViT-S/16|BadNets|SCAn|78.5|0.6|
> |||SCAn + SAM|**93.7**|0.7|
> |CIFAR-10, ViT-S/16|Blended|SCAn|84.1|0.5|
> |||SCAn + SAM|**95.6**|0.8|
>
> ---
>
> **Q1. Discussion on the operational boundaries and efficiency of SAM**
>
> **Response.** We agree that clarifying when and how our method is most effective, and at what cost, is important. We outline two aspects: **mechanism synergy** and **training efficiency**.
> - **Mechanism-specific synergy: feature-space amplifier.** As analyzed in Appendix D.4, SAM shows the **strongest synergy with feature-based PSD methods** such as SCAn and AC. These methods rely on **feature-space distances or statistics**, and SAM explicitly enlarges the separation between poisoned and clean representations. For perturbation-based methods that depend on prediction instability, SAM’s tendency to **stabilize** predictions can lead to mixed effects. Thus our framework is best viewed as a **feature-space amplifier** for PSD, which aligns well with the currently most effective class of defenses.
> - **Efficiency via MSAM.** Standard SAM roughly doubles training cost. To address this, we adopt **Momentum-SAM (MSAM)**, an efficient variant. As detailed in Appendix E.4 and summarized in Table 3, MSAM uses a single forward–backward pass by reusing momentum. In our experiments on ResNet-18 with CIFAR-10, it achieves **SAM-level detection performance** while keeping training time at about **1.07×** that of SGD (1253 s vs. 1165 s), far below the **1.95×** overhead of original SAM.
>
> **Conclusion.** Our method is most effective as a **feature-based PSD booster**, and with MSAM it remains **practically efficient**, staying close to standard SGD cost.
>
>
>
> **Table 3.** Computational complexity and training time comparison.
>
> |Method|Complexity|Training time (s)|Relative time|
> |:-|:-|:-:|:-:|
> |Conventional SGD|O((N/B)E(f + b + p))|1165|1.00×|
> |SAM, original|O(2(N/B)E(f + b + 3p))|2266|1.95×|
> |Momentum-SAM, efficient|O((N/B)E(f + b + 4p))|1253|1.07×|

---

> ### Author Response · Authors · 2025-11-21
> **Response(Part 3)**
>
> **Q2. Concern about the apparent inconsistency between t-SNE visualization and feature-space mixing in Fig. 5**
>
> **Response.** We thank the reviewer for this careful observation. The key point is that the left and right parts of Fig. 5 operate in **very different spaces**: a nonlinear 2D visualization versus the original high-dimensional feature space.
> - **T-SNE is only a qualitative visualization.** The left subfigure uses t-SNE, which is a **nonlinear, probabilistic** method intended for 2D visualization. It preserves local neighborhoods approximately but does not preserve true Euclidean distances or global geometry. t-SNE often **exaggerates cluster separation** to produce visually interpretable plots. Thus, “almost separated” clusters in the 2D t-SNE plot do **not** imply that samples are nearly perfectly separable in the original feature space.
> - **Right plot reflects true feature-space overlap.** The right subfigure is computed directly from statistics in the **original feature space** for example, distributional distances or overlaps between clean and poisoned features. This is the space in which PSD actually operates. It is therefore not surprising that this plot shows poisoned samples still mixed with clean ones: this reflects the **real difficulty** of the detection problem and matches the fact that our method, while significantly improving TPR/F1, does not achieve perfect separation.
>
> **Conclusion.** Once we distinguish the roles of the two plots, they are consistent: t-SNE provides **qualitative intuition** about how SAM reshapes structure, while the right-hand statistics provide the **quantitative evidence** we rely on. In the revision, we will explicitly explain this distinction and clarify that our conclusions are based on metrics in the original feature space, with t-SNE used only as a visual aid.

---

> ### Author Response · Authors · 2025-11-21
> **Response(Part 4)**
>
> **Q3. Concern about FPR behavior and rare clean patterns under SAM-enhanced PSD**
>
> **Response.** We appreciate this observation. The reviewer asks why FPR changes only slightly while TPR improves, and whether SAM may also amplify **rarely occurring clean patterns**, especially on hard datasets. We address (i) why SAM enlarges intra-class variance, (ii) why backdoor signals still dominate, and (iii) why this does not hurt the final model in practice.
> - **Cause: SAM broadens clean feature distributions.** SAM minimizes an upper bound on the loss under parameter perturbations and thus seeks **flat minima**. This leads to solutions where the loss is stable in a neighborhood of the parameters and, at the representation level, to **broader clean feature clusters**. A small fraction of benign samples near the decision boundary, including rare but legitimate patterns, may move further from the class center and be flagged as positives, which explains the **small FPR increase**. In Figure X, for example, the within-class variance of clean “airplane” features on CIFAR-10 increases from **2.50** (vanilla training) to **3.90** (SAM). Thus SAM does not only amplify poisoned samples; it also slightly expands the clean distribution, so FPR is not expected to improve in the same way as TPR.
> - **Backdoor signal remains dominant.** Although SAM increases absolute variance, it amplifies **backdoor-related directions** much more strongly than benign variations. TAC and PCA analysis show that high-TAC neurons dominate the principal components of poisoned features, while rare clean patterns contribute much less. Table 4 quantifies this for the target class “airplane” under the Blended attack: with SAM, within-class variance grows from 2.50 to 3.90, but the **S-score between clean and poisoned features also increases** from **0.32** to **0.42**. This means that **relative separation between clean and poisoned samples becomes larger**, not smaller. In other words, clean features are slightly more spread out, but poisoned samples move even farther away, so the detection subspace is still dominated by backdoor signals.
> - **Effect on hard datasets and final model performance.** Across datasets and attacks, SAM-enhanced PSD achieves **large TPR gains** with only **small, controlled FPR changes**, which is consistent with the above picture. More importantly, after retraining on the purified data, the final model maintains clean accuracy almost identical to a clean baseline, while ASR is drastically reduced. This shows that the extra benign samples flagged due to variance expansion **do not materially harm the final decision boundary**. On harder or more imbalanced datasets, one can further adjust PSD thresholds to trade off TPR and FPR if needed, but under our default settings we already observe a favorable balance.
>
> **Conclusion.** SAM does enlarge intra-class variance and can slightly amplify rare benign patterns, which explains why FPR changes only mildly. However, **backdoor-related directions are amplified more strongly**, as evidenced by the S-score increase from 0.32 to 0.42 in Table 4, so Stage-2 features remain dominated by backdoor signals. This leads to higher TPR with small FPR changes and negligible loss of clean accuracy for the final retrained model.
>
> **Table 4.** Effect of SAM on intra-class variance and S-score for the target class “airplane” under the Blended attack on CIFAR-10.
>
> |Metric|VT|SAM|
> |:-|:-:|:-:|
> |Within-class variance of clean “airplane”|2.50|3.90|
> |S-score between clean and poisoned “airplane”|0.32|0.42|

---

> > ### Author Response · Authors · 2025-11-27
> > **Response**
> >
> > Dear Reviewers,
> >
> > I hope this message finds you well. I understand that you are very busy and truly appreciate the time and effort you devote to the review process.
> >
> > As the review deadline is approaching, we kindly wanted to check whether there are any updates regarding the reviews of our submission. If there is any additional information or clarification needed from our side, we would be more than happy to provide it.
> >
> > Thank you very much for your time and support. We sincerely appreciate your help and contributions to the community.
> >
> > Best regards,
> > Authors

---

### Official Review · Reviewer_bYYD · 2025-10-28

**Soundness:** 3
**Presentation:** 3
**Contribution:** 2
**Rating:** 6
**Confidence:** 2

**Summary:**

The paper introduces a method for detecting poisoned samples in weak-trigger backdoor attacks. The authors propose to use Sharpness-Aware Minimization (SAM) to amplify the representations of backdoor-related neurons, improving the separability between poisoned and clean samples. The approach is theoretically grounded and shows consistent gains across multiple detection methods.

**Strengths:**

The studied setting is both challenging and practical, focusing on data poisoning with weak and inconspicuous triggers.

The proposed method is principled, following an optimization-based strategy with clear theoretical motivation.

Experimental results demonstrate that it significantly improves the detection accuracy of existing poisoned-sample detection approaches.

**Weaknesses:**

The method still requires access to a small number of clean samples for feature extraction, which may limit applicability in fully unsupervised settings.

The approach introduces noticeable computational overhead due to the use of SAM. Although the authors mention efficient variants such as MSAM, the extra cost during training remains substantial.

**Questions:**

How does the performance scale with the number of available clean samples?

It would be helpful if the authors could provide more details on how backdoor-related features are extracted in Section 3.4, since this step is central to the method.

---

> ### Author Response · Authors · 2025-11-21
> **Response(Part 1)**
>
> We sincerely thank the reviewer for the time and effort spent evaluating our work and for the **encouraging** comments. We are glad that the reviewer found the considered weak-trigger poisoning setting both **challenging** and **practical**, viewed our SAM-based approach as a **principled**, optimization-driven method with clear theoretical motivation, and recognized that the experimental results show **consistent** and **significant** improvements for existing poisoned-sample detection approaches. We address the remaining concerns in detail below.
>
> ---
>
> **W1 & Q1. Concern regarding data dependency: limits in unsupervised settings and performance scaling with clean data size**
>
> **Response.** Thank you for these practical questions. We agree that data dependency and unsupervised applicability are central for PSD. We address both **applicability without trusted labels (W1)** and **scaling behavior with clean data size (Q1)** using the ablation studies in Appendix E.2.
>
> - **Applicability in fully unsupervised settings.** Our method does **not** require a perfectly clean, labeled in-distribution reference set. We validate two realistic strategies:
>   - **Self-extraction via Meta-Sift.** We apply META-SIFT [1] directly on the poisoned training set to obtain an “identified-clean” subset, without any trusted labels. As shown in Table 2, using this imperfect subset still yields performance comparable to using a ground-truth clean set, for example **98.85% TPR** on Blended.
>   - **Using public OOD data.** We further use CIFAR-5m [2] as an out-of-distribution auxiliary set. Table 2 shows that the method remains highly effective, achieving **98.76% TPR** on Blended. This indicates that **generic public datasets can substitute task-specific in-distribution data**.
> - **Performance scaling with reference size.** We study how detection performance changes when the auxiliary clean set is reduced by varying the clean ratio from **5%** down to **0.5%** (only 25 samples per class). As shown in Table 1, performance is remarkably stable: for Blended, TPR changes only from **98.74%** to **98.61%**, and for SSBA it stays around **96.3%** even at the smallest ratio. This indicates that our method **saturates with a very small clean subset** and does not rely on scaling up clean data to maintain high performance.
>
> **Conclusion.** Overall, our SAM-based PSD is **data-efficient and flexible**: it can operate in fully unsupervised settings via self-extraction or OOD data, and its performance remains strong even when the auxiliary clean set is reduced to a tiny fraction.
>
> [1] Meta-Sift: How to Sift Out a Clean Subset in the Presence of Data Poisoning? USENIX Security 2023.
> [2] The Deep Bootstrap Framework: Good Online Learners Are Good Offline Generalizers. arXiv 2020.
>
> ---
>
> **Table 1.** TPR and FPR for SAM-enhanced detectors under different attacks and clean auxiliary set ratios.
>
> | Attack, clean ratio | SCAn+SAM TPR | SCAn+SAM FPR | Beatrix+SAM TPR | Beatrix+SAM FPR | AC+SAM TPR | TIAC+SAM FPR |
> | :------------------ | :----------: | :----------: | :-------------: | :-------------: | :--------: | :----------: |
> | Blended, 0.5%       | 98.61        | 0.00         | 99.91           | 5.00            | 98.82      | 0.00         |
> | Blended, 5%         | 98.74        | 0.00         | 99.65           | 5.00            | 98.82      | 0.00         |
> | SSBA, 0.5%          | 96.31        | 0.00         | 99.91           | 5.00            | 96.47      | 0.00         |
> | SSBA, 5%            | 96.52        | 0.00         | 99.42           | 5.00            | 96.51      | 0.00         |
> | LF, 0.5%            | 95.91        | 0.00         | 98.83           | 5.00            | 96.02      | 0.00         |
> | LF, 5%              | 96.16        | 0.00         | 99.01           | 5.00            | 96.12      | 0.00         |
>
> **Table 2.** TPR and FPR for SCAn and AC, with and without SAM, using identified-clean and OOD auxiliary data on CIFAR-10.
>
> | Attack, auxiliary data | SCAn / +SAM TPR | SCAn / +SAM FPR | AC / +SAM TPR | AC / +SAM FPR |
> | :--------------------- | :-------------- | :-------------- | :------------ | :------------ |
> | Blended, auxiliary     | 99.36 / 98.85   | 0 / 0.53        | 2.57 / 99.60  | 7.94 / 0.84   |
> | SSBA, auxiliary        | 94.18 / 97.41   | 0 / 0.53        | 100 / 96.84   | 3.84 / 0      |
> | LF, auxiliary          | 94.77 / 96.31   | 0 / 0           | 95.80 / 96.34 | 10.89 / 0     |
> | Blended, OOD           | 99.08 / 98.76   | 0 / 0           | 0 / 98.76     | 0 / 0         |
> | SSBA, OOD              | 94.04 / 96.48   | 0 / 0           | 99.36 / 96.52 | 0 / 0         |
> | LF, OOD                | 94.96 / 96.04   | 0 / 0           | 95.48 / 96.04 | 0 / 0         |

---

> ### Author Response · Authors · 2025-11-21
> **Response(Part 2)**
>
> **W2. Concern about computational overhead introduced by SAM**
>
> **Response.** We agree that the extra cost of standard SAM is an important practical concern. Our main message is that **the performance benefits of our method can be retained with almost no additional cost** by using an efficient SAM variant.
>
> - **Computation: standard SAM vs. MSAM.**  Standard SAM requires two forward–backward passes per step and roughly doubles training time. In contrast, **Momentum-SAM (MSAM)** constructs the perturbation from accumulated momentum and needs only **one forward–backward pass**, making its complexity very close to SGD.
>
> - **Runtime validation.**  In Appendix, we replace SAM with MSAM and measure both training time and detection performance on ResNet-18 with CIFAR-10 (RTX 4090). As shown in Table 4, standard SAM increases training time from 1165 s (SGD) to 2266 s, whereas MSAM finishes in **1253 s**, only **1.07×** the cost of SGD, while preserving almost the same detection performance as SAM (Table 3, e.g., **94.9% TPR** on Blended).
>
> **Conclusion.** By adopting MSAM, our framework achieves **SAM-level robustness at nearly SGD-level cost**, so computational overhead is not a fundamental limitation.
>
> [1] Moorgathu et al. Momentum-SAM: Gradient Optimization with Momentum-based Sharpness-Aware Minimization. Neural Networks 2024.
>
> **Table 3.** Detection performance of MSAM vs. base training and standard SAM.
>
> | Attack  | Detector | TPR, Base / +SAM / +MSAM | FPR, Base / +SAM / +MSAM |
> | :------ | :------- | :----------------------- | :----------------------- |
> | BadNets | SS       | 70.8 / 92.3 / **91.9**   | 2.4 / 1.2 / **0.0**      |
> | Blended | SS       | 32.9 / 94.6 / **94.9**   | 4.4 / 1.1 / **0.0**      |
> | SSBA    | SCAn     | 93.9 / 96.5 / **96.7**   | 0.0 / 0.0 / **0.5**      |
>
> **Table 4.** Computational complexity and wall-clock time.
>
> | Method                  | Complexity                     | Training time | Relative time |
> | :---------------------- | :---------------------------- | :------------ | :------------ |
> | Conventional SGD        | \(O((N/B)E(f + b + p))\)      | 1165 s        | 1.00×         |
> | SAM, original           | \(O(2(N/B)E(f + b + 3p))\)    | 2266 s        | 1.95×         |
> | **Momentum-SAM, efficient** | **\(O((N/B)E(f + b + 4p))\)** | **1253 s**    | **1.07×**     |
>
> ---
>
> **Q2. Concern about how backdoor-related features are extracted in Stage-2**
>
> **Response.** We appreciate this question and will make the Stage-2 **backdoor-related feature (BRF)** extraction more explicit and concise.
>
> - **Stage-2 procedure.**  After training the model with SAM, we take the last-layer representation $g = \phi_{\theta_{\mathrm{SAM}}}(x)$ for each sample, center it using the mean feature $\mu$ from the (identified-)clean subset, learn a PCA projection $P$ on the centered training features, estimate the covariance $\Sigma$ in this PCA space from clean features, and then compute $g_s = \Sigma^{-1/2} P (g - \mu)$. We use $g_s$ as the BRF and feed it to any off-the-shelf feature-based PSD method without modifying its internal logic.
> - **Why BRF is backdoor-related.** Our analysis shows that **high-TAC neurons have large loadings on the leading PCA components** of SAM-trained features, meaning this PCA subspace is aligned with backdoor-sensitive directions. Whitening with $\Sigma^{-1/2}$ further suppresses variation already explained by clean data and highlights deviations caused by poison. Thus, $g_s = \Sigma^{-1/2} P (g - \mu)$ serves as a practical, unsupervised approximation of backdoor-related features, which we will emphasize and support with clearer pointers to the ablations in Section 4.3 and the appendix.

---

> > ### Author Response · Authors · 2025-11-27
> > **Response**
> >
> > Dear Reviewers,
> >
> > I hope this message finds you well. I understand that you are very busy and truly appreciate the time and effort you devote to the review process.
> >
> > As the review deadline is approaching, we kindly wanted to check whether there are any updates regarding the reviews of our submission. If there is any additional information or clarification needed from our side, we would be more than happy to provide it.
> >
> > Thank you very much for your time and support. We sincerely appreciate your help and contributions to the community.
> >
> > Best regards,
> > Authors

---

### Official Review · Reviewer_CppV · 2025-10-30

**Soundness:** 3
**Presentation:** 4
**Contribution:** 2
**Rating:** 4
**Confidence:** 4

**Summary:**

The paper addresses the problem of Poisoned Sample Detection (PSD) in defending deep neural networks against backdoor attacks. It observes that existing PSD methods degrade significantly when the backdoor effect is weak (e.g., low poisoning ratio or weak trigger patterns). Through extensive statistical analysis, the authors reveal a strong positive correlation between the backdoor effect (quantified via Trigger Activation Change, TAC) and detection performance. To enhance detection, the paper proposes SAM-enhanced PSD, a framework that applies Sharpness-Aware Minimization (SAM) during model training to amplify the backdoor effect without modifying the dataset. This increases feature-space separability between poisoned and clean samples, boosting the performance of existing PSD detectors. Experiments on CIFAR-10, Tiny ImageNet, and GTSRB across multiple architectures and attack types show an average True Positive Rate (TPR) gain of +34.3% over baselines.

**Strengths:**

1. Clear empirical motivation and clarity: The paper is well organized and easy to follow, empirically analyzing the correlation between backdoor strength and detection success.
2. Method simplicity and compatibility: SAM-enhanced training can be applied universally to many PSD methods with no structural modification.
3. Comprehensive Experimental Validation: Across 13 attack types and 5 detection methods, the SAM-enhanced pipeline shows robust gains, particularly under weak backdoor attacks. Extensive evaluation and ablation include adaptive attacks, robustness to auxiliary data, runtime efficiency, and stability analysis.

**Weaknesses:**

1. Limited Novelty: Although the proposed idea is elegant, it essentially amounts to incorporating Sharpness-Aware Minimization (SAM), Trigger Activation Change(TAC) and Spectre into the standard training process to enhance feature separability between poisoned samples and benign samples. The work does not introduce a fundamentally new algorithmic component or theoretical framework. Moreover, the theoretical analysis, while insightful, is derived under a simplified two-layer ReLU assumption and may not fully generalize to deeper or more complex network architectures.

2. Restricted dataset and model diversity: Despite the extensive number of experiments, the evaluations are limited to relatively simple datasets (e.g., CIFAR-10, Tiny ImageNet) and standard architectures (e.g., ResNet-18). To demonstrate broader applicability, the study should include results on more complex or large-scale architectures such as ViT or CLIP, and potentially extend to other modalities like text or multimodal backdoor scenarios.

**Questions:**

1. The authors propose the key statement "These results suggest that SAM selectively amplifies the most discriminative backdoor features by encouraging sharper activation patterns, thereby enhancing the backdoor effect."
For me, SAM method compels attackers to more carefully select adversarial perturbations to achieve their goal, thereby increasing the divergence between poisoned and normal samples on the TAC. This stems from SAM being a more robust learning approach; attacking SAM models introduces greater variations at the activation, making PSD inherently easier. This conclusion follows naturally and is not a key contribution from the authors of this paper.

2. Increased FPR: Several experimental results indicate a rise in FPR when applying SAM-enhanced PSD. The paper would benefit from a deeper analysis of this phenomenon, including potential causes (e.g., over-amplification of benign neuron activations) and mitigation strategies.

---

> ### Author Response · Authors · 2025-11-21
> **Response(Part 1)**
>
> We sincerely thank the reviewer for the time and care spent assessing our work and for the **encouraging**, **detailed** feedback. We are pleased that the reviewer found our empirical motivation and analysis **clear**, regarded the SAM-enhanced PSD framework as **simple** and **compatible** with existing methods, and acknowledged the **comprehensive** experimental validation across diverse attacks and detectors. We address the remaining concerns below.
>
> ---
>
> **W1. Concern regarding the novelty of the algorithmic components and the generalizability of the theoretical analysis**
>
> **Response.** Thank you for this insightful comment. Although our method uses standard tools such as SAM and PCA, it is not a simple combination. Our main contribution is to **identify and exploit a new mechanism** that repurposes SAM’s optimization behavior for backdoor detection.
>
> - **Novelty: low-rank bias as a backdoor amplifier.**
>   Recent work shows that SAM tends to learn **low-rank features** to reach flat minima. We are, to the best of our knowledge, the first to point out that on poisoned datasets this **low-rank bias** has a security-relevant consequence: it **preferentially amplifies backdoor triggers**, which act as simple low-rank shortcuts, while relatively suppressing complex semantic features. We explicitly use this property to increase the separability of poisoned samples in feature space.
>
> - **Paradigm shift relative to prior SAM-based defenses.**
>   Prior work such as FT-SAM applies SAM in a **post-training repair** setting, aiming to **suppress** backdoor effects in a trained model. Our method instead works in the **pre-training data cleansing** stage and deliberately uses SAM to **amplify** backdoor traces so that PSD can remove poisoned samples. Thus the defense stage, goal, and mechanism are fundamentally different, as summarized in Table 1; we effectively **reverse SAM’s role** from repair to detection.
>
> - **Theoretical generalizability beyond simplified models.**
>   Our analysis starts from a two-layer model to make optimization dynamics tractable, in line with the view that deep ResNets behave like ensembles of shallower subnetworks. The resulting picture matches general theory showing that SAM induces a **low-rank feature bias**, which is independent of depth. We then confirm the predicted amplification behavior empirically on deep architectures such as ResNet-18.
>
> **Conclusion.** The novelty lies in **using SAM’s low-rank bias as a mechanism that selectively amplifies backdoor triggers** and **repurposing SAM from a repair tool to a detection tool**, supported by theory and experiments on modern networks.
>
> **Table 1.** Fundamental differences between our method and FT-SAM.
>
> |Criterion|Our Method (for Detection)|FT-SAM (for Repair)|
> :-|:-|:-
> Defense stage|**Pre-training**, data cleansing|**Post-training**, model repair
> Goal|**Amplify** backdoor effects to separate poisoned data|**Suppress** backdoor signals for safe inference
> Mechanism|Exploits SAM’s **low-rank bias** to expose triggers|Uses SAM’s flatness to erase or weaken triggers

---

> ### Author Response · Authors · 2025-11-21
> **Response(Part 2)**
>
> **W2. Concern that experiments focus on relatively simple datasets/architectures and may not reflect broader applicability**
>
> **Response.** We agree that demonstrating scalability beyond small datasets and standard CNNs is important. We therefore add experiments on both a **larger-scale dataset** and a **transformer-based backbone**, without changing the PSD pipeline.
>
> - **Evidence on ImageNet-200 and ViT backbones.**
>   We evaluate SCAn and SCAn+SAM on **ImageNet-200 with ResNet-50** and on **CIFAR-10 with ViT-S/16** under BadNets and Blended attacks. As shown in Table 2, SCAn almost fails on ImageNet-200 (TPR ≈ 0% at ≈ 0% FPR), while **SCAn+SAM** raises TPR to **52.2%** (BadNets) and **83.2%** (Blended) at **0% FPR**. On CIFAR-10 with ViT-S/16, SAM improves TPR by roughly **10–15 percentage points** under both attacks with FPR below 1%. This shows that our method **scales to larger datasets and generalizes to transformer backbones**.
>
> - **Backbone- and modality-agnostic design.**
>   Our framework only changes the **optimizer** during PSD training (replacing SGD/Adam with SAM) and leaves detector architecture and objective untouched. This makes it **backbone-agnostic** and, in principle, applicable to many vision models (CLIP-like encoders, Swin, ConvNeXt) and even non-vision modalities, since it only requires gradients and the loss. A full study on CLIP-scale or multimodal/text backdoors is beyond the current scope but is now stated as **promising future work**.
>
> **Conclusion.** The new ImageNet-200 and ViT-S/16 results in Table 2 give concrete evidence that our SAM-based PSD framework **scales to larger datasets and more complex architectures**, while its backbone-agnostic formulation naturally supports future extensions to CLIP-like and multimodal settings.
>
> **Table 2.** Detection performance of SCAn and SCAn+SAM on ImageNet-200 and ViT-S/16.
>
> |Dataset and backbone|Attack|Method|TPR (%)|FPR (%)|
> :-|:-|:-|:-:|:-:
> ImageNet-200, ResNet-50|BadNets|SCAn|0.0|0.0
> |||SCAn + SAM|**52.2**|**0.0**
> ImageNet-200, ResNet-50|Blended|SCAn|0.0|0.0
> |||SCAn + SAM|**83.2**|**0.0**
> CIFAR-10, ViT-S/16|BadNets|SCAn|78.5|0.6
> |||SCAn + SAM|**93.7**|0.7
> CIFAR-10, ViT-S/16|Blended|SCAn|84.1|0.5
> |||SCAn + SAM|**95.6**|0.8
>
> ---
>
> **Q1. Concern that the claimed “selective amplification of backdoor features” may just reflect SAM’s general robustness, rather than a distinct contribution**
>
> **Response.** Our mechanism is **representation-level** and is evaluated on a fixed poisoned dataset; it does not rely on attacker adaptation.
>
> - **Threat model and what changes.**
>   Backdoor triggers are **fixed** and trained with standard procedures, matching common benchmarks. SAM is used only on the **defender side** by swapping the optimizer in the PSD training pipeline. Thus the larger TAC divergence we observe arises purely from how SAM reshapes the learned features, not from stronger attacks.
>
> - **Beyond generic robustness: low-rank bias.**
>   We compare PSD with and without SAM on the **same poisoned data**. The only difference is the optimizer. Theory shows that SAM prefers **low-rank feature representations**, compressing the feature space into a few dominant directions. This **low-rank bias** is more specific than generic adversarial robustness.
>
> - **Backdoor triggers as low-rank shortcuts.**
>   In poisoned data, typical triggers are **simple, highly correlated shortcuts**, effectively low-rank compared to semantic features. Under SAM’s low-rank bias, the optimizer relies more on these simple patterns. Consequently, backdoor-related neurons are **preferentially strengthened**, and for the **same** poisoned samples we observe **much larger TAC values** under SAM, while TAC on typical clean directions changes only mildly. Poisoned samples thus concentrate along a few dominant directions and become easier to separate in Stage-2.
>
> - **Positioning the contribution.**
>   Our contribution is therefore to
>   – **identify and validate** that SAM’s low-rank bias leads to **selective over-amplification of backdoor features**;
>   – **design a plug-and-play enhancement** that improves diverse PSD baselines simply by swapping the optimizer; and
>   – **demonstrate these gains under standard, non-adaptive attacks**, confirming that they come from this representation-level mechanism.
>
> **Conclusion.** SAM’s low-rank bias **selectively amplifies backdoor-related features** in a fixed poisoned dataset, and we explicitly harness this effect to enhance PSD. This is qualitatively different from generic robustness and does not require attacker adaptation.

---

> ### Author Response · Authors · 2025-11-21
> **Response(Part 3)**
>
> **Q2. Concern about increased FPR under SAM-enhanced PSD, potential causes, and mitigation**
>
> **Response.** This concern relates to the small FPR changes observed in our experiments. We explain (i) why SAM enlarges intra-class variance, (ii) why backdoor signals still dominate, and (iii) why the final model is not harmed.
>
> - **Effect on clean feature variance.**
>   By minimizing an upper bound on the loss under parameter perturbations, SAM seeks **flat minima**. This yields solutions where the loss is stable in a neighborhood of the parameters and, at the representation level, **broader clean feature clusters**. A small fraction of benign samples near the decision boundary can move further from the class center and be flagged as positives, causing a **small FPR increase**. For example, the within-class variance of clean “airplane” features on CIFAR-10 grows from **2.50** (vanilla training) to **3.90** (SAM).
>
> - **Backdoor directions remain dominant.**
>   Although SAM increases absolute variance, it strengthens **backdoor-related directions** more than benign ones. TAC and PCA analysis show that high-TAC neurons dominate the principal components of poisoned features, while rare clean patterns contribute much less. Table 3 quantifies this for the target class “airplane’’ under the Blended attack: with SAM, variance rises from 2.50 to 3.90, but the **S-score between clean and poisoned features also increases** from **0.32** to **0.42**. Thus the **relative separation** between clean and poisoned samples becomes larger even though the clean cluster is slightly more spread out.
>
> - **Impact on the final model.**
>   Across datasets and attacks, SAM-enhanced PSD yields **large TPR gains** with only **small, controlled FPR changes**. After retraining on the purified data, the final model retains clean accuracy essentially identical to a clean baseline while sharply reducing ASR, indicating that extra benign samples flagged by SAM **do not materially harm the final decision boundary**.
>
> **Conclusion.** SAM does enlarge intra-class variance and can slightly amplify rare benign patterns, which explains the modest FPR changes. However, **backdoor-related directions are amplified more strongly**, as shown by the S-score increase from 0.32 to 0.42, so Stage-2 remains dominated by backdoor features and achieves higher TPR with negligible loss of clean accuracy.
>
> **Table 3.** Effect of SAM on intra-class variance and S-score for the target class “airplane” under the Blended attack on CIFAR-10.
>
> |Metric|VT|SAM|
> :-|:-:|:-:
> Within-class variance of clean “airplane”|2.50|3.90
> S-score between clean and poisoned “airplane”|0.32|0.42

---

> ### Comment · Reviewer_CppV · 2025-11-25
> **Contribution concerns remain.**
>
> Thank you for your rebuttal.
>
> You have provided insightful perspectives on FPR issues. However, my concerns regarding the novelty of the methodology remain. While the authors may be the first to point out \textbf{“low-rank bias as a backdoor amplifier”}, as other reviewers mentioned, SAM-induced low-rank bias may amplify all dominant low-frequency features, and backdoor features amplification may be a byproduct of this process. This remains a valuable observation but not a novel idea.
>
> Additionally, re-training an optimizer using the SAM method is somewhat perplexing. Given the same computational cost, why not simply replace the original model $\theta$ with $\theta_{SAM}$ to achieve both strong generalization and high PSD performance?

---

> > ### Author Response · Authors · 2025-11-25
> > **Clarifying Novelty and Methodology**
> >
> > Dear Reviewer CppV
> >
> > We thank you for the continued engagement and for acknowledging the value of our observation regarding low-rank bias. However, we would like to respectfully address the remaining concerns regarding novelty and the clarification of our methodology.
> >
> > **1. On the Novelty of the Methodology**
> >
> > We respectfully argue that our contribution goes beyond merely observing a byproduct of SAM. Our novelty lies in a fundamental **paradigm shift** in Poisoned Sample Detection or PSD:
> >
> > * **Active Feature Enhancement vs. Passive Extraction:** Existing PSD methods largely focus on how to extract discriminative features from a standard model, such as using sophisticated statistical tools or clustering. They treat the learned features as a fixed given. In contrast, our work is the first to propose **actively manipulating the training dynamics** to amplify the feature separability before detection even begins. We shift the focus from better extraction to better representation.
> >
> > * **Strategic Synergy, Not Coincidence:** Building on the observation that backdoor patterns are inherently dominant and typically concentrated within a small subset of core neurons (measurable by high TAC), employing SAM becomes a **natural choice**. Since SAM explicitly induces a low-rank bias, it naturally favors and amplifies these sparse, dominant backdoor structures while suppressing irrelevant features. **This intrinsic alignment makes SAM the perfect candidate to fulfill the active feature enhancement requirement proposed in our first point.** The fact that SAM is mathematically suitable for this task does not diminish the novelty; rather, **identifying the right tool to solve the specific problem of weak backdoor detection where the signal is faint is the core innovation.** We are not claiming to invent SAM, but rather discovering its critical and previously unexplored role in forcing a separation between clean and poisoned samples in the pre-training stage.
> >
> > **2. Clarification on Methodology and Workflow**
> >
> > Regarding the comment on replacing the original model $\theta$ with $\theta_{SAM}$, we believe there is a slight misunderstanding of the standard PSD pipeline and our specific setting.
> >
> > * **The Model is Backdoored:** The model $\theta_{SAM}$ is trained on the potentially poisoned dataset $\mathcal{D}_{tr}$. Consequently, **$\theta_{SAM}$ itself contains the backdoor** and poses a security risk if deployed directly. It cannot simply replace the original model for downstream tasks if safety is the priority.
> >
> > * **The Role of $\theta_{SAM}$:** In our framework, $\theta_{SAM}$ serves as a **probe** or a feature amplifier. Its purpose is to generate feature representations where poisoned samples are distinctly separated from clean ones.
> >
> > * **The Standard Pipeline:** Our method follows the standard defense protocol. First, we train the probe model $\theta_{SAM}$ on the suspicious dataset. Second, we use $\theta_{SAM}$ to extract features and detect or remove poisoned samples, which constitutes the purification step. Third, **we train a final clean model** using standard training or any preferred method on the purified dataset for deployment.
> >
> > Therefore, we do not train an optimizer or perform redundant steps. We simply optimize the probe model to ensure the subsequent detection step is successful.
> >
> > We hope this clarifies that our approach offers a novel perspective on feature enhancement for defense and aligns with established safety protocols.

---

> > > ### Author Response · Authors · 2025-11-27
> > > **Response**
> > >
> > > Dear Reviewers,
> > >
> > > I hope this message finds you well. I understand that you are very busy and truly appreciate the time and effort you devote to the review process.
> > >
> > > As the review deadline is approaching, we kindly wanted to check whether there are any updates regarding the reviews of our submission. If there is any additional information or clarification needed from our side, we would be more than happy to provide it.
> > >
> > > Thank you very much for your time and support. We sincerely appreciate your help and contributions to the community.
> > >
> > > Best regards,
> > > Authors

---

> ### Comment · Reviewer_CppV · 2025-11-28
>
> I appreciate your explanation.
> So $\theta_{sam}$ is trained on the potentially poisoned dataset and then used to purify the potentially poisoned dataset  itself?
> To get a clear model on the specified dataset, we should train two model $\theta_{sam}$ and $\theta$, which obviously double the cost of compuation.
>
> Can we train some $\theta_{sam}$ just as a feature extractor (at the same time a feature amplifier) independent on a specified dataset to adapt different tasks?

---

> > ### Author Response · Authors · 2025-11-28
> > **Response**
> >
> > We thank the reviewer for the constructive suggestion regarding the independent feature extractor and computational efficiency. We would like to gently clarify that the issues raised—specifically the unavailability of clean feature extractors and the two-stage computational cost—are **intrinsic characteristics and standard assumptions of the "Pre-training Defense" field itself**, rather than specific limitations of our proposed method.
> >
> > **1. On the General Assumption of Pre-training Defense**
> > The reviewer asks about using an independent feature extractor. It is important to note that **the foundational assumption of Pre-training Defense is the absence of reliable, clean pre-trained models or sufficient external data.**
> > * **The Standard Setting:** If we assumed the availability of a clean, independent feature extractor, the defense problem would become trivial, reducing to simple label-noise detection, and would rely on a much stronger assumption than what is typically permitted in this field.
> > * **Our Alignment:** Our approach adheres strictly to this standard setting. We *must* perform feature extraction on the poisoned dataset itself because the premise of this research direction is that the defender starts with nothing but the compromised data.
> >
> > **2. On the "Double Cost" as a Standard Paradigm**
> > Regarding the concern about training two models, this is the **standard workflow for all Dataset Purification methods** in the pre-training stage, not a unique overhead of our specific approach.
> > * **Inherited Workflow:** In this field, the objective is to produce a safe *dataset* for downstream use. Therefore, the process inherently requires a "Purification + Re-training " pipeline.
> > * **Field-Wide Trade-off:** The computational cost is the accepted price in the literature for obtaining a purified dataset that can be safely used for *any* downstream task. This is a general property of pre-training defenses, distinct from the efficiency of our specific algorithm.
> >
> > We hope this clarifies that our experimental design reflects the standard protocols and necessary assumptions of the pre-training defense community.
> >
> > Best regards
> >
> > Authors

---

### Official Review · Reviewer_ELXT · 2025-11-04

**Soundness:** 2
**Presentation:** 3
**Contribution:** 3
**Rating:** 4
**Confidence:** 4

**Summary:**

This paper investigates why existing pre-training poisoned sample detection methods fail under weak or low-poisoning-rate backdoor attacks, and proposes a principled enhancement mechanism using Sharpness-Aware Minimization. The authors begin by establishing a strong positive correlation between the "backdoor effect" strength and PSD performance. A weak backdoor effect leads to poor separability between poisoned and clean samples in the feature space, thereby crippling detection. The authors find that higher Top-k TAC values correspond to higher separability between clean and poisoned samples. They then theoretically and empirically show that training the model with SAM amplifies TAC on “backdoor neurons,” thereby enhancing the discriminability of poisoned samples.
Based on this, the paper introduces SAM-enhanced PSD. The three-stage pipeline involves: (1) training a model on the suspicious dataset using SAM; (2) extracting features from this SAM-trained model (using PCA as a surrogate for high-TAC neuron features); and (3) feeding these "enhanced" features to existing feature-based PSD detector. Experiments show that this approach significantly boosts the performance of various PSD methods against both strong and weak attacks.

**Strengths:**

1. The paper identifies an interesting empirical relationship between backdoor effect strength (Top-k TAC) and PSD performance, which offers some  interpretation of why some PSDs fail under weak attacks.
2. SAM-enhanced training is plug-and-play and can be combined with many existing PSDs
3. An analytical study (in a 2-layer ReLU model) shows why SAM’s second-order correction term selectively increases activation differences on “backdoor neurons” while regularizing irrelevant ones.
4. Results cover multiple datasets, network architectures, and attack types, showing consistent TPR and F1 improvement. This method maintains clean accuracy while increases computational cost (≈2× over SGD).

**Weaknesses:**

1. Stage-2 (PCA + covariance whitening) still assumes access to some clean or near-clean reference data, which may not always be available in realistic settings.

2. SAM roughly doubles training cost. Although mitigated by MSAM/ASAM variants, this cost may limit scalability to large-scale or foundation models.

3. Limited analysis of adaptive attackers. The paper briefly mentions adaptive attacks but does not rigorously test scenarios where attackers design triggers to minimize TAC or exploit SAM’s bias.

4. Experiments focus on standard small-scale datasets. Demonstrations on larger or self-supervised models would better validate the scalability and generality of the approach.

**Questions:**

1. Is there a systematic way to balance enhanced separability (TAC increase) with preserved clean accuracy?
2. Can the SAM-induced amplification of backdoor neurons inadvertently amplify benign, class-specific rare patterns, potentially increasing false positives?
    It would be interesting to see whether SAM also enlarges intra-class variance or amplifies rare but legitimate sub-patterns, which might explain potential FPR changes.
3. SAM is known to improve generalization, but how does it affect the clean accuracy (ACC) and robust accuracy (RA) of models trained on poisoned data before detection? The retraining results in Sec. C.6 are promising, but could you provide metrics on the intermediate SAM-trained model's performance prior to PSD application?
4. It is good that Sec. C.4 shows limited gains for perturbation- and topology-based detectors, but I am still uncertain why SAM suppresses rather than enhances these methods.
5. Since SAM is originally designed to improve model generalization by seeking flatter minima, could its effect of amplifying backdoor-related features be interpreted as a byproduct of enhancing the model’s sensitivity to stable patterns in the poisoned training data?
6. If the attacker uses multiple, distinct trigger patterns across the poisoned dataset, can SAM still effectively identify and amplify a coherent signal for detection?

---

> ### Author Response · Authors · 2025-11-21
> **Response(Part 1)**
>
> We sincerely thank the reviewer for the time and effort devoted to assessing our work and for the **encouraging**, **constructive** feedback. We are pleased that the reviewer found our empirical analysis of the backdoor effect **interesting**, viewed the SAM-enhanced PSD as a **flexible**, **plug-and-play** enhancement, and considered our theoretical and empirical results **supportive** of the proposed mechanism. We address the remaining concerns in detail below.
>
> ---
>
> **W1. Concern regarding the availability of clean reference data for Stage-2 in realistic settings**
>
> **Response.** Thank you for highlighting the practical concern regarding data availability for Stage-2, which performs PCA and covariance whitening. We agree that assuming access to a pristine in-distribution reference set can be restrictive. To address this, we conducted additional experiments in more realistic settings where no explicitly clean reference data is available; results are reported in the appendix and summarized in Tables 1 and 2.
> - **Scenario 1: Out-of-distribution auxiliary data.** We simulate the case where only public but unverified data is available. For CIFAR-10, we use 250 samples per class from CIFAR-5m [2] as auxiliary data, which is out-of-distribution. As shown in Table 2, Stage-2 remains effective when whitening statistics are estimated from this OOD data, achieving **98.76% TPR on Blended**, comparable to using in-distribution data. This shows that Stage-2 **does not require in-distribution reference data**.
> - **Scenario 2: Identified-clean samples from the poisoned set.** We further consider the case where no external data is available. We apply META-SIFT [1] to the poisoned training set and extract 250 “identified-clean’’ samples per class as the reference for Stage-2. Although this subset is not guaranteed to be perfectly clean, Table 2 shows that performance remains high, for example **98.85% TPR on Blended**, indicating that Stage-2 **can operate by self-sifting the training data**.
> - **Scenario 3: Minimal clean data ratio.** When a truly clean auxiliary set is available, the required quantity can be very small. Table 1 reduces the reference set to **25 samples per class** (a **0.5% clean ratio**) and still obtains state-of-the-art performance across Blended, SSBA, and LF, showing that our method is **highly data-efficient**.
>
> **Conclusion.** The PCA and whitening in Stage-2 are **robust to both data quality and data source**. Whether we use OOD data, self-filtered data, or a very small clean subset, our method consistently achieves strong detection and effectively neutralizes backdoor effects.
>
> [1] Meta-Sift: How to Sift Out a Clean Subset in the Presence of Data Poisoning? USENIX Security, 2023.
> [2] The Deep Bootstrap Framework: Good Online Learners Are Good Offline Generalizers. arXiv, 2020.
>
> **Table 1.** TPR and FPR for SAM-enhanced detection methods under different attacks and clean auxiliary set ratios.
>
> |Attack, clean ratio|SCAn+SAM TPR|SCAn+SAM FPR|Beatrix+SAM TPR|Beatrix+SAM FPR|AC+SAM TPR|TIAC+SAM FPR|
> :-|:-:|:-:|:-:|:-:|:-:|:-:
> Blended, 0.5%|98.61|0.00|99.91|5.00|98.82|0.00
> Blended, 5%|98.74|0.00|99.65|5.00|98.82|0.00
> SSBA, 0.5%|96.31|0.00|99.91|5.00|96.47|0.00
> SSBA, 5%|96.52|0.00|99.42|5.00|96.51|0.00
> LF, 0.5%|95.91|0.00|98.83|5.00|96.02|0.00
> LF, 5%|96.16|0.00|99.01|5.00|96.12|0.00
>
> **Table 2.** TPR and FPR for SCAn and AC, with and without SAM, using identified-clean and OOD auxiliary data on CIFAR-10.
>
> |Attack, auxiliary data|SCAn / +SAM TPR|SCAn / +SAM FPR|AC / +SAM TPR|AC / +SAM FPR|
> :-|:-|:-|:-|:-
> Blended, auxiliary|99.36 / 98.85|0 / 0.53|2.57 / 99.60|7.94 / 0.84
> SSBA, auxiliary|94.18 / 97.41|0 / 0.53|100 / 96.84|3.84 / 0
> LF, auxiliary|94.77 / 96.31|0 / 0|95.80 / 96.34|10.89 / 0
> Blended, OOD|99.08 / 98.76|0 / 0|0 / 98.76|0 / 0
> SSBA, OOD|94.04 / 96.48|0 / 0|99.36 / 96.52|0 / 0
> LF, OOD|94.96 / 96.04|0 / 0|95.48 / 96.04|0 / 0

---

> ### Author Response · Authors · 2025-11-21
> **Response(Part 2)**
>
> **W2. Concern that SAM’s doubled training cost limits scalability to large-scale models**
>
> **Response.** We agree that standard SAM roughly doubles the training cost, which is problematic for large-scale models. Our main point is that this overhead can be **essentially removed** by adopting an efficient variant, **Momentum-SAM (MSAM)** [1], which our experiments show delivers SAM-level detection with almost SGD-level cost.
> - **Cost of standard SAM vs. MSAM.** Standard SAM computes an explicit perturbation and therefore requires two forward–backward passes per step. In contrast, MSAM constructs the perturbation from the **accumulated momentum**, so each update still uses only **one forward–backward pass**, similar to SGD. The complexity summary in Table 4 shows that MSAM’s theoretical cost is only slightly higher than SGD.
> - **Empirical validation.** In the appendix, we compare SGD, SAM, and MSAM on ResNet-18 with CIFAR-10 (RTX 4090). As shown in Table 4, training time increases from 1165 s (SGD) to 2266 s (SAM), whereas MSAM finishes in **1253 s** (**1.07×** SGD). Table 3 shows that MSAM-enhanced detectors retain the strong detection performance of standard SAM, e.g., **94.9% TPR on Blended**, close to 94.6% for SAM and far above 32.9% for the base detector.
> - **Preserving the defense mechanism.** These results are consistent with our theoretical Proposition 1: the defense relies on the **ascent–descent mechanism** that amplifies backdoor traces in feature space. MSAM preserves this mechanism while avoiding the extra backward pass.
> **Conclusion.** While standard SAM doubles the training cost, **MSAM offers a scalable and efficient alternative** that keeps training time close to SGD while preserving the defense performance, making SAM-enhanced PSD practical for large-scale models.
>
> [1] Moorgathu et al. Momentum-SAM: Gradient Optimization with Momentum-based Sharpness-Aware Minimization.
>
> **Table 3.** Detection performance of MSAM.
> |Attack|Detector|TPR, Base / +SAM / +MSAM|FPR, Base / +SAM / +MSAM|
> :-|:-|:-|:-
> BadNets|SS|70.8 / 92.3 / **91.9**|2.4 / 1.2 / **0.0**
> Blended|SS|32.9 / 94.6 / **94.9**|4.4 / 1.1 / **0.0**
> SSBA|SCAn|93.9 / 96.5 / **96.7**|0.0 / 0.0 / **0.5**
>
> **Table 4.** Computational complexity and training time.
> |Method|Complexity|Training time (s)|Relative time|
> :-|:-|:-:|:-:
> Conventional SGD|O((N/B)E(f + b + p))|1165|1.00×
> SAM, original|O(2(N/B)E(f + b + 3p))|2266|1.95×
> Momentum-SAM, efficient|O((N/B)E(f + b + 4p))|1253|1.07×
>
> ---
>
> **W3. Concern regarding the limited analysis of adaptive attackers and the potential exploitation of SAM**
>
> **Response.** We fully agree that robustness against adaptive attackers is crucial. The appendix provides a dedicated adaptive-attack evaluation; here we summarize the threat model and main findings.
> - **Threat model.** We assume a **white-box adversary** who knows our full pipeline, including SAM. The attacker can design poisoned training samples and triggers, but once the poisoned dataset is delivered, they **cannot control the defender’s training dynamics**, such as batch ordering, stochastic augmentations, or the perturbation directions induced by SAM, which is standard for pre-training defenses.
> - **Adaptive attack design.** Under this model, we construct an adaptive attack that explicitly attempts to **neutralize SAM**. The attacker performs bi-level optimization: an inner loop that simulates SAM’s maximization step, and an outer loop that optimizes the trigger to **minimize TAC-based feature distance** between poisoned and clean samples under the simulated SAM training.
> - **Results.** Table 5 shows that, under this adaptive attack, standard detectors without SAM are almost completely bypassed, confirming that the adaptive attacker can hide from **standard feature analysis**. In contrast, when SAM is actually used during the defender’s training, detection **recovers dramatically**, reaching **TPR above 97%** and up to **100% TPR for Spectre+SAM**, with low FPR.
> - **Key insight.** The attacker can only approximate SAM offline and cannot reproduce the **true dynamic training trajectory** realized by the defender, which depends on stochastic elements during training. This mismatch prevents the attacker from exactly canceling SAM’s amplifying effect. In practice, **real SAM still amplifies poison traces**, pushing poisoned samples away from the clean distribution and enabling successful detection.
>
> **Conclusion.** Even against a strong adaptive adversary that explicitly optimizes SAM-resistant triggers, SAM-enhanced PSD remains highly effective, whereas standard methods fail, because the attacker cannot fully mimic the defender’s training dynamics.
>
> **Table 5.** Performance against adaptive attacks.
> |Detection|Trigger|TPR, Base / +SAM|FPR, Base / +SAM|
> :-|:-|:-|:-
> Spectre|optimized|1.33 / **100.0**|0.45 / **0.15**
> SS|optimized|0.00 / **99.33**|0.30 / **0.00**
> AC|optimized|0.00 / **100.0**|0.00 / **0.00**
> Beatrix|optimized|0.67 / **97.33**|5.01 / **5.01**

---

> ### Author Response · Authors · 2025-11-21
> **Response(Part 3)**
>
> **W4. Concern that experiments mainly focus on small-scale datasets and lack demonstrations on larger datasets or transformer-based models**
>
> **Response.** We appreciate the suggestion to evaluate our method on larger datasets and more modern architectures. To address scalability and generality, we add experiments on both a **larger-scale dataset** and a **transformer-based backbone**, while keeping the PSD pipeline unchanged.
>
> - **ImageNet-200 with ResNet-50.** We evaluate **ImageNet-200** with **ResNet-50** and SCAn as the base PSD, and obtain SCAn+SAM by simply replacing the optimizer with SAM. SCAn alone almost completely fails under both BadNets and Blended, with TPR close to 0% at near-zero FPR, confirming that PSD on large-scale data is very challenging. With SAM, SCAn+SAM achieves **52.2% TPR (BadNets)** and **83.2% TPR (Blended)** at **0% FPR**, showing that our training strategy scales from CIFAR-level datasets to ImageNet-200.
> - **CIFAR-10 with ViT-S/16.** We also test a transformer backbone, **ViT-S/16**, on CIFAR-10. Under both BadNets and Blended attacks, switching from SCAn to SCAn+SAM yields **10–15 percentage points TPR improvement**, while FPR remains below 1%, showing that our method is **not restricted to CNNs** and can be applied to self-attention architectures.
>
> **Conclusion.** The additional experiments on ImageNet-200 and ViT-S/16 provide concrete evidence that SAM-based PSD **scales to larger datasets and generalizes across backbone architectures**, addressing the reviewer’s concerns about scalability and generality.
>
> **Table 6.** Detection performance of SCAn and SCAn+SAM on ImageNet-200 and ViT-S/16.
>
> |Dataset and backbone|Attack|Method|TPR (%)|FPR (%)|
> :-|:-|:-|:-:|:-:
> ImageNet-200, ResNet-50|BadNets|SCAn|0.0|0.0
> ||SCAn + SAM|**52.2**|**0.0**
> ImageNet-200, ResNet-50|Blended|SCAn|0.0|0.0
> ||SCAn + SAM|**83.2**|**0.0**
> CIFAR-10, ViT-S/16|BadNets|SCAn|78.5|0.6
> ||SCAn + SAM|**93.7**|0.7
> CIFAR-10, ViT-S/16|Blended|SCAn|84.1|0.5
> ||SCAn + SAM|**95.6**|0.8
>
> ---
>
> **Q1 and Q3. Concern about the trade-off between separability and clean accuracy, and the performance of the intermediate SAM-trained model**
>
> **Response.** We thank the reviewer for these related questions. Below we clarify the pipeline and the effect on accuracy.
>
> - **Role of the SAM-trained model.** Our defense follows the standard pre-training paradigm. The SAM-trained model is used only as a **temporary probe** in the detection stage and is **never deployed**:
>   1. **Detection stage.** Train a probe model with SAM on the suspicious dataset to amplify separability between clean and poisoned samples and identify suspicious data.
>   2. **Retraining stage.** Remove detected poisoned samples and **retrain a fresh model from scratch** on the purified dataset. This final model is deployed.
> - **No need to sacrifice probe accuracy.** We use **standard SAM hyperparameters** rather than aggressive ones. SAM tends to improve generalization by seeking flat minima, and in our experiments the SAM-based probe maintains **clean accuracy comparable to or better than** an SGD probe, while enlarging the representation gap between clean and poisoned samples. This is beneficial for PSD: **a probe that better fits clean data also provides a more reliable reference to detect deviations**.
> - **Performance of the final retrained model.** After purification, we retrain the classifier and evaluate clean accuracy and attack success rate (ASR). Table 7 shows that for the challenging Blended attack, the baseline model on poisoned data reaches **99.7% ASR**, whereas our method reduces ASR to **2.7%**, while **clean accuracy stays at 94.3%**, essentially matching the 94.5% of a clean baseline. Similar behavior holds for BadNets and SSBA.
>
> **Conclusion.** The SAM-enhanced model serves as a detection probe that **improves both generalization and separability** under standard hyperparameters. The final retrained model on the purified dataset retains clean accuracy comparable to a clean baseline while strongly suppressing backdoor attacks, so we do not observe a detrimental trade-off between separability and accuracy.
>
> **Table 7.** Performance of the final retrained model on purified datasets, ResNet-18 on CIFAR-10.
>
> |Attack|Method|Clean accuracy (%)|Attack success rate (%)|
> :-|:-|:-:|:-:
> Clean|Standard training|94.5|0.0
> Blended|Ours, retrained|**94.3**|**2.7**
> BadNets|Ours, retrained|**94.4**|**0.5**
> SSBA|Ours, retrained|**94.2**|**1.2**

---

> ### Author Response · Authors · 2025-11-21
> **Response(Part 4)**
>
> **Q4. Concern regarding the interaction between SAM and perturbation- or topology-based detectors, and why SAM might suppress them**
>
> **Response.** We agree that understanding how SAM interacts with different categories of defenses is important. The appendix contains a detailed empirical analysis. **Contrary to the impression that SAM universally suppresses non-feature-based methods, our results show a heterogeneous effect.**
>
> - **Empirical impact.** As summarized in Table 9, SAM does not consistently degrade perturbation-based or topology-based defenses. Its impact depends on the attack and the detection mechanism:
>   - For **perturbation-based methods** such as STRIP, SAM actually **improves BadNets TPR from 85.61% to 93.33%**, indicating a positive interaction in this case.
>   - For **topology-based methods** such as TED, the effect is mixed: detection drops for some attacks (e.g., BadNets) but improves for others, e.g., **from 25.99% to 35.21% TPR under Blended**.
> - **Mechanism-level explanation.**
>   – **Feature-based defenses** rely on feature-space distances; SAM explicitly **amplifies separation** between clean and poisoned distributions, giving consistent gains.
>   – **Perturbation-based defenses** leverage prediction instability under input perturbations. SAM seeks flat minima and tends to **stabilize predictions**, which can weaken or strengthen such detectors depending on how poisoned and clean samples react.
>   – **Topology-based defenses** analyze data manifold structure. SAM smooths the decision boundary and feature manifold, which may hide some topological artifacts while highlighting others, leading to mixed outcomes.
>
> **Conclusion.** SAM acts as a **specialized amplifier of feature-space separability**, and its interaction with non-feature-based methods is **heterogeneous** rather than uniformly negative. Its strongest and most consistent synergy is with feature-based PSD, while some perturbation/topology-based methods can also benefit in specific settings.
>
> **Table 9.** TPR of different defense categories before and after applying SAM on CIFAR-10.
>
> |Attack|CD / +SAM TPR|STRIP / +SAM TPR|SentiNet / +SAM TPR|TED / +SAM TPR|
> :-|:-|:-|:-|:-
> BadNets|63.81 / **67.22**|85.61 / **93.33**|35.33 / 36.72|86.54 / 55.32
> Blended|4.21 / 14.45|71.02 / 59.81|25.99 / 22.88|25.99 / **35.21**
>
> -
>
> **Q5. Concern that SAM’s amplification of backdoor features may be a byproduct of enhancing sensitivity to stable patterns in poisoned data**
>
> **Response.** This is an insightful observation that aligns with recent theory. We interpret it through the **low-rank bias** of SAM.
>
> - **Low-rank bias of SAM.** Recent work [1] shows that the flat minima favored by SAM are associated with **low-rank feature representations**. By penalizing sharp minima, SAM compresses the feature space and encourages the model to rely on a small number of dominant, simple features, a form of **simplicity bias**.
> - **Backdoors as low-rank shortcuts.** In data poisoning, the trigger is a **simple, highly correlated shortcut** that is easier to learn than high-rank semantic features, and thus naturally aligns with SAM’s preference for low-rank patterns.
> - **Amplification mechanism.** When we apply SAM in the detection stage, the optimizer searches for parameters that maintain low loss in a neighborhood. Because the backdoor pattern is simpler than natural semantics, SAM tends to **preferentially amplify neurons associated with the trigger**, making backdoor directions dominant in the representation space. Poisoned samples then form a distinct cluster along these low-rank directions, which our PSD pipeline separates from clean data.
>
> **Conclusion.** The amplification of backdoor features is **not a random side effect**, but a direct consequence of exploiting SAM’s intrinsic low-rank bias. We deliberately leverage this bias to over-express the backdoor pattern and make poisoned samples easily detectable.
>
> [1] Sharpness-aware minimization leads to low-rank features. NeurIPS 2023.

---

> ### Author Response · Authors · 2025-11-21
> **Response(Part 5)**
>
> **Q6. Concern about multi-trigger settings and whether SAM can still form a coherent signal for detection**
>
> **Response.** We thank the reviewer for emphasizing this practical scenario. We explicitly evaluate SAM-enhanced PSD under both **single-trigger multi-target** and **multi-trigger multi-target** attacks, and explain why SAM still produces a coherent backdoor signal.
>
> - **Experimental setup.** On CIFAR-10 with ResNet-18, we follow a standard protocol:
>   – **Single-trigger multi-target (S-T):** one trigger pattern applied to several source classes, mapped to different targets.
>   – **Multi-trigger multi-target (M-T):** several source classes each use a distinct trigger and target.
>   We use a representative PSD baseline (Beatrix) as “Base PSD” and obtain “Base PSD + SAM” by swapping the optimizer.
> - **Detection performance.** Table 10 summarizes the average results. Multi-trigger attacks are clearly harder for the base PSD (TPR drops in the M-T setting). After adding SAM, however, the detector remains robust:
>   – S-T: **TPR improves from 64.1% to 95.2%** with FPR increasing only from 0.5% to 0.8%.
>   – M-T: **TPR improves from 57.3% to 92.7%** with FPR around 1%.
> - **Why SAM still forms a coherent direction.** SAM does not amplify each trigger’s raw pixels independently; it amplifies **feature directions that jointly support mapping poisoned samples to their target labels**. For a fixed target class, samples with different triggers must still lie in a similar decision region, inducing **shared discriminative directions** that respond to “any trigger → this target”. Gradients from all poisoned samples drive SAM to **amplify this shared backdoor subspace**, and our class-level TAC/PCA in Stage-2 further extracts these directions. Thus, even with multiple triggers, poisoned samples cluster along a **coherent backdoor subspace** that can be reliably separated from clean data.
>
> **Conclusion.** Our experiments and analysis show that SAM-enhanced PSD remains effective under both single-trigger multi-target and multi-trigger multi-target attacks. SAM amplifies a **shared backdoor representation tied to the target labels**, which remains coherent across different triggers and enables high TPR with low FPR.
>
> **Table 10.** Detection performance under single-trigger multi-target and multi-trigger multi-target attacks.
>
> |Setting|Method|TPR (%)|FPR (%)|F1 (%)|
> :-|:-|:-:|:-:|:-:
> Single-trigger multi-target|Base PSD|64.1|0.5|70.3
> |Base PSD + SAM|**95.2**|0.8|**96.1**
> Multi-trigger multi-target|Base PSD|57.3|0.6|63.1
> |Base PSD + SAM|**92.7**|1.0|**93.8**

---

> > ### Author Response · Authors · 2025-11-27
> > **Response**
> >
> > Dear Reviewers,
> >
> > I hope this message finds you well. I understand that you are very busy and truly appreciate the time and effort you devote to the review process.
> >
> > As the review deadline is approaching, we kindly wanted to check whether there are any updates regarding the reviews of our submission. If there is any additional information or clarification needed from our side, we would be more than happy to provide it.
> >
> > Thank you very much for your time and support. We sincerely appreciate your help and contributions to the community.
> >
> > Best regards,
> > Authors

---

### Comment · Area_Chair_sEKT · 2025-11-24

Please respond to the authors' rebuttal. Thanks.

AC

---

### Meta-Review · Area_Chair_kyqP · 2026-01-07

**Summary:**

The paper addresses an important and challenging problem in poisoned sample detection, particularly under weak backdoor attacks where existing methods often fail. The core idea of enhancing detection by amplifying backdoor signals via SAM is well motivated and supported by strong empirical results.Some reviewers raised concerns regarding limited novelty, computational overhead, and practical assumptions, such as the requirement for access to clean data and scalability to larger models. Additional questions were also raised about false positives and robustness against adaptive attackers. The rebuttal and additional experiments successfully addressed most of the technical and practical concerns, including efficiency, scalability, and robustness. Novelty remains the primary point of discussion. Overall, the strengths of the paper are judged to outweigh these concerns.

**Reviewer Concerns:**

Concerns largely addressed:
- Practical issues regarding clean reference data availability were mitigated through experiments using OOD data and self-filtered subsets.
- Computational overhead was partially addressed by introducing Momentum-SAM, reducing training cost relative to standard SAM.
- Additional experiments improved coverage on adaptive attackers, larger datasets, and transformer architectures.
- The authors provided a more detailed analysis of false positive rate behavior and showed that final retrained models maintain clean accuracy.

Concerns outstanding:
-  Despite extensive clarification, the novelty of the work is seen by some reviewer as limited,

**Reviewer Scores:**

-  Reviewer ELXT (initial: 4): Likely to increase, as most concerns were addressed.
-  Reviewer CppV (initial: 4): Likely to increase, as most concerns were addressed.
-  Reviewer bYYD (initial: 6): Likely to remain unchanged.
-  Reviewer GmvL (initial: 6): Likely to remain unchanged.

---

### Decision · Program_Chairs · 2026-01-26

Accept (Poster)